



# Lower-tropospheric $CO_2$ from near-infrared ACOS-GOSAT observations

Susan S. Kulawik[1], Chris O'Dell[2], Vivienne H. Payne[3], Le Kuai[4], Helen M. Worden[5], Sebastien C. Biraud[6], Colm Sweeney[7], Britton Stephens[7], Laura T. Iraci[8], Emma L. Yates[1], Tomoaki Tanaka[8]

(1) Bay Area Environmental Research Institute, Sonoma, CA, USA
(2) Cooperative Institute for Research in the Atmosphere, Colorado State University, Fort Collins, CO, USA
(3) Jet Propulsion Laboratory, California Institute of Technology, Pasadena, CA, USA
(4) UCLA Joint Institute for Regional Earth System Science and Engineering (JIFRESSE), Los Angeles, CA, USA
(5) Atmospheric Chemistry Observations & Modeling (ACOM) Laboratory National Center for Atmospheric Research, Boulder CO 80307 USA
(6) Lawrence Berkeley National Laboratory, Earth Science Division, Berkeley, CA, USA
(7) NOAA/ESRL/GMD, Boulder, CO, USA
(8) NASA Ames, Moffett Field, CA, USA

## Abstract

We present two new products from near-infrared GOSAT observations: lower tropospheric (LMT, from 0-2.5 km) and upper tropospheric/stratospheric (U, above 2.5 km) carbon dioxide partial columns. We compare these new products to aircraft profiles and remote surface flask measurements and find that the seasonal and year-to-year variations in the new partial columns significantly improve over the ACOS-GOSAT initial guess/a priori, with distinct patterns in the LMT and U seasonal cycles which match validation data. For land monthly averages, we find errors of 1.9, 0.7, and 0.8 ppm for retrieved GOSAT LMT, U, and $XCO_2$; for ocean monthly averages, we find errors of 0.7, 0.5, and 0.5 ppm for retrieved GOSAT LMT, U, and $XCO_2$. In the southern hemisphere biomass burning season, the new partial columns show similar patterns to MODIS fire maps and MOPITT multispectral CO for both vertical levels, despite a flat ACOS-GOSAT prior, and $CO/CO_2$ emission factor consistent with published values. The difference of LMT and U, useful for evaluation of model transport error, has also been validated with monthly average error of 0.8 (1.4) ppm for ocean (land). The new LMT partial column is more locally influenced than the U partial column, meaning that local fluxes can now be separated from $CO_2$ transported from far away.

## 1 Introduction





The Greenhouse Gases Observing Satellite (GOSAT) has been measuring
global satellite $CO_2$ columns since 2009, achieving accuracy to 0.3 ppm in
regional biases and 1.7 ppm single observation error (Kulawik et al., 2016).
The sensitivity of near-infrared radiances to $CO_2$ varies by altitude differently
in the strong and weak bands, resulting in the capability of retrieving
multiple pieces of vertical information from near-infrared observations, with
3+ degrees of freedom (i.e. independent pieces of information) for TCCON
(Connor et al., 2015; Kuai et al., 2012), 1.6 degrees of freedom for GOSAT
(this paper), and 2.0 degrees of freedom for OCO-2 (Kulawik, unpublished
result).  In this paper we use the intermediate retrieved profile from ACOS-
GOSAT processing to construct, bias-correct, and validate two partial
columns from near-infrared GOSAT observations (schematically shown in
Fig. 1).  The partially correlated errors and sensitivity of these two partial
columns are characterized so that they can be used for flux estimation and
other science analyses.
An important goal of carbon cycle research is to improve top-down estimates
of $CO_2$ fluxes, which use model assimilation to trace the observed variability
in the long-lived tracer backwards to sources and sinks.  Historically, such
top-down flux estimates have relied on surface observations (e.g. Peters et
al., 2007; Chevallier et al., 2010), though it was postulated 15 years ago
that satellite-based measurements of column $CO_2$ could dramatically reduce
top-down based flux uncertainties (Rayner and O'Brien, 2001; O'Brien and
Rayner, 2002).  Guided by this early work, most GOSAT analyses have
focused solely on total column $CO_2$ (or $XCO_2$).  Separation of $XCO_2$ into two
vertical columns has several advantages over column $XCO_2$ and surface
observations which should improve our ability to accurately estimate fluxes:
• flux estimates from column measurements rely on observations up to
continent-scales away (Liu et al., 2015; Feng et al., 2016); whereas
LMT back-trajectories show a more local influence of surface fluxes
versus for $XCO_2$, making flux estimates more data driven by local
measurements rather than relying on model transport, a major driver
of flux uncertianties (Houweling et al., 2015; Liu et al., 2015;
Chevallier et al., 2014; Liu et al., 2011; Prather et al., 2008)
• Stephens et al. (2007) show that measuring atmospheric values of
carbon dioxide at 2 vertical levels better constrains model transport
and partitioning between southern hemisphere land and northern mid-
latitude land fluxes, since vertical transport is an uncertainty in flux
estimates (Deng et al., 2015; Lauvaux and Davis, 2014; Stephens et
al., 2007)
• The LMT covers at least the entire boundary layer which partially
mitigates one source of flask assimilation error, the boundary layer





height (Denning et al., 1996; Gurney et al., 2002; Rayner and O'Brien,
2001); and
• GOSAT provides observations in many areas which are sparsely
covered by surface-based measurements.
In this work, we evaulate the precision and accuracy of these new LMT and
U partial column products derived from GOSAT, with the goal of providing
more and better information to the flux inversion estimates than is available
from the total column alone.  This paper is structured as follows.  We
introduce the datasets used in Section 2, and the theoretical basis in Section
3.  Section 4 describes methodology, e.g. the coincidence criteria and
GOSAT bias correction.  Section 4.1 uses back-trajectories to estimate the
distance to peak sensitivity to surface fluxes for LMT and U.  Section 5 shows
comparisons to aircraft observations and surface sites, including maps of the
two partial columns.  Section 5.4 shows patterns of the two partial columns
versus MOPITT multi-spectral CO retrievals, and Section 5.5 looks at errors
of LMT minus U.  Section 6 discusses and summarizes these results.
**2. Datasets**
There are two datasets used for validation of the new partial columns.
Aircraft profiles, which fly from the surface to somewhere between 5 and 13
km, can be used to directly validate what is seen with the two GOSAT partial
columns.  The second dataset that is used is from remote surface flasks,
which are used to compare to the lower GOSAT partial column, assuming
that 0-2.5 km is well mixed at remote sites.  TCCON, which currently
measures full columns, is used to diagnose discrepancies between aircraft
and GOSAT at the sites where both exist.  Additionally, we compare signals
from burning and transport in southern hemisphere land from GOSAT,
MOPITT, and MODIS fire counts.  Figure 2 shows aircraft and surface
validation locations, along with GOSAT coincidences.
**2.1 GOSAT**
The Greenhouse gases Observing SATellite (GOSAT) takes measurements of
reflected sunlight in three shortwave bands with a circular footprint of
approximately 10.5 km diameter at nadir (Kuze et al., 2016; Yokota et al.,
2009; Crisp et al., 2012).  ACOS-GOSAT v3.5 from the lite product with
quality flag of 0 are used along with the full profile, profile averaging kernel,
and full profile error matrices from ancillary GOSAT files.  We use both nadir
land observations (looking straight down) and ocean glint observations
(sunglint tracking mode), but not medium gain over land, as there is not
sufficiently co-located validation data to validate medium gain observations.
**2.2 ESRL aircraft profiles**



Aircraft and ocean measurements are obtained from an obspack
(co2_1_PROTOTYPE_v1.0.4b_2014-02-13).  The measurements are
extended down to the surface using the lowest measured value, and
extended to the tropopause using the aircraft value at the highest altitude
(The Tropopause is from NCEP,
http://www.esrl.noaa.gov/psd/data/gridded/data.ncep.reanalysis.html).  The
CarbonTracker model (CT2015, see below) is used to extend the profile
through the stratosphere.
**2.3 Remote oceanic surface in situ measurement sites**
Remote surface sites are from the Earth System Research Laboratory
Observation Package (ObsPack) Data Product surface flask measurements
(Conway et al., 1994), which are accurate to 0.1 ppm with excellent
coverage in the US and Europe and sparser coverage elsewhere.  For each
station, there can be different options, represented by file names.  The
"afternoon" file is preferred, as it best matches the satellite overpass.  If
"afternoon" is not found, "nighttime" and "marine" (which filters
observations by their source) are excluded, and "allvalid", "representative",
and "continental" are accepted.
The "remote oceanic" locations used are selected to have at least 97% ocean
at a 5 degree radius around the location.  The locations are shown in Fig. 2.
Although the airmass observed by GOSAT LMT will not match the airmass
observed by the surface site, the remote location should result in boundary
layer mixing that will make the comparisons useful.  Additionally, these sites
are not used in development of the bias correction terms (described in
Section 3.5 and Appendix A) and are an independent test of bias correction
for ocean.
**2.4 HIPPO aircraft profiles**
The HIAPER Pole-to-Pole Observations (HIPPO) project samples the
atmosphere in a series of profiles from the surface to 9-13 km, from about
80N to 60S. The campaigns covered different years as well as different
seasons, namely: HIPPO 1: January, 2009, HIPPO 2: November 2009, HIPPO
3: March-April, 2010, HIPPO 4: June-July, 2011, and HIPPO 5: August-
September, 2011.  Frankenberg et al. (2016) recently were successful in
evaluating satellite measurements of column $CO_2$ over ocean (including
GOSAT) using HIPPO. In this paper, we look at comparisons between GOSAT
and HIPPO 2-5 (HIPPO 1 occurs prior to GOSAT launch) using the HIPPO-
identified profiles and the CO2_X field, based on 1s data averaged to 10s,
from two (harmonized) sensors: CO2-QCLS and 15 CO2-OMS.  Due to the
GOSAT glint coverage span of about 40 degrees with additional screening for
the new products, many of the comparisons had fairly limited latitudinal
spans with the GOSAT improvement over the prior found more in improving



the bias rather than improving the standard deviation.  The combined
campaigns span a wide range of GOSAT latitudes.
**2.5 AJAX aircraft profiles**
The Alpha Jet Atmospheric eXperiment (AJAX) project collects in situ $CO_2$
vertical profiles from the surface to 8 km in several locations, including
Railroad Valley, NV; Merced, CA, and other locations in the West Coast.
Most of the version 4 profiles used in this paper were collected to coincide
with GOSAT overpasses. Trace gas instruments and the Meteorological
Measurement Sensor are housed in an unpressurized sensor pod that is
mounted under the wing. A cavity ring-down spectrometer (Picarro Inc.
G2301-m) which has been modified for flight conditions is routinely
calibrated to NOAA/ESRL gas standards. Calculated $1\sigma$ overall uncertainties
are 0.16 ppm for $CO_2$ (Hamill et al. 2016; Tanaka et al., 2016).
**2.6 MOPITT v6 multispectral CO retrieval**
In section 5, we utilize satellite-based CO observations from MOPITT to
attribute spatial variability in LMT and U. The MOPITT instrument on EOS-
Terra is in a sun-synchronous orbit with mean local time overpasses of
10:30 and 22:30. It has global coverage in ~3 days with a 22km x 22km
horizontal footprint. MOPITT uses gas filter correlation radiometry (GFCR) to
measure atmospheric CO at 4.6 µm (Thermal Infrared) and 2.3 µm (Short-
wave Infrared) and is the only satellite instrument capable of simultaneous
multispectral retrievals of CO with enhanced sensitivity to near-surface CO
for daytime/land observations (Worden et al., 2010). MOPITT CO data have
been validated for each retrieval algorithm version using aircraft in situ
measurements (Deeter et al., 2014). Here we use daytime only MOPITT V6J
(multispectral) data that have been filtered to require cloud free scenes from
both MOPITT and MODIS cloud detection. We also use a measure of
sensitivity to near-surface CO computed from the trace of the averaging
kernel for the bottom 2 pressure levels to select scenes that contain
relatively more information from the measurement.
**2.7 MODIS fire counts**
MODIS fire counts (found at https://lance.modaps.eosdis.nasa.gov/cgi-
bin/imagery/firemaps.cgi) are used to identify biomass burning locations.
Fire maps are created by Jacques Descloitres with fire detection algorithm
developed by Louis Giglio. Blue Marble background image created by Reto
Stokli (Giglio et al., 2003; Davies et al., 2004).
**2.8 CarbonTracker model**
CarbonTracker CT2015, http://carbontracker.noaa.gov, (Peters et al., 2007)
is used to extend aircraft profiles from the stratosphere to the top of the



atmosphere (similarly to in Frankenberg et al., 2016 and Inoue et al., 2013)
and to quantify co-location error (similarly to Kulawik et al. (2016).
**2.9 TCCON**
The Total Carbon Column Observing Network (TCCON) observations, version
GGG2014 (Wunch et al., 2011a) at Lamont (Wennberg et al., 2014) and
Park Falls (Wennberg et al., 2014), where both aircraft and TCCON
observations have co-located measurements, are used to evaluate $XCO_2$
calculated from the aircraft observations (extended as described by Section
3.7). Although the TCCON observations contain information to split into 2 or
3 vertical columns, the focus of the TCCON project has been on the
validation of $XCO_2$ from OCO-2 and GOSAT. Recent work by Kuai et al.
(2012), Dohe et al. (2012), and Connor et al. (2015) have explored vertical
profile retrievals from TCCON, but there is not yet an operational product.
**3.0 LMT and U theoretical basis**
In Section 3.1, equations are presented describing the sensitivity and errors
of the new products. In Section 3.2, a simulation is shown of what GOSAT is
expected to see from space using the developed equations and aircraft
profiles from the Southern Great Plains (SGP) aircraft site.
**3.1 Equations describing sensitivity and errors**
The ACOS retrievals (O'Dell et al., 2012) utilize an optimal estimation
approach with a priori constraints (Rodgers, 2000). It is common practice to
represent the state parameter to be retrieved on an altitude grid that is finer
than the altitude resolution of the instrument (e.g., Bowman et al., 2006;
Deeter et al., 2003; von Clarmann et al., 2003). A major advantage of this
approach is that it allows the calculation of diagnostics, such as averaging
kernels, which can be used to characterize the sensitivity of the
measurement. Constraints (regularization) must be applied in order to
stabilize the retrieval (e.g., Rodgers, 2000; Tikhonov, 1963; Twomey, 1963;
Steck and von Clarmann, 2001; Kulawik et al., 2006). The constraints
include a constraint vector and a constraint matrix, which may be chosen
constrain absolute values and/or the shape of the retrieved result.
In the ACOS processing, $CO_2$ is first retrieved as a 20-level profile, which
averages 1.6 degrees of freedom (DOF) with about 0.8 DOF below 2.5 km
above the surface (levels 16-20, where level 20 is the surface), and 0.8 DOF
above 2.5 km (levels 1-15). This intermediate profile has significant
altitude-dependent biases and cannot be used scientifically as-is, but rather
this profile is compacted to a single column quantity, $XCO_2$ as the final step
in the ACOS processing. In this work, we post-process the ACOS-GOSAT
intermediate profile to calculate and characterize the partial column
represented by levels 16-20, which is named $LMT\_XCO_2$ or LMT for short,



and the partial column represented by levels 1-15, which is named U_XCO$_2$
or U for short.

The equation for the linear estimate of **x**, the retrieved CO$_2$ profile (Connor
et al., 2008; Rodgers, 2000) is:

$$\hat{\mathbf{x}} = \mathbf{x}_a + \mathbf{A}_{xx}(\mathbf{x}_{true} - \mathbf{x}_a) + \mathbf{A}_{xy}(\mathbf{y} - \mathbf{y}_a) + \mathbf{G}_x \boldsymbol{\varepsilon}$$    (1)

where $\hat{\mathbf{x}}$ is the retrieved CO$_2$ profile with size $n_{CO2}$ (20 for ACOS-GOSAT),
$x_{true}$ is the true value, $\mathbf{A}_{xx}$ is the $n_{CO2}$ x $n_{CO2}$ CO$_2$ profile averaging kernel, and
$\mathbf{A}_{xy}(\mathbf{y} - \mathbf{y}_a)$ is the cross-state error representing the propagation of error from
non-CO$_2$ retrieved parameters **y** (aerosols, albedo, etc., size 26 for ACOS-
GOSAT as there are 26 non-CO$_2$ retrieved parameters) into retrieved CO$_2$,
size $n_{interf}$ x $n_{CO2}$; and $\mathbf{G}_x \boldsymbol{\varepsilon}$ is the measurement error.

The pressure weighting function, "**h**" (size $n_{CO2}$) is used to convert the
retrieved CO$_2$ profile to XCO$_2$ by tracking each level's contributes to the
column quantity.

**h**$_{xco2}$ = [0.026 0.052 0.052 0.052 … 0.052 0.052 0.052 0.026]    (2a)

The sensitivity to the top or bottom levels is half other levels, as these levels
contribute to only one layer, rather than two adjacent layers.  The GOSAT
levels are chosen such that the pressure weighting is the same for all layers
and the same for all observations.

The LMT pressure weighting function is obtained by starting with the
pressure weighting function for XCO$_2$, setting levels 1-15 to zero, then
normalizing so that the sum of all entries adds to 1.  For the U pressure
weighting function, levels 16-20 are set to zero, then the vector is
normalized so that the sum is 1.  The LMT and U pressure weighting
functions are:

**h**$_{LMT}$ = [0 0 0 0 … 0 0.22 0.22 0.22 0.22 0.11]    (2b)
**h**$_U$ = [0.035 0.069 0.069 … 0.069 0.069 0 0 0 0 0]    (2c)

To calculate XCO$_2$, the equation is:

XCO$_2$ = **h**$_{xco2}$ · $\hat{\mathbf{x}}$    (3)

The fraction of total air in each of the partial columns are:

$f_{XCO2}$ = 1    (4a)



$f_{LMT} = 0.23446$ (4b)
$f_U = 0.76554$ (4c)

Combining Eqs. 1, 2a, and 3, the $XCO_2$ estimate is:

$(X\hat{C}O_2) = (XCO_2)_a + \mathbf{h}_{XCO2}\mathbf{A}_{xx}(\mathbf{x}_{true} - \mathbf{x_a}) + \mathbf{h}_{XCO2}\mathbf{A}_{xy}(\mathbf{y} - \mathbf{y_a}) + \mathbf{h}_{XCO2}\mathbf{G_x}$ (5a)
$(X\hat{C}O_2) = (XCO_2)_a + \mathbf{a}_{xx}(\mathbf{x}_{true} - \mathbf{x_a}) + \mathbf{a}_{xy}(\mathbf{y} - \mathbf{y_a}) + \mathbf{g_x}\boldsymbol{\varepsilon}$ (5b)

where $\mathbf{a}_{xx}$ is the column averaging kernel, $\mathbf{a}_{xx} = \mathbf{h}_{XCO2}{}^T\mathbf{A}_{xx}$ (see Appendix A of
Connor, 2008).

Similarly, to calculate the linear estimate for the 2-vector [LMT, U], Equation
1 is multiplied by the 2 x $n_{CO2}$ pressure weighting function, $\mathbf{h} = [\mathbf{h}_{LMT}, \mathbf{h}_U]$:

$$\begin{pmatrix} L\hat{M}T \\ \hat{U} \end{pmatrix} = \begin{pmatrix} LMT \\ U \end{pmatrix}_{\mathbf{a}} + \mathbf{h}\mathbf{A}_{xx}(\mathbf{x}_{true} - \mathbf{x_a}) + \mathbf{h}\mathbf{A}_{xy}(\mathbf{y} - \mathbf{y_a}) + \mathbf{h}\mathbf{G_x}\boldsymbol{\varepsilon}$$ (6a)
$$\begin{pmatrix} L\hat{M}T \\ \hat{U} \end{pmatrix} = \begin{pmatrix} LMT \\ U \end{pmatrix}_{\mathbf{a}} + \mathbf{a}_{xx}(\mathbf{x}_{true} - \mathbf{x_a}) + \mathbf{a}_{xy}(\mathbf{y} - \mathbf{y_a}) + \mathbf{g_x}\boldsymbol{\varepsilon}$$ (6b)

where now $\mathbf{a}_{xx} = [\mathbf{h}_{LMT}, \mathbf{h}_U]\mathbf{A}_{xx}$, a 2 x $n_{CO2}$ matrix, $\mathbf{a}_{xy} = [\mathbf{h}_{LMT},\mathbf{h}_U]\mathbf{A}_{xy}$, a 2x
$n_{interf}$ matrix, and $\mathbf{g_x} = [\mathbf{h}_{LMT}, \mathbf{h}_U]\mathbf{G_x}$, a 2x$n_s$ matrix, where $n_s$ are the number of
of spectral points.

The last two terms in Eq. 6 represent the cross-state and measurement
error, respectively, and are often jointly called the observation error
(Worden et al., 2004). The error in [LMT, U] is estimated by taking the
covariance of $\begin{pmatrix} L\hat{M}T \\ \hat{U} \end{pmatrix} - \begin{pmatrix} LMT \\ U \end{pmatrix}_{True}$ . The errors can be calculated either from
taking the covariance (6a) or from (6b). The covariance of (6a) has a fairly
simple form, in terms of the standard definitions of the error covariances for
the full profile, $\mathbf{S}_{interf}$, and $\mathbf{S}_{meas}$ , which are included in the ACOS-GOSAT
ancillary products, and $\mathbf{S}_{smoothing}$ can be calculated from the standard
equation, $\mathbf{S}_{smoothing} = \mathbf{(I-A)S_a(I-A)^T}$ (Rodgers, 2000), with $\mathbf{A}$ the $n_{CO2}$ x $n_{CO2}$
$CO_2$ profile averaging kernel and $\mathbf{S_a}$ the a priori covariance, both included in
the ACOS-GOSAT products.

$\sigma_{[LMT,U]} = \mathbf{hS_{smoothing}h^T} + \mathbf{hS_{interfer}h^T} + \mathbf{hS_{meas}h^T}$ (7a)
$$= \begin{pmatrix} \mathbf{h}_{lmt}{}^T\mathbf{S}_{smooth}\mathbf{h}_{lmt} & \mathbf{h}_{lmt}{}^T\mathbf{S}_{smooth}\mathbf{h}_U \\ \mathbf{h}_U{}^T\mathbf{S}_{smooth}\mathbf{h}_{lmt} & \mathbf{h}_U{}^T\mathbf{S}_{smooth}\mathbf{h}_U \end{pmatrix} + \begin{pmatrix} \mathbf{h}_{lmt}{}^T\mathbf{S}_{obs}\mathbf{h}_{lmt} & \mathbf{h}_{lmt}{}^T\mathbf{S}_{obs}\mathbf{h}_U \\ \mathbf{h}_U{}^T\mathbf{S}_{obs}\mathbf{h}_{lmt} & \mathbf{h}_u{}^T\mathbf{S}_{obs}\mathbf{h}_U \end{pmatrix}$$ (7b)





$$= \begin{pmatrix} \sigma_{LMT}{}^2 & \sigma_{LMT}\sigma_U\,corr \\ \sigma_{LMT}\sigma_U\,corr & \sigma_U{}^2 \end{pmatrix}$$
(7c)


The resulting error covariance $\sigma_{[LMT,U]}$ is a 2x2 matrix, where the diagonals
are the square of the predicted error for each parameter, and the off
diagonals also depend on the correlated errors between these parameters.
Table 1 shows the predicted errors for LMT, U, and the error correlation
between LMT and U.  The predicted errors in Table 1 are larger than the
actual errors, seen in Tables 2 and 3; and the error for averaged
observations is estimated in section 4.1.1.  It is worth noting that the *a*
*priori* errors are much larger for LMT and U, at 34 and 9 ppm, respectively,
than the posterior errors, indicating that these quantities are largely
unconstrained by the retrieval's prior assumption.

Through the same process as Eqs 6-7, the $XCO_2$ error is:

$$\sigma_{XCO_2} = \mathbf{h}_{XCO\,2}{}^T \mathbf{S}_{smooth}\, \mathbf{h}_{XCO\,2} + \mathbf{h}_{XCO\,2}{}^T \mathbf{S}_{obs}\, \mathbf{h}_{XCO\,2}$$
(8)


$XCO_2$ can also be calculated as a function of LMT and U, and the $XCO_2$ errors
can be calculated as a function of the errors in [LMT, U].  These are shown in
Eq. 9.

$$XCO_2 = f_{lmt} LMT\_CO_2 + f_u U\_CO_2$$
(9a)


$$\sigma_{XCO_2} = \sqrt{ \begin{pmatrix} f_{lmt} & f_u \end{pmatrix} \begin{pmatrix} \sigma_{lmt}{}^2 & \sigma_{lmt}\sigma_u corr \\ \sigma_{lmt}\sigma_u corr & \sigma_u{}^2 \end{pmatrix} \begin{pmatrix} f_{lmt} \\ f_u \end{pmatrix} }$$
(9b)


$$\sigma_{XCO_2} = \sqrt{ 0.23^2 \sigma_{lmt}{}^2 + \sigma_u{}^2 0.77^2 + 2*0.77*0.23 \sigma_{lmt}\sigma_u corr }$$
(9c)


where $f_{LMT}$ and $f_U$ are the air masses for the LMT and U partial columns
(0.236, 0.764), respectively,  $\sigma_{lmt}$ is the error for LMT, and *corr* is the error
correlation between LMT and U.

The normalized column averaging kernel is used to see the sensitivity of the
column to the true state at different levels, with a value of 1 meaning
perfect sensitivity, and a value of 0 meaning no sensitivity.  The normalized
column averaging kernel is the column averaging kernel, **a**, divided by the
pressure weighting function for each layer, $\mathbf{h_{XCO2}}$, and multiplied by the
fraction of air in the partial column.

$\mathbf{a\_norm_{LMT}} = (\mathbf{h_{LMT}\, A_{CO2}})/\mathbf{h_{XCO2}}*f_{LMT}$ 
(10a)
$\mathbf{a\_norm_U} = (\mathbf{h_U\, A_{CO2}})/\mathbf{h_{XCO2}}*f_U$ 
(10b)






Figure 3 shows the normalized column averaging kernels for LMT, U, and
XCO$_2$ for land and ocean scenes.  Although the LMT partial column sums the
5 levels within about 2.5 km of the ground, the LMT has some sensitivity to
the true state at all 20 levels because the GOSAT radiances are not able to
fully resolve between CO$_2$ within the surface to 2.5 km versus above this.
As expected, the sensitivity for LMT plus U is equal to the sensitivity for
XCO$_2$, and the sensitivity for LMT is weighted to the surface whereas the
sensitivity for U is weighted to the top of the atmosphere.  The negative
averaging kernels for LMT in the stratosphere are partially a consequence of
the ACOS prior constraint, which allows no stratospheric variability.  Actual
stratospheric variability is transferred to the closest levels that are allowed
to vary, and the surface compensates for the radiance error induced by this,
resulting in a negative sensitivity of the LMT to the true state in the
stratosphere.  If the stratospheric truth matches that of the *a priori*, then
there will be no propagation of error into LMT or U.  The averaging kernels
shown in Fig. 3 are similar to those calculated for TCCON in Figure 2 of
Connor et al. (2015).

**3.2 Seasonal behavior of LMT, U, and XCO$_2$ estimated using *only***
**aircraft measurements and GOSAT sensitivity (no GOSAT**
**observations)**

This section answers the following questions: (1) Do U and LMT have unique
seasonal signatures?  (2) How much of the XCO$_2$ variability is due to LMT
versus U variability (3) How much does the prior influence the LMT and U
retrievals?

This section uses simulated GOSAT retrievals using the linear estimate, Eqs.
5 and 6, given the aircraft in situ profiles at the SGP site (37N, 95W), the
GOSAT prior, and the GOSAT averaging kernels.  Using this analysis, the
importance of the prior is assessed by using either a flat prior or the GOSAT
prior in Eqs. 5 and 6.  We assess how much LMT and U contribute to the
variations seen in XCO$_2$ using the variability of the LMT and U partial
columns combined with the weighting each has in the full column.  The
seasonal cycles of each partial column are studied by converting all aircraft
measurements to lie between 2012 and 2013 by applying a 2 ppm/year
secular trend, and averaging by month.  This method was used rather than
fitting the aircraft observations using the NOAA CCGCRV to estimate the
seasonal cycle shape because the aircraft observations are not sufficiently
smooth to result in a consistent fit.
Figure 4 shows the estimates of LMT, U, and XCO$_2$ using SGP aircraft profiles
calculated as described above.  There is significant variability in the
individual aircraft measurements, seen in panel (a) but this is smoothed out
on monthly timescales, seen in the remaining panels.  The dashed lines in





panel a represent fits using the NOAA fitting software CCGCRV. The U
partial column is rarely more than 1 ppm different from the fit, whereas LMT
can be up to 5 ppm different (e.g. see summer, 2009; January, 2010;
Summer, 2011).
Figure 4 (b) and (c) show the difference between simulated retrievals with
the GOSAT a priori (b) versus a flat a priori (c). The patterns are very
similar, indicating that the signal is primarily coming from the data rather
than the prior, with standard deviations of 0.8 ppm for LMT and 0.3 ppm for
U (these changes are fully characterized when applying the GOSAT prior to
the aircraft true profile with the specified a priori vector).
Figure 4, panel (d) shows U versus XCO2. At first glance these look the
same, but by comparing panel (d) and (b), the $XCO_2$ deviations from U are
towards LMT. The seasonal variabilities of $XCO_2$, LMT, and U (maximum
minus minimum), are 3.3 ppm, 4.8 ppm, and 3.3 ppm, respectively. Note
that the seasonal variations in LMT and U have a 0.56 correlation. A
straightforward calculation of variation * airfraction (Eq. 4) show that the
fraction of variation of $XCO_2$ resulting from variations in LMT is
approximately 30%, and the fraction of the variation in $XCO_2$ coming from U
variation is roughly 70%. So even though LMT has more variability, U has
the much larger impact on $XCO_2$ due to the fact that the full column is 77%
LMT. A similar calculation at Park Falls, where the LMT seasonal cycle is 20
ppm and the U seasonal cycle is 5 ppm finds 45% of the seasonal variability
in $XCO_2$ results from U and 55% from LMT at Park Falls (46N).
Figure 4 indicates that LMT and U do have unique seasonal cycles which
result from the data rather than the prior. The LMT partial column, which
contributes to 30% of the variations observed in $XCO_2$, has a much larger
seasonal variability than the U partial column or the $XCO_2$ column, and
earlier seasonal cycle maximums and minimums.
**4.0 Methods**
We test the sensitivity of the new products to surface fluxes using back-
trajectory footprints in Section 4.1. Section 4.2 discusses how GOSAT is
compared to aircraft. Sections 4.3-5.6 describe the bias correction, how the
aircraft data is extended to the full atmosphere and the coincidence criteria.
**4.1 Sensitivity of the LMT and U partial columns to surface fluxes**
To compare LMT and U sensitivity to surface fluxes, we look at 10-day back-
trajectory footprints created using WRF-STILT (Nehrkorn et al., 2010). The
"footprint" for an observation is a map of the surface locations to which an
observation is sensitive. Footprints are created for each of the 20 GOSAT
levels, then convolved with the LMT and U averaging kernels. The averaging



kernel estimates the sensitivity of the GOSAT measurement of each quantity
to the true state at each level. Footprint maps are created which show the
sensitivity of each type of GOSAT observation to sources and sinks. This
was done for 10 GOSAT observations in the Amazon. The distance for the
nearest 10% of footprints is 260 km for LMT and 790 km for U. It is likely
that there is also a very long tail in the U sensitivity, based on the work of
Liu et al. (2015) and Feng at el. (2016).
**4.2 Comparisons to aircraft**
The correct way to validate GOSAT estimates of [LMT, U] is to compare the
GOSAT observations to an estimate of what GOSAT should observe, given its
sensitivity, when the true atmospheric state is set to the aircraft $CO_2$ profile
using Eq. 6. The agreement should be within the GOSAT observation error,
as the smoothing term's effects on the comparison are removed by the
application of the GOSAT averaging kernel to the validation data.
**4.3 GOSAT bias correction**
The GOSAT standard $XCO_2$ product has regional biases and errors which can
be partially corrected using jointly retrieved parameters, pre-filters, or
radiance properties, e.g. the ratio of the signal in the strong vs. weak band,
retrieved albedo slopes or values, retrieved aerosol slopes or values; and
through post-processing screening, e.g. removing fits where the difference
in the retrieved versus prior surface pressure is greater than 4 hPa (). We
apply the same techniques to the LMT partial column in Appendix A which is
briefly described here. After LMT is corrected, the corrected U partial
column is set using Eq. 9a, so that $XCO_2$ is consistent with LMT and U.
To correct the LMT partial column, a set of pairs of "true" and "retrieved"
values is compiled, using validation data. GOSAT minus true is plotted
versus various GOSAT parameters described in Appendix A, and if a slope is
found for GOSAT error versus a parameter, a correction is applied. The
robustness of the correction is tested by applying the correction on data
withheld from the fit, as described in Appendix A. Following the initial bias
correction, GOSAT LMT is compared for closely occurring ocean and land
pairs; the a constant bias term is added to the land bias correction so that
land and ocean, on average, are consistent.
**4.4 Coincidence criteria**
"Geometric criteria", defined as +-3 degrees latitude, +-5 degrees longitude
+-1 week time are used to select coincident GOSAT observations for
particular sites. 5 degrees latitude/longitude, 1 hour has previously be used
for GOSAT criteria (Kulawik et al., 2016), however this did not yield enough
matches for aircraft profiles. With the above criteria, the total matches
range from 64 (PFA) to 4800 (SGP), with median 430, which is





approximately 9/month. A tight spatial criteria was selected to best capture
the seasonal cycle at a given location, especially for land where spatial
variability is large. Because aircraft and surface observations are more
infrequent than TCCON, extended time was used for the comparisons to
obtain sufficient comparison data. Other methods that were tried were
dynamic coincidence criteria (Wunch et al., 2011b) which considers a larger
area (+- 10 degrees latitude, +- 30 degrees longitude) but also matches
atmospheric temperature, and a variant of Basu criteria (Guerlet et al.,
2013), which used dynamic coincidences which had model-model differences
less than 0.5 ppm. All three criteria gave similar results overall, with
different criteria performing better at different stations, but no clear overall
best criteria. For HIPPO data, which mainly tests latitude gradients over
ocean, the dynamic coincidence approach was used following Frankenberg et
al. (2016)
**4.5 Extension of the aircraft profile**
The aircraft measurements go from the surface to between 5.5 km to 8 km
for most ESRL land to 9-13 km for HIPPO observations. As GOSAT LMT, U,
and $XCO_2$ have sensitivity above 5.5 km and even above 13 km, as seen in
the averaging kernel shown in Fig. 3, the aircraft profile needs to be
extended from the top measurement to the top of the atmosphere. Four
different methods of extension were tested: extending with the GOSAT
prior, extending the top aircraft measurement through the tropopause and
extending with the GOSAT prior above this, extending with the CT2015
model, and extending the top aircraft measurement through the tropopause
and extending with the CT2015 model above this. The different extensions
mainly had an effect on the overall LMT, U, and $XCO_2$ biases, rather than the
standard deviation, with a spread of 0.4 ppm, as seen in Table A4. The
extension that was used in the rest of the paper is extending the top aircraft
measurement through the tropopause and extending with the CT2015 model
above this.
**5. GOSAT results**
Figure 5 shows GOSAT comparisons for LMT and U versus the aircraft
measurements at the SGP site at 37N, 95W which can be compared to the
simulated results shown in Fig. 4. The GOSAT LMT and U products show the
same seasonal patterns as seen in the aircraft data. Figure 5b shows
CarbonTracker matched to GOSAT (CT@GOSAT ) and CarbonTracker
matched to the aircraft measurements (CT@aircraft). The difference of
CT@GOSAT and CT@aircraft estimates the co-location error. Large
differences are seen between CT@GOSAT and CT@aircraft in early 2010,
Summer, 2010, and Summer, 2011. In Fig. 5c, the seasonal cycle is shown
by transforming all data to lie within 2012 using 2 ppm/year adjustment to
$CO_2$. There are systematic differences seen in the drawdown, which is



underestimated by GOSAT.  However, when months that have differences of (CT@GOSAT -CT@aircraft) more than 2.5 ppm are removed, Figure 5d shows agreement within the GOSAT predicted errors between GOSAT and aircraft.

GOSAT U improves over the a priori for real (Figs. 5c-d) and in simulated (Fig. 4b) results.  The black (aircraft) vs. blue (GOSAT) in Fig. 5c shows much better agreement in July-November than prior (green) vs. black (aircraft).  The bias seen in the U partial column versus the aircraft U estimate is also found in $XCO_2$ versus the aircraft.

## 5.1 Summary of comparisons to all validation data

GOSAT LMT, U, and $XCO_2$ are compared to aircraft profiles, where the aircraft profile has the GOSAT averaging kernel applied so that the sensitivity is considered.  The comparison locations are shown in Fig. 2.  More detailed comparisons, showing results for each location and/or campaign, are shown in Appendix B.

Table 2 shows the biases with respect to aircraft data and Table 3 shows the standard deviation with respect to aircraft, for single and averaged observations.  The bias or standard deviation is calculated for every site (or campaign).  The mean represents the average of all site means, and the ± represents the standard deviation of all the sites (or campaigns).  The variability of the bias by location or time is a key metric in the data quality.  Biases that vary by season or location are the biggest detriment of use of satellite data (or any other type), as the assimilation will attribute these biases to spurious fluxes.

The co-location error is estimated by comparing CarbonTracker to itself at the satellite location/time and CarbonTracker at the aircraft location/time.  For the ocean surface sites, a vertical co-location error is estimated by comparing CarbonTracker with the LMT averaging kernel to CarbonTracker at the surface.  In Tables 2-4, the top entry in the ocean surface ct_ct difference is from discrepancies in horizontal location and time.  The bottom entry is the ct_ct difference between the CarbonTracker LMT quantity and CarbonTracker at the surface.

## 5.1.1 Bias

In Table 2, the "ct_ct bias" is the estimate of the co-location bias; it is the mean difference between CarbonTracker at the satellite location/time and CarbonTracker at the aircraft location/time.  The ct_ct bias is largest for aircraft land, with an overall bias of -0.6 ppm and bias variability of 0.7 ppm.  This gives an approximate best case of what could be achieved by GOSAT-AIRCRAFT comparisons.  An investigation of the -2 ppm ct_ct bias at CAR in



July, during the drawdown finds that the GOSAT observations are always
taken 3-4 hours later than the aircraft in July. The CarbonTracker model
estimates the effect of +3 hours as resulting in a -2 ppm change. The ct_ct
bias reflects spatial, diurnal, and seasonal co-location errors. Taking out the
5 sites that have errors > 0.5 ppm (see Appendix B, table b): WBI, BNE,
CAR, HIL, and CMA, reduces the ct_ct bias to -0.2 ± 0.3 ppm.

In Table 2, the "true mean by site/campaign" is the mean value averaged by
location (or campaign number). The ± represents the standard deviation of
the mean by location (or campaign number). The GOSAT retrieval must
improve on the ± at the very least to provide information on the
atmospheric state. The GOSAT a priori bias represents how well the GOSAT
prior does. The GOSAT prior improves over the true variability on land but
not for ocean cases for LMT. For U, the a priori variability to true is the
same size as the true variability. The "GOSAT bias" is the bias of the
retrieved GOSAT values. Comparing to the GOSAT prior, there is
improvement in all entries of the absolute bias, except $XCO_2$ for ESRL ocean,
and U, $XCO_2$ for AJAX. Issues with both U and $XCO_2$ suggests a possible
issue with the profile extension above the aircraft. The improvement in the
GOSAT ± bias occurs in all ocean comparisons and in LMT land. For ESRL
land, GOSAT LMT has an overall bias of -0.3 ppm and bias variability of 1.0
ppm. If the 5 stations with large ct_ct variability are taken out, the bias
decreases to -0.3 ± 0.7 ppm.

The location-dependent bias is important because this bias variability will be
attributed to phantom fluxes. The bias variability is 0.5 (1.0) ppm for
GOSAT LMT, 0.2 (0.9) ppm for U, and 0.4 (0.9) for $XCO_2$ ocean (land). The
LMT location dependent bias is no worse than the $XCO_2$ location dependent
bias, whereas the LMT signals are much more variable than $XCO_2$. The bias
variability for $XCO_2$ and U are possibly too high due to uncertainty of the
aircraft profile extension because the bias variability is much larger than the
0.3 ppm seen in Kulawik et al. (2016) versus TCCON. The variability of the
LMT ct_ct bias is 0.7 ppm, and when the 5 sites with ct_ct co-location error
larger than 0.5 ppm are taken out, the GOSAT LMT bias variability drops to
0.7 ppm. Taking out sites with large ct_ct biases for $XCO_2$ does not improve
the GOSAT $XCO_2$ bias variability. Taking out the top 4 GOSAT bias outliers
results in a GOSAT bias variability of 0.5 ppm for the remaining sites,
however these 4 sites are not the same sites where LMT has bias issues, nor
are they sites where ct_ct $XCO_2$ shows a large bias.

**5.1.2 Standard deviation**
Table 3 shows standard deviations of key quantities versus aircraft data.
The co-location standard deviation, which is estimated using the standard
deviation of CarbonTracker at the satellite location/time minus



CarbonTracker at the aircraft location/time, is less than 0.3 ppm for aircraft
ocean, 2.1 ppm for LMT land, 0.5 ppm for U land, and 0.8 ppm for $XCO_2$
land.  The surface ocean has 1.0 ppm co-location error, also including the
vertical co-location.  The AJAX comparisons, which are primarily from GOSAT
underflights, has a co-location error half that of the ESRL land matches,
which have coincidence criteria of 7 days, and 3-5 degrees.
The next entry is the predicted error, given by Eqs. 7 and 9, which is on the
order of 4.5 ppm for LMT, 1.7 ppm for U, and 0.7 ppm for $XCO_2$.  The actual
standard deviation of GOSAT versus aircraft, however, is about half that for
LMT and U, and double the predicted error for $XCO_2$.  This is discussed in
Section 5.1.5.
The next entry, "true variability" shows how much the different partial
columns vary seasonally.  The variability of LMT for land, at 5.4 ppm, is
about double that of U or $XCO_2$ (at 2.0 and 2.5 ppm, respectively), and the
variability for ESRL ocean, at 1.1 ppm, about 50% larger than U, XCO2 at
0.8 ppm.
The prior standard deviation (n=15), and GOSAT standard deviation (n=15)
look at the error of averaged GOSAT values, which is important for
assimilation.  Similarly to Kulawik et al. (2016), the GOSAT error does not
drop off as the inverse square root of the number of observations, like it
would if the error were fully random.  The error for 15 observation averages
is about 0.4 times that of 1 observation for land, with a similar factor for all
quantities; and about 0.5 times that of 1 observation for ocean, similarly for
all quantities.  Note that the co-location error has been subtracted out (in
quadrature) for both the a priori and GOSAT errors.
The standard deviations for LMT, U, and $XCO_2$ improve over the prior for
land cases but improve only marginally or do not improve over ocean.  The
location-dependent bias, however, does show improvement for LMT and U in
Table 2.  For surface ocean sites, which are only compared to LMT, the
improvement over the prior is much better, mainly because the prior is not
very good at these sites.
**5.1.3 Errors separated into co-location, random, and correlated error**
The errors between aircraft and GOSAT observations can be parametrized by
the number of GOSAT observations that are averaged.  Kulawik et al. (2016)
found the form in Eq. 11 matched well to the observed errors.
$$error = \sqrt{(a^2 + b^2/n)} \tag{11}$$

$$error = \sqrt{(\varepsilon_{coloc}^2 + a_o^2 + b^2/n)} \tag{12}$$






where $n$ are the number of GOSAT observations that are averaged (all of the
averaged observations match a single aircraft measurement), $a$ is error that
does not reduce with averaging, and $b$ is the random error. $a$ is further split
into co-location error, $\varepsilon_{coloc}$, plus $a_o$, the correlated error in Eq. 12. Correlated
error means that no matter how many observations are taken, this error
does not reduce, and can be due to interferents or spectroscopy in
combination with attributes specific to different locations and times.

The co-location error is the error resulting from imperfect matching of the
aircraft and satellite observations, and is approximated by the standard
deviation of the CarbonTracker model at the validation location and time and
the model at the satellite observation location and time, and is tabulated in
Table 4. This term, as seen in Table 4, is comparable to or even larger than
$a$ for LMT land cases. Some co-location schemes (e.g. as implemented by S.
Basu described in Guerlet et al. (2013)) use the model-model differences to
select the best satellite observations to match validation data. Equation 11
is used to determine a and b, and then $a_o$ is calculated from $a$ and $\varepsilon_{coloc}$.

The co-location error is subtracted from the correlated error, to try to
remove the effect of co-location on the error estimate. The three quantities
from Eq. 12 are shown in Table 4. For LMT the co-location error is about the
same size as the correlated error for ocean, and the co-location error is
larger than correlated error for land. For U and $XCO_2$, the correlated errors
are larger than the co-location error for ocean, and comparable for land.

**5.1.4 Comparison of $XCO_2$ results to previous results**
We compare GOSAT $XCO_2$ comparisons to the previous validations using
TCCON (Wunch et al., 2011b; Kulawik et al., 2016) and HIPPO observations
(Frankenberg et al., 2016). The GOSAT comparisons to HIPPO in
Frankenberg et al. (2016) were for at least 6 averages and did not subtract
co-location error (which is only 0.1 ppm over ocean). Using Eq. 12 and
Table 4, we find that the $XCO_2$ error for n=6 is 0.43 ppm, in agreement with
0.45 from Frankenberg et al. (2016). We would expect the same result, as
we are comparing to the same dataset with the same coincidence criteria.
Without co-location error, the $XCO_2$ from n=6 is 0.42 ppm. For ESRL land,
several quantities in Tables 2-4 can be directly compared to previous
GOSAT/TCCON validation: the co-location error (0.8 ppm) is larger than co-
location for geometric coincidence (0.4 ppm) but smaller than for dynamic
coincidence (0.9 ppm) from Kulawik et al. (2016). This makes sense as
Kulawik et al. (2016) had a 1 hour coincidence with TCCON whereas 7 days
is used in this paper, since aircraft measurements are sparser in time than
TCCON observations. $a_o$ and b values of 0.7±0.5 ppm and 1.6±0.2 ppm in





this work are consistent with 0.8 ±0.2 ppm and 1.6±0.1 ppm, for a
(corrected) and b, respectively, from Kulawik et al. (2016) Table 2.
Additionally, the predicted error of 0.9±0.1 which is a factor of 1.9 less than
the actual error of 1.7±0.4 are identical to the values and relative sizes of
predicted versus actual error in Kulawik et al. (2016) at the end of section
3.1. Kulawik et al. (2016) estimated that the XCO2 location-dependent bias
was 0.3 (after removing outlying stations north of 50N and locally-influenced
stations).
As discussed in Section 5.1.1, the location-dependent bias found in Kulawik
et al. (2016) versus TCCON sites was 0.3 (after removing outlying stations
north of 60N and locally-influenced stations). In this paper, we find this
variability to be 1.0 ppm. One reason for the discrepancy could be from the
extension of the profile above the aircraft measurement (about 5-6 km).
However, as seen in Table A4, extension of the aircraft profile by 4 different
methods did not make more than 0.1 ppm difference in the bias variability of
U. Another possible cause for the discrepancy is that GOSAT has been
extensively tested against TCCON and issues that show up at TCCON
locations have been previously addressed. This was tested by fitting bias
correction factor for U specifically, rather than calculating bias-correction
factors for LMT and subtracting the LMT partial column from GOSAT XCO2 to
estimate U. The bias variability for U did not improve when bias correction
factors were calculated directly for U. In Section 4.2, discrepancies between
the XCO2 calculated from the aircraft and TCCON XCO2 are seen in locations
where both measurements are co-located, so this does point towards the
reason being the profile extension.
**756 5.1.5 Predicted and actual error correlations**
One surprising finding is that LMT and U actual errors are less than the
predicted errors whereas the actual XCO2 errors are larger than predicted,
even though all three errors are calculated from the same error covariance
(see Eqs. 7-8). Equation 9c relates the errors in LMT, U, and XCO2. For
land, the XCO2 predicted error of 0.9 ppm is consistent with LMT predicted
error of 4.6 ppm, U error of 1.8 ppm, and error correlation of -0.8. The
XCO2 actual error (1.7 ppm) is much *larger* than the predicted error whereas
the LMT and U errors are *smaller* than predicted.
The discrepancy between the actual and predicted errors arises from the
actual correlation of the LMT and U partial column errors. The predicted
error correlation between LMT and U is -0.8. This means that values too low
in LMT should be matched with values too high in U, such that the total
column should have lower relative errors than either partial column
separately. The actual error correlation of (LMT-aircraft) and (U-aircraft)
average +0.6, meaning that when LMT is high, U also tends to be high, and





$XCO_2$ does not gain precision when combining LMT and U. So the finding is
that the LMT-U error correlation must be changed from the predicted value
of -0.8 to the measured value of +0.6. When this error correlation is
changed and all error terms multiplied by approximately *0.6, the predicted
and actual errors of LMT, U, and $XCO_2$ are consistent. Over ocean, the error
correlation is the same, but the multiplication factor is
The errors in Table 3 are the standard deviation with respect to validation
data at one location. The persistent regional biases captured in the "GOSAT
bias" standard deviation also reflect errors in the GOSAT measurement and
should somehow be combined into the full error. These regional biases likely
result from persistent interferent errors, such as due to aerosols, or an
interaction between spectroscopic errors and local conditions. Some but not
all of the bias, particularly for LMT land, can be attributed to co-location
error (see bias ct_ct (ppm) in Table 3). The correlation of the LMT and U
location-dependent biases (using biases separated by location from Table
B1) is also positive, 0.6, similar to the correlation of the individual errors in
LMT and U, so this would not account for the discrepancy between the
predicted correlation of -0.8 and actual correlation of 0.6 between the LMT
and U errors. Another possible reason for the positive error correlation in
LMT and U is that it is a consequence of the bias correction. The error
correlation on the uncorrected data was found to be -0.8, which supports
that the bias correction modifies the error correlation between U and LMT.
In summary, the single-sounding errors of GOSAT LMT and U over land
(ocean), based on the ESRL aircraft comparison, are 3.4 and 1.3 ppm (1.5
and 0.8 ppm) respectively, with a positive correlation of 0.6. This is
consistent with the $XCO_2$ error of 1.8 (1.0) ppm for land (ocean), using Eq.
9c. To find the error of averaged LMT and U, the single-sounding errors can
be replaced by Eq 11, with a and b values given in Table 4, and the same
LMT-U correlation of 0.6.
**5.2 Variability within the U.S**.
The CarbonTracker model identifies 19 eco-regions within North America
([http://www.esrl.noaa.gov/gmd/ccgg/carbontracker/CT2011_oi/documentati](http://www.esrl.noaa.gov/gmd/ccgg/carbontracker/CT2011_oi/documentati)
[on_assim.html](http://www.esrl.noaa.gov/gmd/ccgg/carbontracker/CT2011_oi/documentation_assim.html)). The ESRL aircraft stations can be broadly grouped into
conifer forest: pfa, etl, esp, thd; grass/shrub: car, bne; crops: hil, wbi,
sgp; forest/field: dnd, lef, nha, cma, sca; and mixed: tgc. The variability
at these sites is a combination of the local activity at the site, latitude of the
site, and transport into/out of the site.
Maps of GOSAT LMT, U, and $XCO_2$ along with aircraft, surface, tower, and
TCCON observations for February and July are shown in Fig. 6. In February,
the lower troposphere has already reached near peak values, whereas the U





partial column is continuing to increase through April. In July, there is a
large gradient in the LMT, primarily west to east, but also north to south,
seen also in the stations shown in Fig. 6. The LMT pattern agrees with
aircraft and tower patterns, showing that GOSAT LMT is able to see
variations in the summertime $CO_2$ depletion near the surface due to
biospheric processes. The U partial column shows more discrepancies with
aircraft than LMT which is in general agreement, and the same pattern of
discrepancies are also seen for $XCO_2$ versus aircraft. At the two sites where
aircraft and TCCON are jointly observed, SGP in Oklahoma and LEF in
Wisconsin, $XCO_2$ agrees with TCCON rather than the aircraft. This indicates
an issue with the extension of the aircraft profile from the top aircraft
measurement (about 6 km) to the top of the atmosphere.
Figure 7 shows the seasonal cycle at 5 sites arranged west-to-east (a-e) and
north-to-south (f-j). The seasonal cycle amplitude in LMT increases
traversing the region in both west-to-east and south-to-north directions.
There is also a shift to later in the seasonal cycle minimum going either west
to east and north to south. The seasonal cycle maximum is harder to
quantify for the LMT. The LMT $CO_2$ rises and stays fairly flat from January to
April, therefore the maximum can be influenced by small temporal variations
in the data, in contrast to U or $XCO_2$ which rises steadily until April.

### 5.3 Comparisons to remote ocean sites
Remote surface sites are useful as comparisons to LMT as these locations
are expected to be fairly well mixed in the boundary layer. These
comparisons are not used for estimating errors or bias corrections because
there could be a mismatch in airmass. Comparisons to surface sites are
direct comparisons without the averaging kernel applied, as there is no
profile information at these sites. The vertical co-location error is estimated
by comparing CarbonTracker LMT versus CarbonTracker surface values in
Tables 2 and 3. The GOSAT LMT a priori is significantly worse for these
locations as compared to North America, and this allows the GOSAT product
to show what is in the data versus the prior. In Table 4, the correlated error
for surface sites is higher than for ocean aircraft comparisons (1.0 ppm vs.
0.3 ppm, respectively), and the mean bias is also higher (0.7 ppm vs. 0.1
ppm, respectively). Because of the GOSAT ocean coverage, there are
typically only about 4 consecutive months for each station, but this is
adequate to evaluate the performance. Figure 8 shows an average over all
locations, and the 4 sites with the highest number of matches, arranged
from north to south. Note the improvement of GOSAT (red) over the a priori
(green) when comparing to the surface site measurements (pink).
Unsurprisingly, the performance of $XCO_2$ (blue) shows that surface site
observations are not suitable for $XCO_2$ validation. GOSAT LMT improves
over the prior in terms of the overall bias, the bias variability, and the



### 5.4 Source versus outflow in biomass burning with comparisons to MOPITT CO

The SH region is of particular interest for validation as the GOSAT prior is nearly spatially and vertically constant, varying primarily by month. GOSAT LMT and U partial columns are compared to MOPITT multispectral CO retrievals and MODIS fire counts for February, August, and September, 2010 in Figs. 9 and 10. The GOSAT prior (left column in Figs 9-10) is nearly constant in the southern hemisphere. The scale needed to span the seasonal range is about 13 ppm, about half that needed for the variability in the U.S. The pattern seen in LMT matches MODIS fire count images, shown in the right column, and matches MOPITT near-surface CO shown in the middle column. Because of the different overpass time and the different coverage due to cloudiness between these satellites, an exact match should not be expected. Also, note that multi-spectral CO does not have surface sensitivity over ocean. In February, sub-Saharan Africa has fires and south-central Africa does not, whereas the situation is reversed in August. This pattern is seen in GOSAT LMT, MOPITT near-surface, and MODIS fire counts.

In the mid-troposphere, MOPITT CO shows enhancement in sub-Saharan Africa in February, central Africa in August, and outflow in October, and GOSAT retrieved U shows the same patterns as MOPITT. Interestingly, both MOPITT and GOSAT show no enhancement in South America in August, whereas the surface has very strong enhancements in both.

The LMT signal in the Amazon region is clearly visible by May (not shown), whereas the CO signal seen from MOPITT (http://www.acom.ucar.edu/mopitt/MOPITT/data/plots6j/maps_mon.html) seems to ramp up starting in August. We look at the $\Delta CO/\Delta CO_2$ emission ratio in May and August to check the enhancements seen in LMT relative to MOPITT in these two months has a similar ratio is seen both months. The enhancements and background values for surface CO and LMT $CO_2$ are shown in Table 5.

The emission ratio estimate is calculated by taking the ratio of MOPITT multispectral CO divided by GOSAT LMT, each with their background subtracted. The raw emission ratio (which does not consider the sensitivity) is 4.7% for May, 2010 and 5.8% for August, 2010, using the degrees of freedom (DOFs) > 0.3 data. If "all MOPITT" data is used, the raw emission ratio for May and August, 2010 is 2.2% and 2.3%, respectively. The ratio dropping to half makes sense because the average MOPITT DOFs drops by half between these two categories. Based on the relative DOFs between





MOPITT and GOSAT, the emission ratios are likely about 10%, however, the
dropoff of the vertical sensitivity would also need to be taken into account.
Without utilizing a model as a transfer function, the exact ratio cannot be
estimated, due to the varying sensitivities with altitude and observation
locations and times.  The emission ratio seen by the MOPITT and GOSAT
LMT products are consistent with those estimated from aircraft observations
over tropical forests by Akagi et al. (2011, Table 1), which is 8.8%.  The
ratio is consistent (within 2%) between May and August, and the ratio is
consistent with aircraft observations.  If, instead, the emission ratio were
calculated from column $XCO_2$ and CO, the free troposphere and stratospheric
variations in CO and $CO_2$ would need be either zero or solely from locally
influenced fires.
**5.5 Differences between LMT and U**
The difference between $CO_2$ in the free troposphere and boundary layer can
be used to evaluate model transport.  One previous finding is that surface
assimilation estimates of northern minus southern hemisphere land flux are
correlated with the difference between $CO_2$ at 4 km and 1 km in the
assimilated model.  When the vertical difference of $CO_2$ is larger than aircraft
observations, models tend to predict too large northern hemisphere sinks
and too large southern hemisphere sources (Stephens et al., 2007).
Measurements $CO_2$ and 4 km and 1 km are performed only at a few sites
worldwide, primarily in the U.S.  Therefore, global measurements of the
difference between $CO_2$ in the free troposphere and boundary layer are of
great interest.  In this section we calculate the errors for LMT-U compared to
aircraft profiles and show this difference for GOSAT and CarbonTracker in
the U.S. and the southern hemisphere in two different months.
The error estimate for LMT-U is calculated using Eq. 13.  Note that a positive
correlation in the errors for LMT and U results in a smaller error for the
quantity (LMT – U).
$$\sigma_{(LMT-U)} = \sqrt{\sigma_{lmt}^2 + \sigma_u^2 - 2 * \sigma_{lmt}\sigma_u corr} \qquad (13)$$
Table 5a-c give the bias, standard deviation, and error with averaging for
LMT – U.  In Table 5a, the GOSAT bias and bias variability of (LMT – U)
improves over the prior for all cases.  The bias variability of 0.3, 0.9 and 0.8
ppm of (LMT – U) for HIPPO ocean, ESRL ocean, and ESRL land,
respectively, is comparable to the LMT bias variability of 0.3, 1.0, and 1.0 for
the same categories.  In Table 5b, the 15-observation average standard
deviation for GOSAT LMT-U is 0.6 (1.2) ppm for ocean (land), 0.2 ppm
higher for ocean and 0.7 ppm lower for land than LMT.  In Table 5c, the
correlated error is 0.5 (0.9) ppm for ocean (land), which is 0.2 ppm higher
for ocean and 0.8 ppm lower for land.  The land standard deviation for LMT-





U is 2.3 ppm before subtracting off the 2.1 ppm co-location error. The
difference between the land error for LMT and LMT-U is due to the estimated
size of the co-location error.
Figure 11 shows the seasonal cycle of LMT-U for 3 sites. The differences
between GOSAT and aircraft values at the CAR site in Colorado and LEF in
Wisconsin during the drawdown can be explained by co-location error. The
dotted lines shows CarbonTracker matched to GOSAT (red dotted) or aircraft
(pink dotted) locations/times. The difference between the red dotted and
pink dotted lines estimate the co-location error. If GOSAT were corrected by
this difference, the agreement with aircraft would be much better. The co-
location bias and standard deviation are estimated in Tables 6a and 6b, and
are large compared to the observed GOSAT errors. The error estimates for
GOSAT are corrected by the co-location error.
The predicted error for land in Table 6b is 2.7 ppm, whereas the actual error
is 2.3 ppm. If LMT and U had zero correlation, the predicted error (using Eq.
13) would be 3.6 ppm. This is another corroboration of the positive
correlation between the LMT and U errors.
Figure 12 shows LMT – U for February and July in the U.S. averaged over
2010-2014 for February and 2009-2013 for July. Aircraft values for LMT – U
are shown as squares. The aircraft patterns are captured by GOSAT, with
discrepancies in July for CAR, SGP, and SCA due to co-location error (e.g.
see CAR plot in Fig. 11). The CarbonTracker model captures the aircraft
patterns very well. The two maps show differences in the southwestern U.S.
in July, where there are no aircraft measurements. Figure 12c-d shows LMT
– U for February and October in the southern hemisphere. The only aircraft
site in this region is Rarotonga, where Fig. 11 shows good agreement for
CarbonTracker and GOSAT. The patterns in the southern hemisphere show
some differences between CarbonTracker and GOSAT. In February, GOSAT
shows a high gradient in the eastern Pacific and northern South America not
seen in CarbonTracker, and more negative gradient in central and southern
Africa. In October large gradients are seen by GOSAT in South America and
Africa with outflow into the Atlantic, with little seen in CarbonTracker.
LMT-U is predominantly positive in this southern hemisphere region in
October. Vertical transport from the northern hemisphere would
predominantly show up in the U partial column, whereas flux from land or
ocean would predominantly show up in the LMT partial column. An overall
positive value for LMT – U could either suggest that the overall flux is
positive in this month, or that transport from the northern hemisphere was
negative, though the blank space in the Amazon due to cloudy conditions,
where LMT-U is expected to be negative from plant uptake, creates





uncertainty both in this crude estimate and in the formal assimilated results
from GOSAT data.
**6.0 Discussion and conclusions**
GOSAT near-infrared observations provide information to retrieve two partial
columns, one from the surface to about 2.5 km ($LMT\_XCO_2$), and the second
above about 2.5 km ($U\_XCO_2$).  The LMT partial column is sensitive near the
surface, whereas the U partial column is sensitive to the free troposphere
and above; the two partial columns have distinct seasonal cycles, with peaks
and troughs earlier for the LMT partial column, and later for the U partial
column, as compared to $XCO_2$.  After bias correction, both partial columns
show agreement with aircraft, LMT shows agreement with remote surface
observations, and both show improvement over the GOSAT prior.  Single
observations for land have observation errors of 3.4, 1.3, and 1.7 ppm for
LMT, U, and $XCO_2$, respectively, and single observations for ocean have
observation errors of 1.5, 0.8, and 0.9 ppm for LMT, U, and $XCO_2$,
respectively. These errors are significantly reduced with averaging, though
some systematic errors, generally below 1 ppm, remain. The co-location
errors from mismatch of GOSAT versus validation data, as quantified by
CarbonTracker, makes the errors on LMT challenging to validate.  The value
of observing two partial columns can be seen in Fig. 8, where the GOSAT
LMT agrees with remote surface sites whereas neither the prior nor $XCO_2$
agree with the surface site, and Figs. 9-10, where surface versus
tropospheric $CO_2$ are distinguished for source and outflow of African biomass
burning emissions in August and October.  The observed LMT $CO_2$
enhancements are consistent with MOPITT multispectral CO and emission
ratios in Akagi et al. (2011).  The LMT-U difference, which can be used to
evaluate model transport error (e.g. Stephens et al., 2007), has also been
validated with monthly average error of 0.8 (1.4) ppm for ocean (land).  The
new LMT partial column allows the local boundary air to be distinguished
from the free troposphere, captured in the U partial column, disentangling
local versus remotely influenced signals.
**Funding Acknowledgements:**
This research was funded by NASA and performed under BAER Institute's
ARC-CREST cooperative agreement.
The AJAX team recognizes the support and partnership of H211 L. L. C. and
the NASA Postdoctoral Program; funding for instrumentation and aircraft
integration is gratefully acknowledged from Ames Research Center Director's
funds.



CarbonTracker CT2015 results provided by NOAA ESRL, Boulder, Colorado,
USA from the website at http://carbontracker.noaa.gov.
Part of this work was carried out at the Jet Propulsion Laboratory, California
Institute of Technology, under a contract with NASA.
TCCON at Lamont and Park Falls are funded by NASA grants NNX14AI60G,
NNX11AG01G, NAG5-12247, NNG05-GD07G, and NASA Orbiting Carbon
Observatory Program, with the DOE ARM program providing technical
support in Lamont and Jeff Ayers providing technical support in Park Falls.





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





## Appendix A.  Bias Correction

The ACOS-GOSAT $XCO_2$ product undergoes bias correction (Wunch et al., 2011) which significantly improves the errors (Kulawik, 2016).  We apply this same technique to correct the LMT product.  Following the LMT correction, U is corrected by subtracting the LMT partial column from ACOS-GOSAT corrected $XCO_2$, thus maintaining consistency between the [LMT,U] partial columns and the total $XCO_2$ column after bias-correction.

To determine the LMT bias correction, GOSAT and aircraft data are matched using dynamic coincidence criteria (Wunch, 2011), and the difference between GOSAT LMT and aircraft LMT is calculated for all pairs versus each potential parameter.  In order to identify the critical bias-predicting parameters, for those cases for which this difference has a clear slope, a bias correction is applied iteratively, where the strongest parameter dependence is corrected before the next parameters are tested.  At the end all parameters are fit simultaneously.  Filters are applied to flag the data as bad when the bias is significant even after correction.  The parameters considered for bias correction are:  delta_grad_co2, albedo_1, albedo_2, albedo_3, albedo_slope_1, albedo_slope_2, albedo_slope_3, aod_dust, aod_ice, aod_total, b1offset, ice_height, surfacePressure_xa, surfacePressureDiff, co2_ratio, dp_cld, h2o_ratio, s32, xco2_error, LMT_dofs (degrees of freedom for LMT partial column), U_dofs (degrees of freedom for U partial column), xco2_dofs, asza, lza, and delta_grad_co2_prime.  These parameters are described in the ACOS-GOSAT v3.5 user's guide with the exception of delta_grad_co2_prime which is defined as delta_grad_co2 with the value set to 50 when it is greater than 50 for land, and the value set to -10 when it is greater than -10 for ocean. Two figures of merit were considered for the cutoffs and bias fits, (1) bias variability by location and season and (2) the single-observation standard deviation.  The former is the standard deviation of the biases calculated in 4 seasons and for each location/campaign.  For both of these figures of merit, smaller is better.

By far the strongest bias is related to delta_grad_CO2.  This parameter is the difference between the retrieved $CO_2$ and a priori dry-air molefraction between the surface and vertical level 13 (approximately 630 hPa for soundings near sea level), and represents the slope of the retrieved $CO_2$ profile in the troposphere  The resulting coefficient for this term is 0.396 for ocean and 0.310 for land soundings.  This indicates that, for ocean, approximately 40% of the $CO_2$ attributed to the surface should be moved from LMT to U, indicating that possibly (a) the troposphere is constrained too much relative to the surface, (b) an issue with spectroscopy, or (c) some other retrieval artefact.  As the bias correction for simulated OCO-2 data is very similar factor (Kulawik, unpublished result), with simulations run with



no spectroscopic error, it is likely a consequence of the constraint, or some
other aspect of the retrieval.
The filtering cutoffs and bias terms are shown in Table A1. The errors
calculated by the bootstrap method (Rubin, 1981). The effects of the cutoffs
and bias corrections from Table A1 on biases and standard deviations is
shown in Table A2.
The overall land bias is not zero because the land bias constant correction
undergoes a final step to harmonize land and ocean observations by
matching GOSAT values for pairs of close land and ocean observations. The
results for different matching criteria are: 1 degree and 1 hour (25 matches,
bias -0.54 ppm in LMT and -0.96 ppm in $XCO_2$), 2 degrees and 24 hours
(295 matches, 0.17 ppm in LMT and -0.61 ppm in $XCO_2$), 4 degrees and 48
hours (4095 matches, 1.17 ppm in LMT and -0.09 ppm in $XCO_2$), and using
dynamic coincidence criteria (422,542 matches, 0.29 ppm in LMT, -0.42 in
$XCO_2$). Using the assumption that there is no bias in $XCO_2$, the 4 degree, 48
hours result is used, and 1.17 ppm is added to the LMT constant bias for
land. This constant bias is subtracted from LMT, then the LMT partial
column is subtracted from $XCO_2$ to generate the corrected U partial column.
As seen from Tables A3a and A3b, all bias corrections are superior to the
uncorrected dataset, and all correction tests perform similarly in the bias
standard deviation and mean standard deviation, but with variability in the
overall bias, depending on the development set that is used. The overall
bias has some uncertainty on the order of 0.5 ppm.
Another potential error source that is quantified is the effect of different
profile extension schemes above aircraft observations. The ESRL aircraft
measurements go up to 5-8 km above ground, and the HIPPO observations
go up to 9-13 km above ground. 4 different profile extension methods are
tried above the aircraft: using (1) the GOSAT a priori profile, (2) extending
the top aircraft measurement to the tropopause with the GOSAT prior above
this, (3) the CT2015 model, and (4) extending the top aircraft measurement
to the tropopause with the CT2015 model above this. Table A4 shows the
land and ocean characteristics with each of the profile extension type. The
main effect is on the overall bias (up to 0.3 ppm) in the comparisons. One
issue is likely in the top 4 levels, from which a difference between a priori
and the true profile would propagate as a bias.
There were two ways that the developed bias correction was insulated from
the validation: (1) the bias correction uses dynamic coincidence criteria
(Wunch, 2011), whereas the comparisons to validation data use geometric
coincidence criteria (±5 degrees latitude and longitude, and ±1 week). The



overlap between these two sets is about 50%. (2) remote ocean surface
sites were not used to develop the bias correction. These locations are
expected to have good mixing between the surface and 2.5 km, but since we
do not have profiles at these locations, these observations are not used for
direct validation. These comparisons between GOSAT and remote surface
sites show excellent improvement over the GOSAT prior. (3) No data over
the southern hemisphere biomass burning is used in the bias correction, and
GOSAT compares very well to MOPITT in this region.
The mean and standard deviation of the bias correction is -11.4±7.6,
2.7±2.7 ppm for LMT and U land, respectively and -1.0±3.1 ppm, -1.7±0.9
ppm for LMT and U ocean, respectively. The mean and standard deviations
of the bias correction for $XCO_2$ are: -0.6±1.0 ppm for land and -0.6±0.6 for
ocean. The bias corrections are larger for the partial columns than for $XCO_2$;
the size and variability of the bias correction is an indication of its
importance.



**Appendix B.  Detailed comparisons by site and campaign**

In addition to the averaged results provided previously, Table B1 below
breaks down the validation results for each individual station. This table
could be useful for diagnosing outliers in the comparisons, looking at
correlations of site-to-site biases or standard deviations in LMT and U.






Table 1
Predicted errors and degrees of freedom for LMT and U.  As seen in Table 2,
the predicted errors are much larger than the actual errors.

|  | land | ocean |
|---|---|---|
| LMT error (ppm) | 4.3 ppm | 4.4 ppm |
| U error (ppm) | 1.7 ppm | 1.7 ppm |
| U,LMT pred. error correlation | -0.72 | -0.78 |
| LMT DOFs | 0.86 | 0.86 |
| U DOFs | 0.84 | 0.83 |

Table 2. Biases versus aircraft. The top entries are the co-location biases.
The second row is the mean and variability of the true state.  The 3rd row are
the prior bias, and the fourth set are the GOSAT bias (prior and GOSAT bias
have co-location bias subtracted). All data is first averaged by location or
campaign.  The ± represents the variability of the bias by location or
campaign, a key metric in the data quality.

|  | Type | Ocean surface (ppm) | HIPPO Ocean (ppm) | ESRL Ocean (ppm) | ESRL Land (ppm) | AJAX Land (ppm) |
|---|---|---|---|---|---|---|
| CT_CT bias (estimate of co-location bias) | LMT | -0.3±0.3 -0.3±0.8 | -0.3±0.2 | -0.3±0.4 | -0.6±0.7 | -0.6 |
|  | U |  | 0.1±0.1 | -0.1±0.1 | 0.0±0.2 | 0.0 |
|  | $XCO_2$ |  | 0.0±0.1 | -0.1±0.1 | -0.1±0.3 | -0.1 |
| true mean by site/campaign | LMT | 391.3±1.6 | 392.2±1.6 | 391.7±1.1 | 392.2±3.1 | 393.6 |
|  | U |  | 391.1±1.2 | 391.3±1.6 | 391.2±0.6 | 392.2 |
|  | $XCO_2$ |  | 391.4±1.3 | 391.4±1.5 | 391.5±1.1 | 392.4 |
| prior bias | LMT | -0.8±1.5 | 0.1±2.4 | -1.5±4.5 | -0.4±1.2 | -1.4 |
|  | U |  | 1.2±0.1 | -1.2±1.6 | 0.6±0.6 | 0.4 |
|  | $XCO_2$ |  | 0.9±1.4 | 0.4±2.3 | -0.2±0.6 | -0.1 |
| GOSAT bias | LMT | 1.1±1.1 | 0.1±0.3 | 0.3±0.7 | -0.2±1.0 | 0.4 |
|  | U |  | 0.1±0.3 | 0.7±0.1 | 0.3±0.9 | 1.0 |
|  | $XCO_2$ |  | 0.1±0.2 | 0.6±0.4 | 0.1±0.9 | 0.7 |

Table 3. Standard deviations versus aircraft showing the errors resulting
from imperfect co-location, GOSAT predicted error, and GOSAT error (for
single observations).  The next sets are the true variability, 15-observation
averages of prior error and GOSAT error.  The GOSAT and prior standard
deviation have the co-location error subtracted.

|  | Type | Ocean surface (ppm) | HIPPO Ocean (ppm) | ESRL Ocean (ppm) | ESRL Land (ppm) | AJAX Land (ppm) |
|---|---|---|---|---|---|---|
| colocation standard deviation | LMT | 0.5±0.2 0.9±0.4 | 0.3±0.1 | 0.3±0.1 | 2.1±0.7 | 1.1 |
|  | U |  | 0.1±0.1 | 0.2±0.0 | 0.5±0.3 | 0.1 |
|  | $XCO_2$ |  | 0.1±0.2 | 0.2±0.1 | 0.8±0.3 | 0.2 |

| | Type | | | | | |
|---|---|---|---|---|---|---|
| **predicted error (n=1)** | LMT | 4.3±0.2 | 4.3±0.3 | 4.3±0.1 | 4.6±0.3 | 4.1 |
| | U | | 1.7±0.1 | 1.7±0.0 | 1.8±0.0 | 1.7 |
| | $XCO_2$ | | 0.6±0.1 | 0.7±0.1 | 0.9±0.1 | 0.8 |
| **GOSAT standard deviation (n=1)** | LMT | 1.7±0.4 | 1.7±0.3 | 1.5±0.1 | 3.4±0.7 | 2.9 |
| | U | | 0.8±0.1 | 0.8±0.0 | 1.3±0.3 | 1.1 |
| | $XCO_2$ | | 0.9±0.1 | 0.8±0.1 | 1.7±0.4 | 0.9 |
| **true variability** | LMT | 1.3±0.8 | 0.6± 0.2 | 0.9±0.6 | 5.5±2.0 | 2.8 |
| | U | | 0.4±0.3 | 0.8±0.8 | 2.0±0.2 | 2.0 |
| | $XCO_2$ | | 0.3±0.3 | 0.8±0.8 | 2.5±0.6 | 2.4 |
| **prior standard deviation (n=15)** | LMT | 2.2±0.9 | 0.5± 0.3 | 0.7±0.2 | 2.1±1.0 | - |
| | U | | 0.3±0.1 | 0.5±0.0 | 0.9±0.2 | - |
| | $XCO_2$ | | 0.3±0.1 | 0.5±0.1 | 1.1±0.6 | - |
| **GOSAT standard deviation (n=15)** | LMT | 0.4±0.3 | 0.5± 0.1 | 0.4±0.1 | 1.9±1.1 | - |
| | U | | 0.4±0.1 | 0.6±0.1 | 0.7±0.4 | - |
| | $XCO_2$ | | 0.3±0.1 | 0.4±0.1 | 0.8±0.5 | - |

Table 4. Estimated co-location, correlated, and random errors. The co-location errors are the same as in Table 3.

| | Type | Ocean surface (ppm) | HIPPO Ocean (ppm) | ESRL Ocean (ppm) | ESRL Land (ppm) |
|---|---|---|---|---|---|
| **colocation standard deviation** | LMT | 1.0±0.4 | 0.3±0.1 | 0.3±0.1 | 2.1±0.7 |
| | U | | 0.1±0.1 | 0.2±0.0 | 0.5±0.3 |
| | $XCO_2$ | | 0.1±0.1 | 0.1±0.1 | 0.8±0.3 |
| **correlated error ($a_o$)** | LMT | 0.4±0.3 | 0.3±0.2 | 0.3±0.2 | 1.7±1.3 |
| | U | | 0.3±0.2 | 0.5±0.1 | 0.6±0.4 |
| | $XCO_2$ | | 0.2±0.2 | 0.4±0.1 | 1.1±0.6 |
| **random error (b)** | LMT | 1.6±0.4 | 1.6±0.3 | 1.4±0.2 | 3.0±0.6 |
| | U | | 0.8±0.1 | 0.6±0.1 | 1.2±0.1 |
| | $XCO_2$ | | 0.9±0.1 | 0.4±0.1 | 0.8±0.3 |

Table 5. Enhancements in CO and $CO_2$ for May and August, 2010. The target box is 11 to 18S, 60 to 56W for May, and 55-60S, 13-17W for August. The CO background box is 11 to 18S, 40 to 44W for May and 157.8-161.8W, 19-23S for August. Rarotonga aircraft measurements are used for $CO_2$ background.

| | | CO | | | | | GOSAT LMT $CO_2$ | |
|---|---|---|---|---|---|---|---|---|
| | | backgrnd (ppb) | Target all (ppb) | Target DOFs > 0.15 (ppb) | Target (DOFs > 0.25) (ppb) | Target (DOFs > 0.30) (ppb) | backgrnd from RTA (ppm) | Target (DOFs = 0.8) (ppm) |
| **May, 2010** | Mean | 68±9 | 122±49 | 123±54 | 146±77 | 182±96 | 386.4 | 389.6±2.3 |
| | N | 1502 | 2023 | 1556 | 500 | 215 | | 13 |
| | DOFs | | 0.21 | 0.24 | 0.32 | 0.39 | | 0.85 |
| | ΔCO or Δ$CO_2$ | - | 54 | 55 | 88 | 114 | - | 3.2 |





| August, 2010 | Mean | 91±22 | 305±171 | 311±180 | 336±200 | 372±221 | 387.4 | 392.2±6.7 |
|---|---|---|---|---|---|---|---|---|
| | N | 2989 | 3881 | 3227 | 1887 | 1231 | | 5 |
| | ΔCO or ΔCO₂ | - | 213.7 | 219.3 | 244.8 | 281.1 | - | 4.8 |


Table 6a. Bias terms for LMT – U. Compare to Table 2.

| | HIPPO Ocean (ppm) | ESRL Ocean (ppm) | ESRL Land (ppm) |
|---|---|---|---|
| colocation bias | -0.4±0.2 | -0.2±0.3 | -0.6±0.5 |
| true mean | 1.1±0.8 | 0.4±0.5 | 1.0±2.7 |
| prior bias | -1.0±1.3 | -2.8±2.9 | -1.0±1.2 |
| GOSAT bias | 0.0±0.4 | -0.5±0.9 | -0.5±0.8 |


Table 6b. Standard deviations for LMT – U. Compare to Table 3. The
predicted errors in the table use the errors given at the end of Section 5.1.5.

| | HIPPO Ocean (ppm) | ESRL Ocean (ppm) | ESRL Land (ppm) |
|---|---|---|---|
| colocation stdev | 0.3±0.1 | 0.3±0.1 | 2.1±0.7 |
| predicted error (n=1) | 1.2±0.0 | 1.2±0.0 | 2.7±0.0 |
| GOSAT stdev (n=1) | 1.5±0.4 | 1.3±0.1 | 2.3±0.5 |
| true variability | 0.5± 0.2 | 0.8±0.1 | 4.8±1.5 |
| prior stdev (n=15) | 0.5± 0.2 | 0.8±0.1 | 1.4±0.8 |
| GOSAT stdev (n=15) | 0.5± 0.2 | 0.7±0.1 | 1.2±0.8 |


Table 6c. Error fits for LMT – U. Compare to Table 4.

| | HIPPO Ocean (ppm) | ESRL Ocean (ppm) | ESRL Land (ppm) |
|---|---|---|---|
| colocation stdev | 0.3±0.1 | 0.3±0.1 | 2.1±0.7 |
| correlated (a) | 0.4±0.2 | 0.6±0.0 | 0.9±0.9 |
| random (b) | 1.4±0.4 | 1.1±0.1 | 2.1±0.7 |


Table A.1 Filtering and Bias corrections

| parameter | ocean filtering | ocean bias correction | land filtering | land bias correction |
|---|---|---|---|---|
| albedo_2 | 0.0215< val <0.024 | -1272.02 ± 50 | - | - |
| albedo_slope_2 | val < 8e-6 | - | - | - |
| aod_dust | val < 0.01 | - | - | -36.03 ± 1 |
| aod_total | val < 0.25 | - | - | - |
| h2o_ratio | 0.96 < val < 1.02 | - | - | - |
| co2_grad_delta | -40 < val < 17 | 0.396330 ± 0.004 | - | 0.310 ± 0.003 |
| constant | - | 52.674 ± 6 | - | 0.01259 ± 0.4 |
| b1_offset | - | -1.25204 ± 0.05 | - | - |
| surfacepressure_xa | - | -0.0381105 ± 0.006 | - | - |
| s32 | - | 17.0742 ± 3 | - | - |
| surfacepressurediff | - | 0.869280 ± 0.05 | - | - |
| albedo_1 | - | 144.458 ± 9 | - | - |
| co2_grad_delta_prime | - | -0.171350 ± 0.01 | - | -0.027 ± 0.005 |
| dofs_LMT | - | - | val > 0.68 | - |




| | | | | | |
|---|---|---|---|---|---|
| xco2_error | - | - | val < 1.4 | 6.02 ± 0.3 | |
| albedo_slope_3 | - | - | -1.5e-4 < val <2.0e-4 | - | |
| xco2_dofs | - | - | val > 1.3 | - | |
| ice_height | - | - | val > -0.1 | - | |
| surfacePressureDiff | - | - | -4 < val < 2 | - | |
| albedo_3 | - | - | - | -11.66 ± 0.7 | |
| dp_cld | - | - | - | 0.219 ± 0.01 | |

* parameters also used in ACOS-GOSAT XCO₂ bias correction
Table A2a.  Effects of bias corrections and quality flags on land comparisons
(ESRL aircraft land observations)

| | n | lmt bias (ppm) | lmt bias var. (ppm) | lmt stdev (ppm) | u bias var. (ppm) | u stdev (ppm) |
|---|---|---|---|---|---|---|
| original (XCO₂ flags) | 15143 | 13.54 | 2.79 | 7.70 | 1.61 | 3.05 |
| all quality flags (see appendix A) | 12714 | 13.37 | 2.30 | 7.55 | 1.27 | 2.98 |
| bias correction (see appendix A) | 12714 | -1.18 | 1.43 | 3.47 | 0.79 | 1.36 |
| fit U separately | 11978 | - | - | - | 0.70 | 1.43 |

Table A2b.  Effects of bias corrections and quality flags on ocean
comparisons (HIPPO and ESLR ocean dataset stations/campaigns:  tgc, rta,
aoa, 2S, 2N, 3S, 3N, 4S, 4N, 5S, 5N)

| | n | lmt bias (ppm) | lmt bias var. (ppm) | lmt stdev (ppm) | u bias var. (ppm) | u stdev (ppm) |
|---|---|---|---|---|---|---|
| original (XCO₂ flags) | 9836 | 1.73 | 3.46 | 3.77 | 0.78 | 0.85 |
| with cutoffs (see Appendix A) | 6143 | 1.47 | 1.92 | 3.18 | 0.63 | 0.69 |
| bias correction (see Appendix A) | 6143 | 0.04 | 0.68 | 1.60 | 0.38 | 0.79 |
| fit U separately | 6143 | - | - | - | 0.35 | 0.60 |

The fit parameters are tested for robustness by using a subset of the dataset
to determine the fit and then testing the fit on the independent subset.  For
the ocean data, HIPPO campaigns 2N, 3S, 4, and 5 are used to develop bias
correction, and HIPPO 2S and 3N are used for testing.  For land data,
stations bne, car, cma, dnd, esp, etl, hil, hip, are used for development, and
stations lef, nha, pfa, sca, sgp, tgc, thd, wbi are used for testing.
Table A3a:  Ocean bias correction robustness test.  Comparisons to aircraft
data are tested using (a) no bias correction, (b) bias correction using the
test dataset, (c) an independent dataset, and (d) the entire dataset

| Bias correction testing | Mean bias | Bias std | mean std |
|---|---|---|---|
| no correction | 0.69 | 0.69 | 2.97 |
| subset tested on itself | -0.04 | 0.33 | 1.47 |
| independent subset | -0.26 | 0.46 | 1.58 |
| all data used | -0.14 | 0.49 | 1.54 |






Table A3b: Land bias correction robustness test. Same as Table A3a but for
land.

| Bias correction testing | Mean bias | Bias std | mean std |
|---|---|---|---|
| no correction | 13.00 | 2.47 | 7.54 |
| subset tested on itself | 0.16 | 1.55 | 3.68 |
| independent subset | 1.05 | 1.24 | 3.67 |
| all data used | 0.50 | 1.51 | 3.65 |


Table A4: Effect of profile extension. GOSAT corrected as described in
Table A1 and compared to aircraft data with profile extended 4 different
ways: (a) using the GOSAT prior, (b) extending the aircraft to the
tropopause pressure, with the GOSAT prior above this, (c) using the CT2015
model, and (d) extending the aircraft to the tropopause pressure, with the
CT2015 above this

| Profile extension | LMT bias | LMT Bias std | LMT std | U bias | U Bias std | U std |
|---|---|---|---|---|---|---|
| (a) prior | -0.90 | 1.37 | 3.46 | -0.38 | 0.70 | 1.25 |
| (b) extend+prior | -0.99 | 1.44 | 3.47 | -0.20 | 0.79 | 1.35 |
| (c) CT2015 | -1.20 | 1.39 | 3.47 | -0.02 | 0.66 | 1.26 |
| (d) extend+CT2015 | -1.18 | 1.43 | 3.47 | -0.05 | 0.79 | 1.36 |


Table B1. Actual and predictions of errors by station/campaign. Columns $a$
and $b$ are the fits to actual errors when averaging using Eq. 11a; $\varepsilon_{coloc}$ is the
standard deviation of CT2015 at the observation vs. validation data. The
predicted error is the error calculated from the ACOS full error covariance for
each type of quantity. Bias ct_ct is difference between CT2015 at the
satellite and validation locations. True mean is the average validation value
at each location; prior and GOSAT biases are difference between the prior
and retrieved GOSAT minus validation data. The last three columns are the
standard deviation of the true, prior minus true, and GOSAT retrieved minus
true for 15 observation averages.

| location | latitude, longitude | $\varepsilon_{coloc}$ ct_ct stdev | a corr. error | b rand. error | GOSAT prior (n=1) | GOSAT stdev (n=1) | pred. Error (n=1) | bias ct_ct | true mean | prior bias | GOSAT bias | true stdev (n=1) | prior stdev (n=15) | GOSAT stdev (n=15) |
|---|---|---|---|---|---|---|---|---|---|---|---|---|---|---|
| | | (ppm) | (ppm) | (ppm) | (ppm) | (ppm) | (ppm) | (ppm) | (ppm) | (ppm) | (ppm) | (ppm) | (ppm) | (ppm) |
| | | | | | a) LMT vs. surface ocean flasks at remote sites | | | | | | | | | |
| bmw | 32N,65W | 0.4 | 0.9 | 2.5 | 4.6 | 2.6 | 4.2 | -0.8 | 391.8 | -3.0 | -1.4 | 3.3 | 2.8 | 1.1 |
| mid | 28N,177W | 0.8 | 1.5 | 1.8 | 4.2 | 2.3 | 4.3 | 0.1 | 389.9 | -2.4 | -0.2 | 2.2 | 4.5 | 1.5 |
| mnm | 24N,154E | 0.3 | 0.8 | 1.6 | 3.8 | 1.8 | 4.2 | 0.2 | 393.2 | -3.8 | -0.6 | 1.6 | 2.8 | 0.9 |
| mlo | 20N,156W | 0.8 | 1.0 | 1.4 | 2.6 | 1.7 | 4.5 | -0.6 | 390.9 | -2.1 | -0.3 | 1.7 | 2.2 | 1.0 |
| kum | 20N,155W | 0.7 | 1.5 | 1.2 | 2.6 | 1.9 | 4.5 | -0.6 | 390.0 | -1.1 | 0.7 | 1.7 | 2.5 | 1.5 |
| gmi | 13N,145E | 0.5 | 0.7 | 1.6 | 2.8 | 1.8 | 4.4 | 0.0 | 394.8 | -2.9 | 0.9 | 1.2 | 1.9 | 0.8 |
| chr | 2N,157W | 0.3 | 0.8 | 1.4 | 1.6 | 1.6 | 4.4 | -0.2 | 392.1 | -0.8 | 0.4 | 1.1 | 1.9 | 0.9 |
| sey | 5S,56E | 0.4 | 1.3 | 1.8 | 2.2 | 2.2 | 4.0 | -0.3 | 391.4 | -0.2 | 0.7 | 1.3 | 0.8 | 1.3 |
| asc | 8S,14W | 0.3 | 1.0 | 1.5 | 1.7 | 1.8 | 4.4 | -0.4 | 390.4 | 0.1 | 1.5 | 0.7 | 2.5 | 1.1 |
| smo | 14S,171W | 0.5 | 0.5 | 1.7 | 2.2 | 1.8 | 4.2 | -0.5 | 390.6 | 0.0 | 0.6 | 0.5 | 2.2 | 0.7 |



| | | | | | | | | | | | | | |
|---|---|---|---|---|---|---|---|---|---|---|---|---|---|
| eic | 27S,109W | 0.5 | 0.8 | 1.2 | 2.1 | 1.4 | 4.2 | -0.4 | 389.7 | 0.7 | 2.7 | 0.7 | 1.9 | 0.8 |
| | average | 0.5 | 1.0 | 1.5 | 2.6 | 1.8 | 4.3 | -0.3 | 391.3 | -1.2 | 0.7 | 1.3 | 2.3 | 1.1 |
| | | ±0.2 | ±0.3 | ±0.2 | ±0.8 | ±0.3 | ±0.2 | ±0.3 | ±1.6 | ±1.5 | ±1.0 | ±0.5 | ±0.9 | ±0.3 |
| **b) LMT vs. ESRL aircraft** | | | | | | | | | | | | | |
| pfa | 66N,147W | 1.6 | 5.0 | 1.6 | 2.1 | 5.3 | 5.1 | 0.1 | 388.0 | 1.9 | 0.3 | 8.2 | 1.5 | 5.0 |
| etl | 54N,105W | 2.2 | 2.6 | 2.6 | 3.6 | 3.7 | 4.8 | -0.3 | 388.7 | -1.0 | -0.6 | 6.9 | 3.5 | 2.7 |
| esp | 49N,126W | 3.2 | 3.2 | 4.6 | 4.1 | 5.6 | 5.0 | 0.0 | 386.1 | -2.4 | -0.2 | 4.4 | 3.6 | 3.4 |
| dnd | 47N,99W | 1.4 | 2.9 | 2.4 | 3.8 | 3.8 | 4.5 | -0.1 | 390.0 | -0.6 | -0.7 | 7.8 | 5.0 | 3.0 |
| lef | 46N,90W | 2.6 | 3.5 | 2.2 | 3.7 | 4.1 | 4.7 | -0.3 | 392.1 | -0.9 | -1.4 | 6.8 | 4.5 | 3.5 |
| nha | 43N,71W | 1.6 | 1.9 | 3.5 | 2.8 | 4.0 | 4.8 | -0.3 | 393.3 | -0.1 | 0.1 | 7.7 | 2.6 | 2.1 |
| wbi | 42N,91W | 2.8 | 1.9 | 2.9 | 2.6 | 3.5 | 4.5 | -1.5 | 393.3 | -0.7 | -0.9 | 5.1 | 2.3 | 2.1 |
| thd | 41N,124W | 2.2 | 2.7 | 3.5 | 2.5 | 4.4 | 4.6 | 0.3 | 389.5 | -1.5 | 0.9 | 3.9 | 2.5 | 2.8 |
| bne | 41N,97W | 2.1 | 2.4 | 3.0 | 3.3 | 3.9 | 4.4 | -1.3 | 393.2 | -2.5 | -2.2 | 5.0 | 3.1 | 2.5 |
| car | 41N,104W | 2.7 | 2.7 | 3.3 | 3.6 | 4.2 | 4.2 | -2.2 | 393.0 | -2.7 | -2.6 | 3.5 | 3.3 | 2.8 |
| hil | 40N,88W | 2.2 | 2.2 | 3.0 | 3.4 | 3.8 | 4.5 | -0.9 | 396.3 | -2.0 | -2.4 | 5.7 | 3.1 | 2.4 |
| cma | 39N,74W | 1.8 | 1.8 | 3.7 | 3.0 | 4.1 | 4.8 | -0.6 | 394.9 | -0.7 | -0.5 | 6.1 | 2.3 | 2.0 |
| sgp | 37N,98W | 1.8 | 2.7 | 2.9 | 4.1 | 3.9 | 4.3 | -0.5 | 394.3 | -1.5 | -0.7 | 4.2 | 3.7 | 2.8 |
| sca | 33N,79W | 1.0 | 1.1 | 3.2 | 2.3 | 3.3 | 4.8 | -0.5 | 395.6 | 0.3 | -1.3 | 2.9 | 1.8 | 1.3 |
| aoa | 29N,148E | 0.4 | 0.7 | 1.2 | 1.1 | 1.4 | 4.2 | -0.5 | 392.4 | -5.0 | -0.8 | 1.5 | 0.9 | 0.8 |
| tgc | 28N,97W | 1.1 | 1.5 | 2.5 | 2.7 | 2.9 | 4.2 | -0.1 | 394.9 | -0.2 | 0.0 | 2.7 | 2.3 | 1.7 |
| rta | 21S,160W | 0.4 | 0.2 | 1.6 | 1.0 | 1.6 | 4.3 | 0.0 | 390.9 | 1.3 | 0.7 | 0.7 | 0.7 | 0.5 |
| | average land | 2.0 | 2.5 | 3.0 | 3.2 | 4.0 | 4.6 | -0.5 | 392.2 | -1.0 | -0.8 | 5.4 | 3.0 | 2.7 |
| | | ±0.6 | ±1.2 | ±0.7 | ±0.6 | ±0.7 | ±0.3 | ±0.7 | ±3.1 | ±1.2 | ±1.0 | ±1.8 | ±1.0 | ±0.9 |
| | ave. land, corrected | | 1.5 | | 2.4 | 3.4 | | | | -0.5 | -0.3 | | 2.2 | 1.7 |
| | | | ±1.2 | | ±0.6 | ±0.7 | | | | ±01.2 | ±1.0 | | ±1.0 | ±0.9 |
| aoa, rta | average ocean | 0.4 | 0.4 | 1.4 | 1.1 | 1.5 | 4.3 | -0.3 | 391.7 | -1.9 | -0.1 | 1.1 | 0.8 | 0.7 |
| | | ±0.0 | ±0.5 | ±0.3 | ±0.1 | ±0.1 | ±0.1 | ±0.4 | ±1.1 | ±04.5 | ±1.1 | ±0.6 | ±0.1 | ±0.2 |
| **c) U vs. ESRL aircraft** | | | | | | | | | | | | | |
| pfa | 66N,147W | 0.5 | 1.3 | 1.1 | 1.3 | 1.7 | 1.8 | 0.1 | 392.0 | 1.8 | 1.5 | 2.4 | 1.0 | 1.3 |
| etl | 54N,105W | 0.4 | 1.0 | 1.2 | 1.6 | 1.6 | 1.8 | 0.1 | 390.8 | 1.3 | 0.9 | 1.8 | 1.7 | 1.1 |
| esp | 49N,126W | 1.2 | 2.0 | 1.1 | 1.6 | 2.3 | 1.8 | 0.4 | 389.9 | 1.7 | 2.2 | 2.1 | 1.9 | 2.0 |
| dnd | 47N,99W | 0.6 | 0.7 | 1.3 | 1.6 | 1.5 | 1.8 | 0.2 | 390.5 | 0.8 | 0.4 | 2.2 | 1.8 | 0.8 |
| lef | 46N,90W | 0.5 | 0.5 | 1.2 | 1.4 | 1.3 | 1.8 | 0.0 | 391.3 | 0.4 | 0.1 | 2.1 | 1.5 | 0.6 |
| nha | 43N,71W | 0.5 | 0.8 | 1.2 | 1.3 | 1.5 | 1.8 | 0.0 | 391.5 | 0.4 | 0.3 | 2.5 | 0.9 | 0.8 |
| wbi | 42N,91W | 0.4 | 0.6 | 1.1 | 0.8 | 1.2 | 1.8 | -0.2 | 391.2 | 0.3 | -0.2 | 2.1 | 0.6 | 0.7 |
| thd | 41N,124W | 0.9 | 1.0 | 1.2 | 1.2 | 1.6 | 1.8 | 0.4 | 390.5 | 1.4 | 1.8 | 1.9 | 0.8 | 1.1 |
| bne | 41N,97W | 0.4 | 0.6 | 1.2 | 1.1 | 1.3 | 1.7 | -0.1 | 391.2 | 0.4 | -0.4 | 2.0 | 1.1 | 0.7 |
| car | 41N,104W | 0.6 | 0.8 | 1.3 | 1.0 | 1.5 | 1.7 | -0.2 | 391.1 | 0.4 | 0.0 | 2.0 | 1.0 | 0.8 |
| hil | 40N,88W | 0.5 | 0.7 | 1.1 | 1.1 | 1.3 | 1.8 | -0.1 | 392.1 | -0.4 | -0.9 | 2.0 | 0.9 | 0.8 |
| cma | 39N,74W | 0.3 | 0.5 | 1.4 | 0.9 | 1.5 | 1.8 | -0.1 | 391.5 | 0.3 | 0.1 | 2.1 | 0.5 | 0.6 |
| sgp | 37N,98W | 0.4 | 0.5 | 1.1 | 0.8 | 1.2 | 1.7 | 0.0 | 391.4 | 0.0 | -0.4 | 1.7 | 0.7 | 0.6 |
| sca | 33N,79W | 0.2 | 0.4 | 1.1 | 0.5 | 1.2 | 1.8 | -0.1 | 391.8 | 0.2 | -0.8 | 1.6 | 0.3 | 0.5 |
| aoa | 29N,148E | 0.2 | 0.6 | 0.4 | 0.5 | 0.8 | 1.7 | -0.1 | 392.4 | 0.0 | 0.6 | 1.4 | 0.5 | 0.6 |
| tgc | 28N,97W | 0.2 | 0.3 | 1.0 | 0.5 | 1.1 | 1.7 | 0.0 | 391.6 | 0.4 | -0.3 | 1.9 | 0.5 | 0.4 |
| rta | 21S,160W | 0.2 | 0.5 | 0.7 | 0.5 | 0.8 | 1.7 | 0.0 | 390.1 | 2.3 | 0.8 | 0.2 | 0.5 | 0.5 |
| | average land | 0.5 | 0.7 | 1.2 | 1.1 | 1.4 | 1.8 | 0.0 | 391.2 | 0.6 | 0.3 | 2.0 | 1.0 | 0.8 |
| | | ±0.3 | ±0.4 | ±0.1 | ±0.4 | ±0.3 | ±0.0 | ±0.2 | ±0.6 | ±0.6 | ±0.9 | ±0.2 | ±0.2 | ±0.4 |
| | ave. land, corrected | | 0.6 | | 0.5 | 1.3 | | | | 0.6 | 0.3 | | 0.9 | 0.5 |
| | | | ±0.4 | | ±0.0 | ±0.3 | | | | ±0.6 | ±0.9 | | ±0.2 | ±0.4 |
| aoa, rta | average ocean | 0.2 | 0.6 | 0.6 | 1.0 | 0.8 | 1.7 | -0.1 | 391.3 | -1.2 | 0.7 | 0.8 | 0.5 | 0.6 |
| | | ±0.0 | ±0.1 | ±0.2 | ±0.4 | ±0.0 | ±0.0 | ±0.1 | ±1.6 | ±1.6 | ±0.1 | ±0.8 | ±0.0 | ±0.1 |
| **d) XCO₂ vs. ESRL aircraft** | | | | | | | | | | | | | |
| pfa | 66N,147W | 0.7 | 2.1 | 1.2 | 1.4 | 2.4 | 1.3 | 0.2 | 391.1 | 1.8 | 1.2 | 3.8 | 1.0 | 2.1 |
| etl | 54N,105W | 0.7 | 1.3 | 1.5 | 1.9 | 2.0 | 0.9 | 0.0 | 390.3 | 0.7 | 0.6 | 2.8 | 2.1 | 1.4 |
| esp | 49N,126W | 1.5 | 2.2 | 2.0 | 1.9 | 2.9 | 0.9 | 0.4 | 389.0 | 0.8 | 1.6 | 2.4 | 2.1 | 2.2 |
| dnd | 47N,99W | 0.7 | 1.0 | 1.6 | 2.0 | 1.9 | 0.9 | 0.1 | 390.4 | 0.5 | 0.2 | 3.1 | 2.4 | 1.1 |
| lef | 46N,90W | 0.9 | 1.1 | 1.5 | 1.7 | 1.8 | 1.0 | 0.0 | 391.4 | 0.1 | -0.3 | 2.7 | 2.0 | 1.2 |
| nha | 43N,71W | 0.7 | 0.9 | 1.7 | 1.5 | 1.9 | 1.0 | -0.1 | 391.9 | 0.3 | 0.3 | 3.5 | 1.2 | 1.0 |
| wbi | 42N,91W | 0.9 | 0.8 | 1.4 | 1.0 | 1.6 | 0.8 | -0.5 | 391.7 | 0.0 | -0.3 | 2.3 | 0.8 | 0.9 |
| thd | 41N,124W | 1.1 | 1.2 | 1.7 | 1.1 | 2.1 | 0.9 | 0.4 | 390.3 | 0.7 | 1.6 | 2.2 | 1.0 | 1.3 |
| bne | 41N,97W | 0.6 | 0.8 | 1.5 | 1.2 | 1.7 | 0.7 | -0.4 | 391.7 | -0.3 | -0.8 | 2.2 | 1.3 | 0.9 |
| car | 41N,104W | 1.0 | 1.0 | 1.7 | 1.1 | 2.0 | 0.8 | -0.6 | 391.5 | -0.3 | -0.7 | 2.1 | 1.2 | 1.1 |
| hil | 40N,88W | 0.8 | 1.0 | 1.5 | 1.6 | 1.8 | 0.9 | -0.3 | 393.1 | -0.7 | -1.3 | 2.4 | 1.3 | 1.0 |
| cma | 39N,74W | 0.6 | 0.6 | 1.9 | 1.2 | 2.0 | 0.9 | -0.2 | 392.3 | 0.0 | 0.0 | 2.8 | 0.8 | 0.8 |
| sgp | 37N,98W | 0.7 | 0.9 | 1.4 | 1.2 | 1.7 | 0.8 | -0.1 | 392.1 | -0.3 | -0.5 | 1.9 | 1.1 | 1.0 |





| | | | | | | | | | | | | | | |
|---|---|---|---|---|---|---|---|---|---|---|---|---|---|---|
| sca | 33N,79W | 0.3 | 0.3 | 1.5 | 0.7 | 1.6 | 0.9 | -0.2 | 392.7 | 0.2 | -0.9 | 1.7 | 0.6 | 0.5 |
| aoa | 29N,148E | 0.2 | 0.5 | 0.6 | 0.5 | 0.8 | 0.6 | -0.2 | 392.4 | -1.2 | 0.3 | 1.4 | 0.5 | 0.6 |
| tgc | 28N,97W | 0.4 | 0.5 | 1.2 | 0.9 | 1.4 | 0.7 | 0.0 | 392.3 | 0.3 | -0.3 | 1.9 | 0.9 | 0.6 |
| rta | 21S,160W | 0.1 | 0.3 | 0.8 | 0.5 | 0.9 | 0.7 | 0.0 | 390.3 | 2.0 | 0.8 | 0.2 | 0.4 | 0.4 |
| | **average land** | **0.8** | **1.0** | **1.6** | **1.5** | **1.9** | **0.9** | **-0.1** | **391.5** | **-0.3** | **-0.0** | **2.5** | **1.3** | **1.1** |
| | | ±0.3 | ±0.5 | ±0.2 | ±0.4 | ±0.4 | ±0.1 | ±0.3 | ±1.1 | ±0.6 | ±0.9 | ±0.6 | ±0.6 | ±0.5 |
| | **ave. land, corrected** | | **0.7** | | **0.5** | **1.7** | | | | **-0.2** | **0.1** | | **1.1±0.6** | **0.6** |
| | | | ±0.5 | | ±0.0 | ±0.4 | | | | ±0.6 | ±0.9 | | | ±0.5 |
| **aoa, rta** | **average ocean** | **0.2** | **0.4** | **0.7** | **1.1** | **0.9** | **0.7** | **-0.1** | **391.4** | **0.4** | **0.6** | **0.8** | **0.5** | **0.5** |
| | | ±0.1 | ±0.1 | ±0.1 | ±0.4 | ±0.1 | ±0.1 | ±0.1 | ±1.5 | ±2.3 | ±0.4 | ±0.8 | ±0.1 | ±0.1 |
| | **e) LMT GOSAT HIPPO ocean** | | | | | | | | | | | | | |
| 2S | 30S-0S | 0.3 | 0.3 | 1.5 | 0.5 | 1.5 | 4.0 | -0.1 | 390.9 | 2.0 | -0.4 | 0.5 | 0.4 | 0.5 |
| 2N | 15S-5S | 0.4 | 0.3 | 1.6 | 0.5 | 1.6 | 4.1 | -0.1 | 390.7 | 2.2 | -0.2 | 0.4 | 0.5 | 0.5 |
| 3S | 10S-10N | 0.2 | 0.0 | 2.4 | 0.7 | 2.4 | 4.3 | -0.4 | 393.5 | -0.1 | 0.0 | 1.2 | 0.3 | 0.6 |
| 3N | 5S-10N | 0.5 | 0.3 | 1.9 | 0.5 | 1.9 | 3.9 | -0.4 | 393.4 | -0.1 | -0.4 | 0.6 | 0.4 | 0.6 |
| 4S | 10N | 0.1 | 0.5 | 1.5 | 0.5 | 1.6 | 4.6 | -0.5 | 394.5 | -3.0 | 0.2 | 0.3 | 0.4 | 0.6 |
| 4N | 15-30N | 0.3 | 0.4 | 1.5 | 1.2 | 1.5 | 4.2 | -0.3 | 393.4 | -4.2 | -0.5 | 0.5 | 0.8 | 0.5 |
| 5S | 0-20N | 0.4 | 0.6 | 1.5 | 1.4 | 1.6 | 4.5 | -0.2 | 390.7 | -0.1 | -0.4 | 0.6 | 1.0 | 0.7 |
| 5N | 10S-20N | 0.5 | 0.5 | 1.3 | 1.1 | 1.4 | 4.5 | -0.3 | 390.6 | 2.0 | 0.3 | 0.7 | 0.8 | 0.6 |
| | **average** | **0.3** | **0.4** | **1.6** | **0.8** | **1.7** | **4.3** | **-0.3** | **392.2** | **-0.2** | **-0.2** | **0.6** | **0.6** | **0.6** |
| | | ±0.1 | ±0.2 | ±0.3 | ±0.4 | ±0.3 | ±0.3 | ±0.2 | ±1.6 | ±2.4 | ±0.3 | ±0.3 | ±0.3 | ±0.6 |
| | **f) U GOSAT HIPPO ocean** | | | | | | | | | | | | | |
| 2S | 30S-0S | 0.1 | 0.6 | 0.8 | 0.4 | 1.0 | 1.6 | 0.1 | 390 | 2.6 | 0.1 | 0.3 | 0.4 | 0.7 |
| 2N | 15S-5S | 0.2 | 0.2 | 0.7 | 0.2 | 0.7 | 1.6 | 0.1 | 390.1 | 2.6 | 0.7 | 0.2 | 0.2 | 0.2 |
| 3S | 10S-10N | 0.1 | 0.3 | 0.9 | 0.6 | 1.0 | 1.7 | 0.0 | 391.6 | 0.9 | 0.3 | 1.0 | 0.6 | 0.4 |
| 3N | 5S-10N | 0.3 | 0.1 | 0.8 | 0.4 | 0.8 | 1.6 | 0.1 | 391.1 | 1.3 | 0.4 | 0.4 | 0.3 | 0.2 |
| 4S | 10N | 0.1 | 0.2 | 0.8 | 0.2 | 0.8 | 1.8 | 0.3 | 392.8 | -0.2 | 0.2 | 0.2 | 0.2 | 0.3 |
| 4N | 15-30N | 0.1 | 0.2 | 0.7 | 0.3 | 0.7 | 1.6 | -0.1 | 392.9 | -0.3 | 0.2 | 0.2 | 0.2 | 0.3 |
| 5S | 0-20N | 0.1 | 0.3 | 0.8 | 0.3 | 0.9 | 1.8 | 0.1 | 390.4 | 1.2 | -0.2 | 0.2 | 0.2 | 0.4 |
| 5N | 10S-20N | 0.2 | 0.3 | 0.7 | 0.3 | 0.8 | 1.8 | 0.1 | 390.2 | 1.8 | 0.0 | 0.3 | 0.2 | 0.4 |
| | **average** | **0.1** | **0.3** | **0.8** | **0.3** | **0.8** | **1.7** | **0.1** | **391.1** | **0.3** | **0.2** | **0.4** | **0.3** | **0.4** |
| | | ±0.1 | ±0.2 | ±0.1 | ±0.1 | ±0.1 | ±0.1 | ±0.1 | ±1.2 | ±1.1 | ±0.3 | ±0.3 | ±0.1 | ±0.1 |
| | **g) XCO$_2$ GOSAT HIPPO ocean** | | | | | | | | | | | | | |
| 2S | 30S-0S | 0.1 | 0.4 | 0.8 | 0.2 | 0.9 | 0.5 | 0.0 | 390.2 | 2.5 | 0.0 | 0.2 | 0.2 | 0.5 |
| 2N | 15S-5S | 0.1 | 0.0 | 0.7 | 0.2 | 0.7 | 0.5 | 0.0 | 390.2 | 2.5 | 0.5 | 0.2 | 0.2 | 0.2 |
| 3S | 10S-10N | 0.1 | 0.2 | 1.1 | 0.6 | 1.1 | 0.7 | -0.1 | 392.0 | 0.6 | 0.2 | 1.1 | 0.5 | 0.3 |
| 3N | 5S-10N | 0.3 | 0.0 | 0.9 | 0.4 | 0.9 | 0.5 | 0.0 | 391.6 | 1.0 | 0.2 | 0.5 | 0.2 | 0.2 |
| 4S | 10N | 0.1 | 0.3 | 0.9 | 0.2 | 0.9 | 0.8 | 0.1 | 393.2 | -0.9 | 0.2 | 0.2 | 0.2 | 0.4 |
| 4N | 15-30N | 0.1 | 0.1 | 0.7 | 0.3 | 0.8 | 0.6 | -0.1 | 393.1 | -1.2 | 0.0 | 0.2 | 0.1 | 0.2 |
| 5S | 0-20N | 0.1 | 0.3 | 0.9 | 0.5 | 1.0 | 0.7 | 0.0 | 390.5 | 0.9 | -0.2 | 0.3 | 0.3 | 0.4 |
| 5N | 10S-20N | 0.2 | 0.3 | 0.8 | 0.5 | 0.8 | 0.8 | 0.0 | 390.3 | 1.8 | 0.0 | 0.3 | 0.3 | 0.4 |
| | **average** | **0.1** | **0.2** | **0.9** | **0.4** | **0.9** | **0.6** | **0.0** | **391.4** | **0.9** | **0.1** | **0.4** | **0.3** | **0.3** |
| | | ±0.2 | ±0.2 | ±0.1 | ±0.2 | ±0.1 | ±0.1 | ±0.1 | ±1.3 | ±1.4 | ±0.2 | ±0.3 | ±0.1 | ±0.1 |
| | **h) AJAX** | | | | | | | | | | | | | |
| LMT | | 1.1 | | | 2.2 | 3.1 | 4.1 | -0.6 | 393.6 | -2.0 | -0.2 | 2.8 | | |
| LMT, corrected* | | | | | 1.9 | 2.9 | | | | -1.4 | +0.4 | | | |
| U | | 0.1 | | | 0.9 | 1.1 | 1.7 | 0.0 | 392.2 | 0.4 | 1.0 | 2.0 | | |
| XCO$_2$ | | 0.2 | | | 0.6 | 0.9 | 0.8 | -0.1 | 392.4 | -0.1 | 0.7 | 2.4 | - | - |

*AJAX profiles are co-located within 1 hour and 1 degree and therefore do not have multiple GOSAT
matches to average.





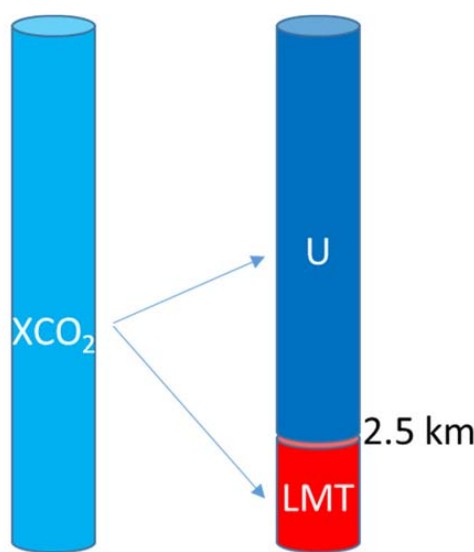

Figure 1. $XCO_2$ full column measurement (left) and the two partial columns that we introduce (right): the lowermost troposphere (LMT), a partial column from the surface to approximately 2.5 km, and the partial column above 2.5 km (U).





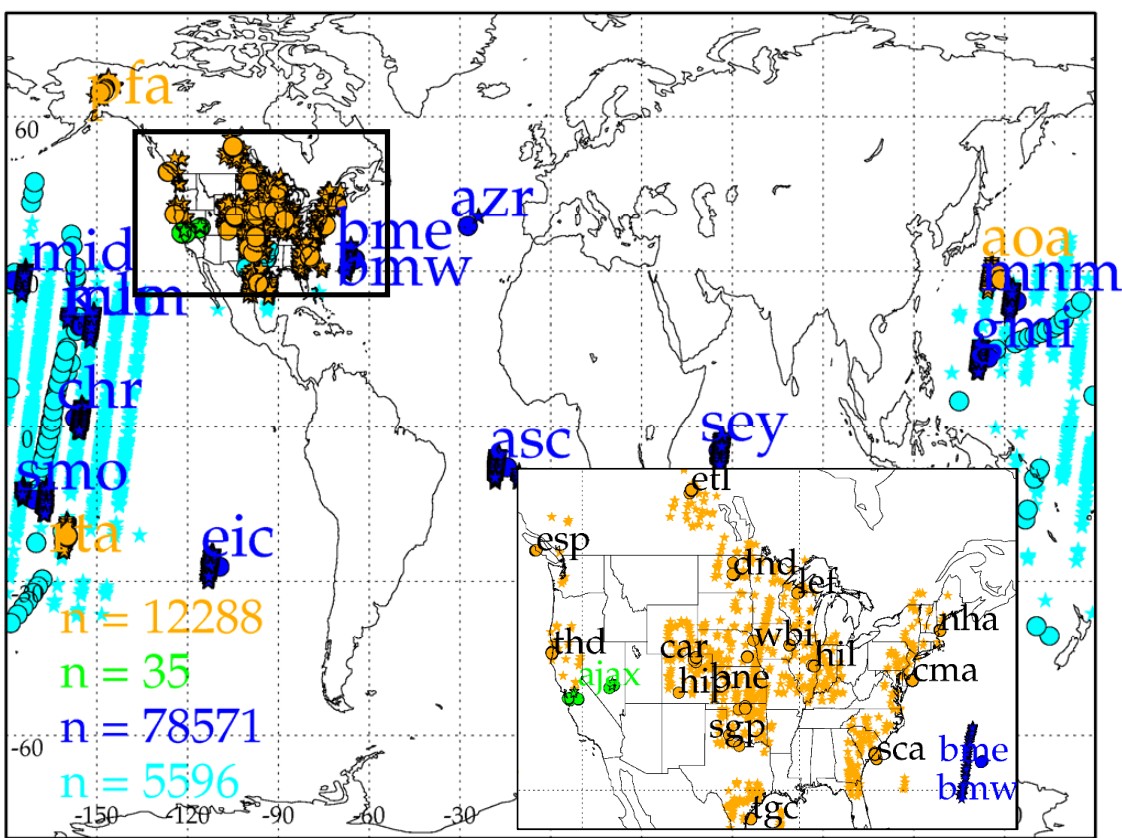

Figure 2. Validation locations. The 4 sets of validation data shown here are: ESRL aircraft (orange), which has both land (in the US) and ocean (RTA and AOA), AJAX aircraft data (green) in the western U.S., the HIPPO aircraft campaign (light blue), and remote ocean surface sites (dark blue). The matching GOSAT locations are shown as stars and the validation locations are shown as outlined circles. The number of GOSAT observations in each set are shown as the "n = " number in the lower left of the plot.





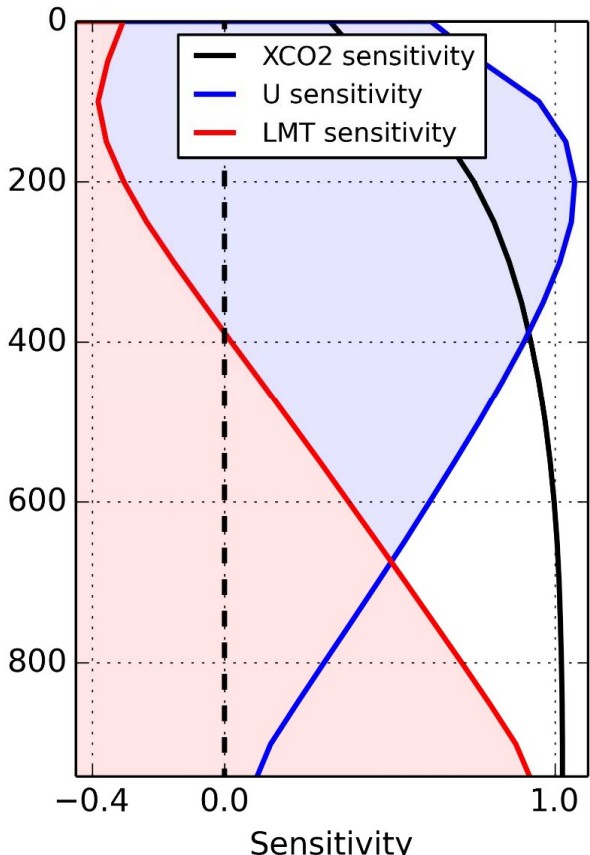

Figure 3. Sensitivity of XCO₂ (black), partitioned into the LMT (red) and U (blue) partial columns for an average land averaging kernel. The LMT sensitivity is approximately 1 near the surface and drops off steadily with decreasing pressure. The ocean averaging kernel is very similar.



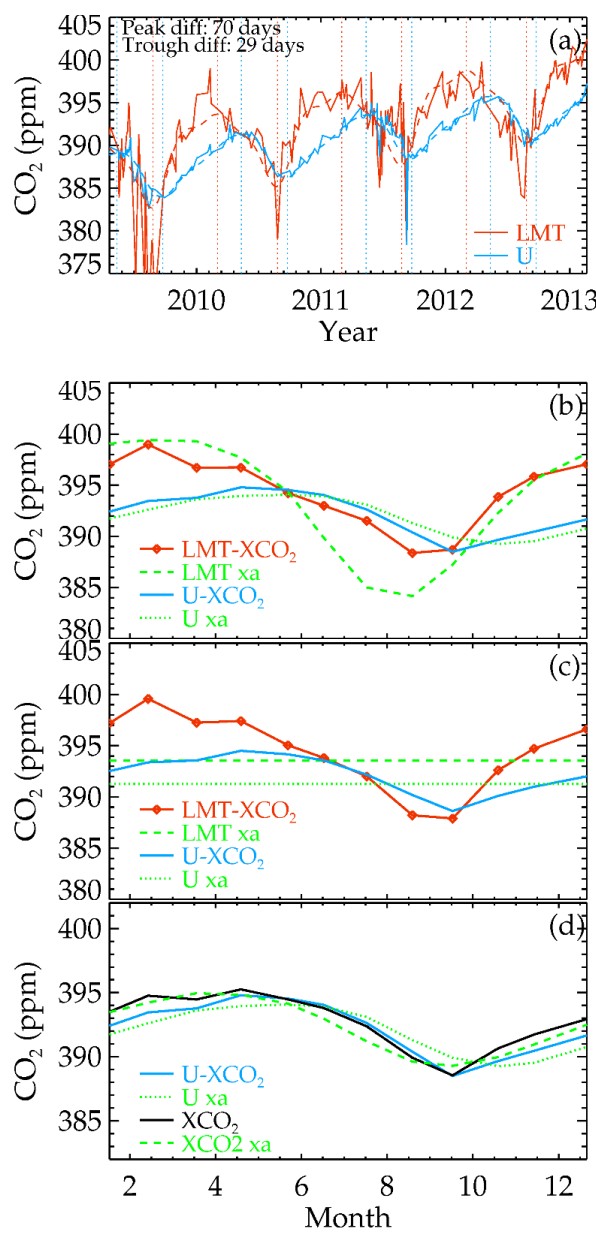

Figure 4. Simulated GOSAT retrievals from SGP aircraft profiles, Eqs. 5-6, and the GOSAT averaging kernels. (a) Time series of LMT (red) and U (blue) with dashed lines the monthly averages; (b) seasonal cycle (created by moving all measurements to be in 2012 by offsetting $CO_2$ by 2 ppm per year), averaging in 1-month increments. Dotted and dashed lines are initial guess/a priori. (c) same as (b) except that the prior is set to a constant (with 2 ppm/year secular increase), showing that LMT and U results are not strongly influenced by the prior. (d) Same as (b) but showing U (blue) vs. $XCO_2$ (black) which shows that $XCO_2$ estimates look most similar to the U partial column. Analysis in Section 3.3 shows that 70% of $XCO_2$ variations result from variations in the U partial column.





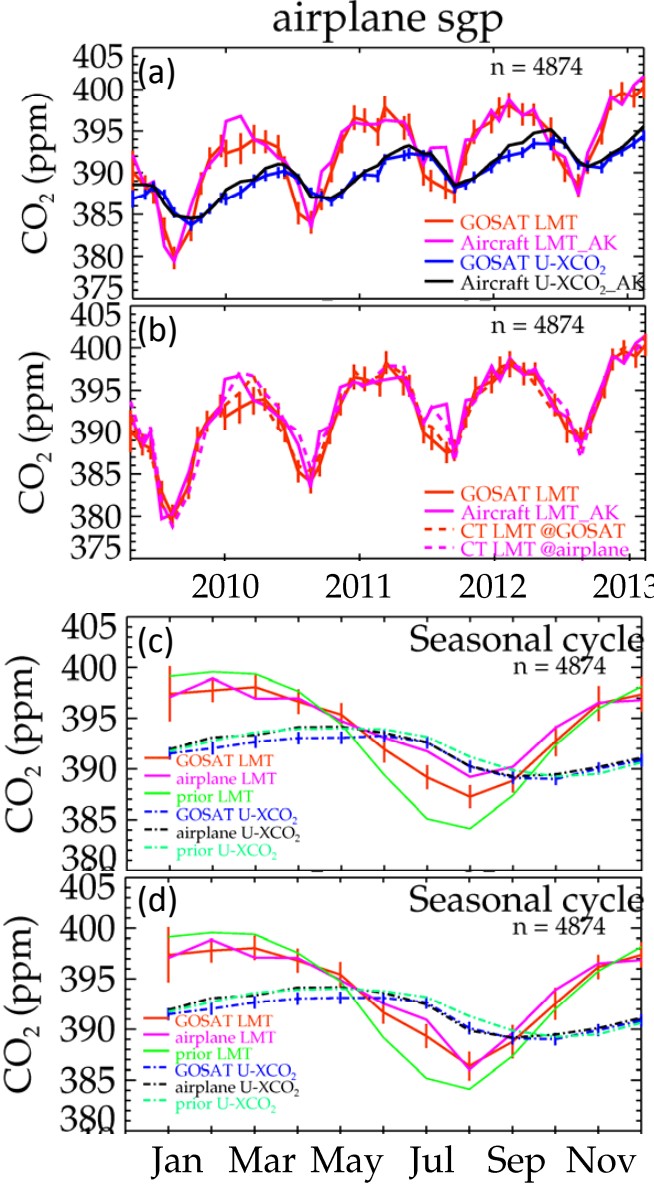

Figure 5. GOSAT versus aircraft data at the SGP site (37N, 95W). (a) aircraft LMT (pink) and U (blue) versus GOSAT LMT (red) and U (black) for monthly averages of GOSAT/airplane matches. (b) same as (a) but also showing CarbonTracker matched to GOSAT (red dotted) and CarbonTracker matched to aircraft (pink dotted) for LMT. (c) Seasonal cycle of GOSAT and airplane, same colors as top panel, and adding the priors in green. (d) Same as (c) but removing months where the CarbonTracker difference from (b) is larger than 2.5 ppm (removing months 6/9, 10/9, 5/10, 7/10, 8/10). This shows that the systematic monthly differences in panel (c) result from co-location error. In (c) and (d), the GOSAT prior overestimates the high and overestimates the drawdown. For all metrics except for the U bias, GOSAT LMT and U improve over the prior estimates.




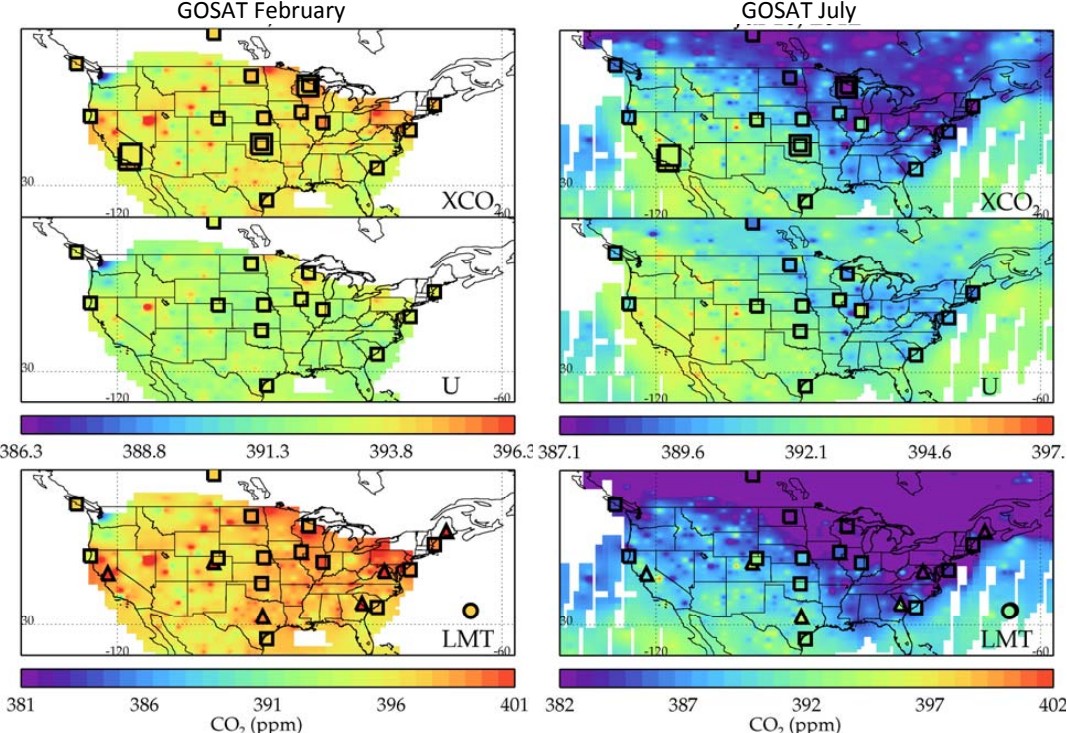

Figure 6. GOSAT $XCO_2$ (top), U (middle), and LMT (bottom) in February (left) and July (right). Aircraft with GOSAT averaging kernels are small squares, towers are triangles, remote ocean surface sites are circles, and TCCON are large squares (only shown on $XCO_2$ panels). Data is averaged over the GOSAT record (converted to 2012 by adding/subtracting 2 ppm per year).







Figure 7. Seasonal cycle at 5 sites arranged from north to south (a-e) and west to east (f-j). The
seasonal cycle minimum is marked for LMT (orange) and U (blue). The seasonal cycle shifts forward
going west and backwards going north, for both LMT and U. There is a consistent delay in the
drawdown for the U prior for (b-j) which is corrected by the retrieval. The LMT prior is consistently too
large for (i-j) with a phase shift in (j) which is corrected by the GOSAT retrieval.




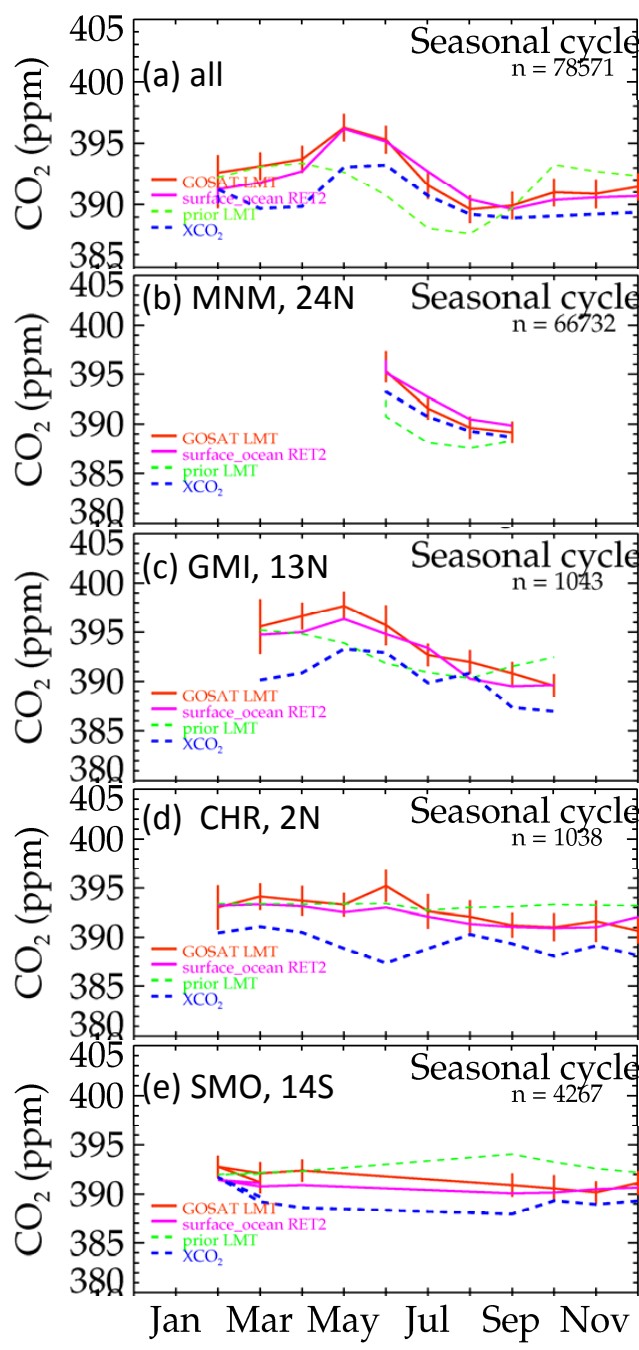

Figure 8. GOSAT LMT compared with remote ocean surface sites for different sites. GOSAT (red) compares well with surface site (pink) for the average of all sites (a) and at the four (of twelve) sites with the most matches (b-e). There is marked improvement over the prior (green). $XCO_2$ values are shown for comparison (blue dashed).





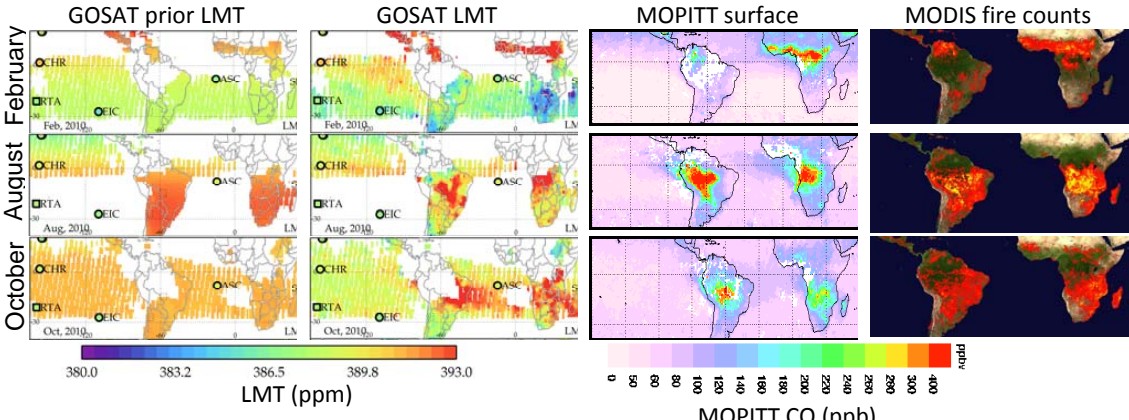

Figure 9. GOSAT LMT versus MOPITT and MODIS fire counts in for February, August, and October, 2010. GOSAT prior (left) and retrieved (second column) LMT compared with MOPITT multispectral CO (third column) and MODIS fire counts (right). The GOSAT prior is approximately constant in the southern hemisphere on monthly timescales. GOSAT LMT shows the pattern of biomass burning in South America and Africa seen by MOPITT and MODIS and reduces from the prior in the Pacific, matching surface and aircraft observations. Note that multi-spectral MOPITT retrievals have little surface sensitivity over ocean, but the outflow seen in October is seen in mid-tropospheric MOPITT.




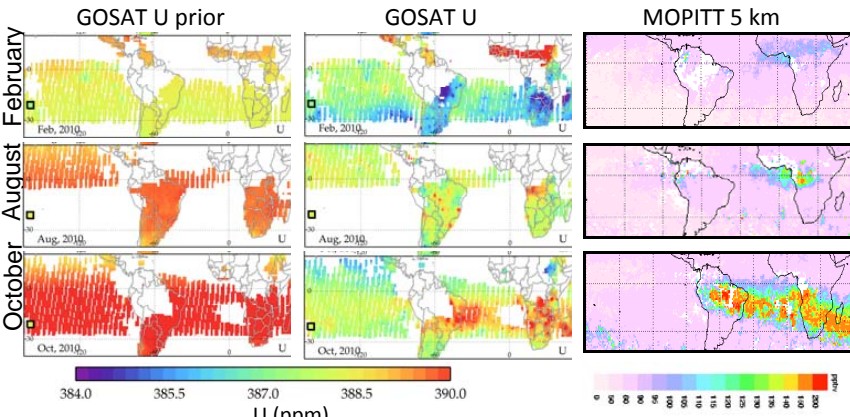

Figure 10.  GOSAT U versus MOPITT for February, August, and October, 2010.  GOSAT prior (left) and retrieved (middle) compared with MOPITT multispectral CO (right) at 5 km.  The GOSAT prior is approximately constant in the southern hemisphere on monthly timescales.  GOSAT U shows low $CO_2$ from the growing season for the Amazon and southern Africa in February.  In August, GOSAT U shows little enhancement in South America, but enhancement in southern Africa, in agreement with MOPITT. In October, GOSAT U shows enhancement over the burning regions and outflow similar to MOPITT.





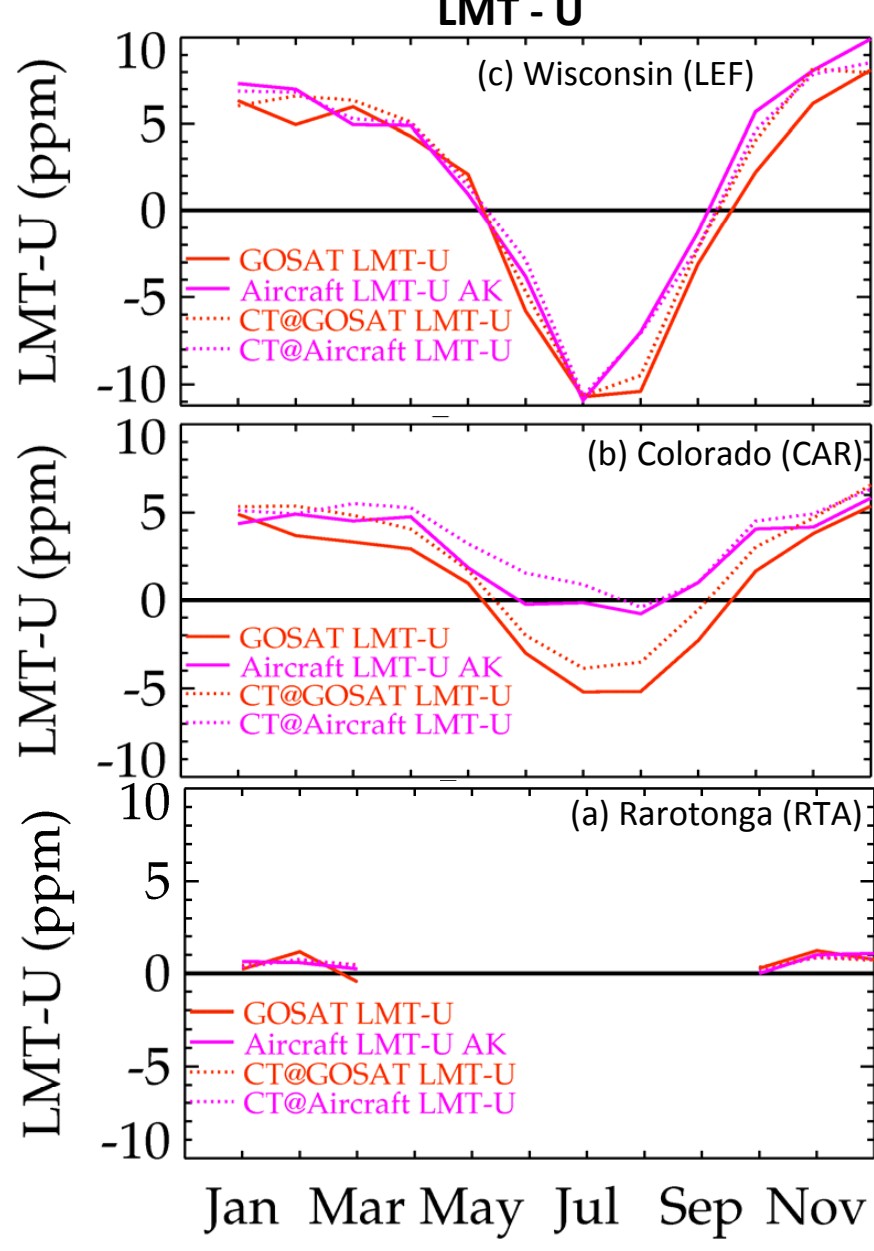

Figure 11. GOSAT LMT - U (red) versus aircraft (pink) at 3 sites. The dotted line show CarbonTracker matched to GOSAT (red dotted) or aircraft (pink dotted). Co-location error explains the discrepancies in the drawdown at CAR and LEF. At CAR the discrepancies are due to mismatch in the time of day the data is collected.





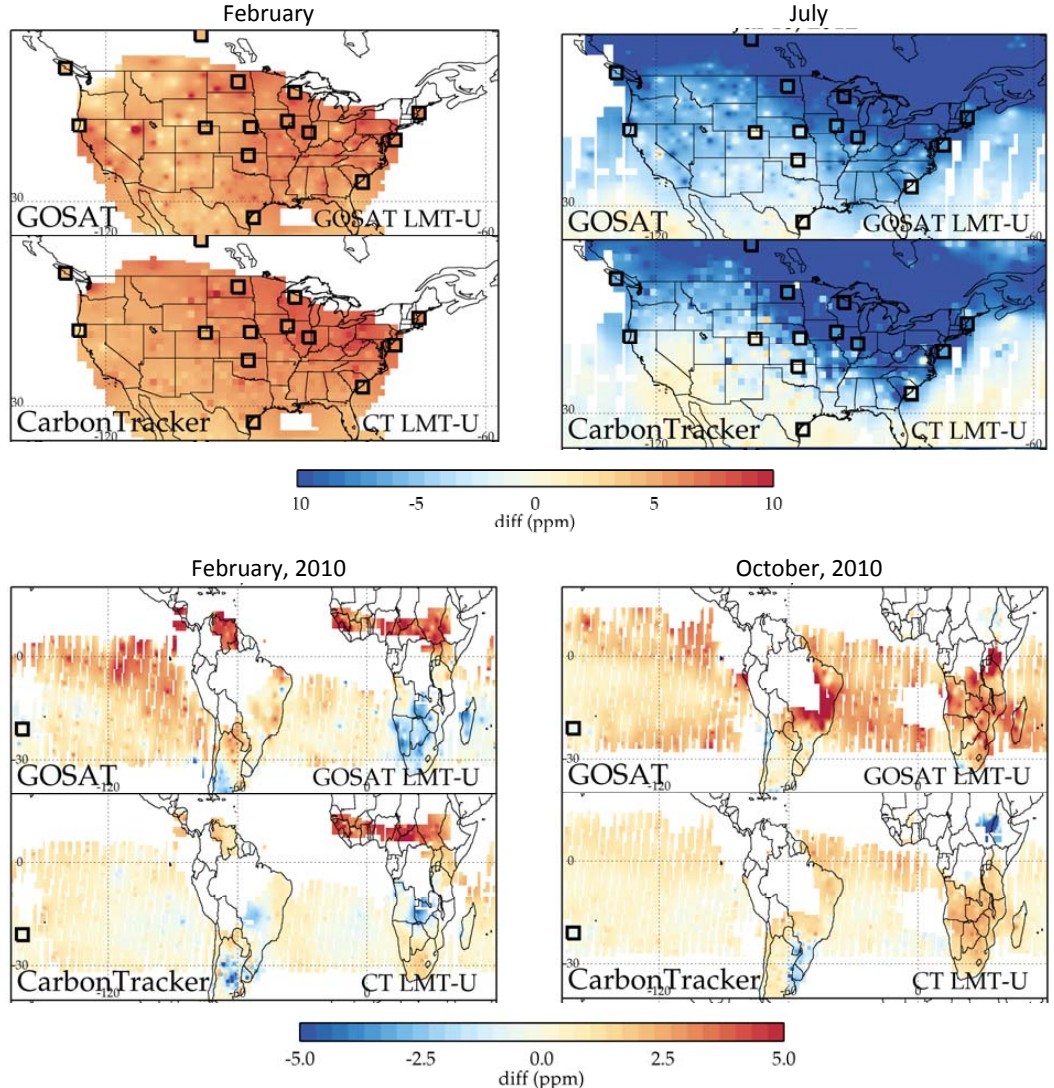

Figure 12. LMT − U differences. LMT − U diagnoses model vertical transport (Stephens, 2007) and transport of outflow (Deeter, 2013). Results shown for the U.S. (top) and South America/Africa (bottom) for two different months. GOSAT is shown on the top and CarbonTracker on the bottom. Aircraft LMT − U differences are shown in the squares. There is agreement in the U.S. where there is a lot of surface-based data to ingest into CarbonTracker, but disagreement in southern Africa during the growing season in February, the Amazon region in the biomass burning season in October, and in the outflow from Africa and South America in October.