# Peer review of "Lower-tropospheric CO2 from near-infrared ACOS-GOSAT"

_Atmospheric Chemistry and Physics, 2016_

## Referee Comment (RC1) · Anonymous Referee #2 · 4 Nov 2016

The paper addresses a very important problem, namely increasing the sensitivity of satellite CO2 observations to surface fluxes, by attempting to separate the retrieved CO2 into lower troposphere (LMT) and above (U). The study is well-conceived and executed, being theoretically sound and testing results extensively with a range of validation data. The results seem to show compelling improvements relative to prior knowledge and to XCO2, particularly in comparisons to LMT.

Major issues: I have one important question: to what extent do the results depend on 'bias correction'? The correction quoted in Appendix A is surprisingly large and variable for land, and its variability for ocean, while smaller, is considerable. Given my concern on this point I would like to see some of the comparisons to validation data (e.g. Figs 5, 7, and/or 8) shown for results without the GOSAT bias correction.

[Figure]

Further, if I read Sec 4.3 and Appendix A correctly, it seems that 'bias corrected' U is derived solely from bias corrected LMT and XCO2, i.e. does not depend on the satellite's radiance measurements directly. If true this should be made explicit.

On another matter, I found the arguments very difficult to follow in many places. The terminology is not always clearly defined, and the definitions are hard to find. Generally, error and related terms need to be better defined, and their definitions better flagged; I am very confused by the table entries: e.g. in Table 3, 'predicted error', 'GOSAT standard deviation', 'true variability'; are these directly comparable quantities? In Table 2, I am not sure what 'bias' means here; 'co-location bias', 'prior bias', and 'GOSAT bias' are not defined and their precise meaning is unclear

Also, careful proofreading would make the paper more readable.

Assuming my concern about 'bias correction' can be allayed, I recommend publication after the other issues raised have been addressed.

Specific items:

Line 51: Connor et al has been published in AMT and the reference should be updated

Line 75: unclear language: 'fluxes versus for XCO2'

Line 130: 'obspack' is used before it is defined

Line 250: language: 'chosen constrain'

Line 267: the use of 'y' as a part of the state vector is confusing to those familiar with the symbols in Rodgers, Rodgers & Connor, 2003 etc., where 'y' is the measurement vector; Rodgers & Connor use 'u' instead.

Line 416: 'CCGCRV' is used before it is defined

Line 458: there is no section 5.6

Line 463: define 'WRF-STILT'

Line 489: '()' what is this?

Line 501: 'the a': 'a' is a mathematical symbol but as used it looks like an English word

Line 507: 'be used'?

Line 598 et al: 'CAR' and many subsequent abbreviations for observing stations are never defined; sometimes they are capitalized and sometimes lower case (or do the 2 cases mean something different?); please list the stations with names and abbreviations

Line 659: 'variability for ESRL ocean' is not clear; variability of what?

Line 778: 'factor is' what is it?

Line 793-5: sounds like there is no doubt that bias correction changes the correlation; why discuss it?

Line 843: why a mismatch in airmass for these but not others?

Sect 5.4: I find the discussion of emission ratios opaque: relative DoFs? vertical sensitivity? why would variations in CO and CO2 'be either zero or solely from locally influenced fires'?

Line 926: 'CO2 and' and what?

---

## Referee Comment (RC2) · Anonymous Referee #1 · 4 Nov 2016

This paper presents a method of extracting two useful pieces of information from the ACOS retrieval of GOSAT spectra: a lower-most tropospheric partial column (LMT, 0-2.5km), and remaining partial column above (U, above 2.5km). This work represents a new, novel data product that will be useful for carbon cycle scientific studies. I recommend this paper for publication after several comments are addressed.

General Comments:

Section 4.5: The authors discuss extending the profiles above the aircraft ceiling, and then later they discuss the errors arising from this unmeasured region. AirCores, which measure from near the surface up to 20-30km, have been made at the SGP, in Sodankyla, in Boulder, and in Lauder (as well as other places). Could those measurements be useful in this discussion? Could they at least be useful to test whether

extending the top aircraft measurement through the tropopause and tacking on CarbonTracker above is indeed a sensible thing to do?

P13L516: It doesn't seem reasonable to me to use a free-tropospheric temperature coincidence criteria (such as in Wunch et al. 2011), which uses 700 hPa (∼3 km) temperatures for coincidences for the LMT or U partial columns. If I look at Figure 3, I see that the XCO2 sensitivity at 700 hPa is very high, but this is not the case for either U or LMT. The free-tropospheric temperature coincidence criteria exploits the sensitivity of the XCO2 to the free troposphere. The LMT and U are explicitly unraveling this sensitivity. Perhaps this could lead the authors to more robust "dynamical coincidence criteria" for their two new products.

P6L253: Please discuss the implications of the fact that neither LMT nor U have a full DOF.

P33L1329: Please comment on the seemingly contradictory information that the strongest bias is related to delta_grad_CO2, which seems to be a measure of how oscillatory the retrieval of the CO2 vertical profile is, yet you are able to extract sensible LMT and U partial columns from the data.

Technical Comments:

In general, there are too many acronyms, the most confusing to me were: ct_ct, the site location acronyms, etc. Please simplify if possible.

The results section contains too many numbers that are already listed in the tables (especially sections 5.1.1, 5.1.2, 5.1.4, 5.1.5). Please include only the most important numbers in the text, especially when the numbers are already in tables and figures.

The figure descriptions are too detailed in the results section. For example, only the figure captions should contain the detailed color, line style, and marker information.

P1L24: Does LMT stand for "lower-most troposphere"?

[Figure]

P2L50: Define TCCON here, as it's the first mention of it.

P3L125: Mention that you use only high gain GOSAT data, if that is indeed the case.

P4L145: "... best matches the satellite overpass" –> do you mean it best matches the overpass *time*?

P4L151: at –> within

P4L170: Is the "15" in this line spurious?

P6L227: TCCON's focus isn't solely on validation.

P6L249: which may be chosen *to* constrain. This sentence as far too many instances of the word "constrain".

P12L489: Missing something in parentheses ().

P14L579: "... the biggest detriment of use of..." –> consider rewording

P15L601: I assume this -2ppm change is at the surface, and not throughout the (partial or total) column?

P16L684: Where should the parentheses go? There is only an open bracket in both equations (11) and (12).

P19L778: The paragraph ends mid-sentence. What is the multiplication factor over ocean?

P22L926: Measurements *of* CO2 *at* 4 km and 1 km.

P24L999-1001: This second sentence of the paragraph seems redundant with the first.

Figure 4, 5, 7, 8: I don't understand the legends. For example, in Fig 4, the red dashed curve isn't the difference between LMT and XCO2, is it? Please clarify and simplify the legends.

Figure 7(f): Expand the lower limit of the y-axes so we can see the seasonal cycle

minimum from GOSAT LMT.

---

## Referee Comment (RC3) · Anonymous Referee #3 · 4 Nov 2016

This work separates the information in the CO$_2$ data from the GOSAT satellite into partial columns of CO$_2$ below approximately 2.5 km elevation ("LMT") and above 2.5 km ("U"); the sensitivity of each necessarily bleeds into the other part of the column, due to the slim vertical information content of the measurements. Bias corrections are computed for both LMT and U by comparing each partial column to aircraft data, sampled in the vertical to be consistent with the sensitivity of each GOSAT partial column. The accuracy of these bias-corrected LMT and U values are then assessed by comparing them to a) aircraft measurements (including some not used in the bias corrections), b) in situ CO$_2$ measurements at surface sites located far away from the influence of continental air, and (more qualitatively) c) to CO and fire count measurements from the MOPITT satellite over the tropics.

In general, the computed lower tropospheric CO$_2$ partial columns (LMT) compare quite

well to the independent surface $CO_2$ data at oceanic sites. The LMT patterns over Africa and South America compare quite well, qualitatively, to patterns of MOPITT CO at the surface as well as to MODIS fire counts; this would be expected if $CO_2$ produced by fires were to be the dominant source of $CO_2$ variability in the tropics. The U patterns also are similar to the seasonal outflow off those continents seen in MOPITT CO at 5 km. Finally, the bias-corrected LMT and U products agree well with aircraft $CO_2$ profile data, when they are sampled with the appropriate GOSAT averaging kernels. Since the aircraft data themselves were what the bias corrections were computed from in the first place, this is perhaps to be expected, to some extent. The real question is how well the bias-corrected LMT and U values would compare to columns computed from aircraft data not used in the bias correction calculations. This is difficult to assess here: the bias correction and validation steps used different sets of aircraft profile data that overlapped each other by about 50% (Line 1384) and which were chosen using two different coincidence criteria. It would have been better to keep the aircraft profiles used in the bias correction separate from those used in the validation.

Further complicating the validation step over land, the criterion used for time coincidence was +/- one week: in other words, aircraft measurements seeing air with a complete synoptic weather cycle or more of $CO_2$ differences were compared to the GOSAT data. This was done to increase the number of comparisons available, but the effect of this was to make it difficult to separate the impact of measurement biases from true $CO_2$ variability over land. This "co-location error" was assessed by sampling the analyzed $CO_2$ fields from CarbonTracker at the same GOSAT - aircraft location differences, but it is not completely clear from the text how these co-location corrections were used (subtracted off on a shot-by-shot basis or just statistically at the end). The overall bias correction applied to the LMT data seems to vary by about 1.7 ppm between using a 48-hour time coincidence criterion and a 1-hour criterion (Lines 1352-1357), suggesting that the uncertainty in the bias corrections is indeed large.

The main conclusion I take away from this manuscript is that the GOSAT data contain

a lot of useful information about lower tropospheric $CO_2$ variability that is mostly lost when packaged in the form of $XCO_2$ that most modelers have used to infer fluxes from up to now. A second conclusion I take is that it is not particularly easy to validate the lower-tropospheric part of the column with aircraft data, due to the sparseness of the coincidences. Overall, the authors have done a very nice job with this work. I have made some suggestions below for clarifying the text. In particular, I think the description of the values presented in the tables is currently quite confusing and could be clarified by better descriptions in the captions.

Detailed comments:

L 25: maybe call them "partial column mixing ratios"? The partial column amount could be in other units...

L 30: rather than saying "errors", could you be more precise and say "root mean square errors" (or whatever is appropriate for the statistic you are using)? We say "errors" colloquially, but it would be better to be more precise in the published work.

L 40: I would say "separated better"

L 46: Again, "uncertainty" would be better than "error" here and elsewhere in the manuscript

L 61: "model assimilation" – do you mean "data assimilation"?

L 81: Stephens et al investigated the separation of the extra-tropical northern land fluxes against the sum of the tropical land fluxes and the extra-tropical southern land fluxes, not just the southern hemisphere land fluxes, as is currently stated. Please reword this.

L 122: For clarity, I would suggest rewording this to: "ACOS-GOSAT v3.5 $XCO_2$ values from the Lite product with quality flag of 0 are used, along with the full $CO_2$ profile...". Also, immediately after, it is not clear whether you mean the averaging kernel matrix or vector by "profile averaging kernel" – maybe you could word that more clearly.

L 130-136: It might be worth noting that the aircraft data have errors themselves, but that you are considering them to be small compared to the other errors in the problem.

L 134: by "Tropopause" do you mean "tropopause height"? If so, please reword (and no caps).

L 140-141: Ok, here you define ESRL – it would be better earlier. Ditto for "ObsPack". You should mention that these are $CO_2$ measurements you are looking at.

L 145-148: This screening approach for the remote ocean data initially had me confused. I understood why "nighttime" data would not be compared to the early afternoon satellite data, but why would you throw out "marine" (which attempts to capture data from the ocean, which should be less prone to diurnal variability than the continental air"? And why would you accept "continental" air without ensuring that it is taken at approximately the same time of day as the satellite data (early afternoon)? And finally, if you accept "allvalid" data, it would seem necessary to check the time stamp on the data to ensure that it is taken at approximately the same early afternoon local time that the satellite data are taken, to avoid potential biases due to diurnal variability of $CO_2$ (especiallu for air coming off of any nearby continents). Looking at Figure 2, I see that all the sites you have selected are well off the continents, so maybe that is the answer: you are probably not seeing much diurnal variability coming off the continents. Why do you throw away "marine" air, though?

L 170: "15 $CO_2$-OMS" is this correct? (What is the "15" for?)

L 204: "the bottom 2 pressure levels" – of what, the retrieval? If so, what pressure thickness do they go up to?

L 210-211: Could you give a reference for the fire map work of Descloitres?

L 252-255: It would be interesting if you could come up with a figure that showed how these degrees of freedom split nicely at 2.5 km above the surface. Not required, but would be interesting...

L 258-262: If this is how LMT and U are defined (on a pressure grid), it would be good to note here explicitly that the 2.5 km definition that you use elsewhere in the document is an average definition, and that the split height goes up or down depending on surface pressure.

L 272: replace "26" with "$n_{interf}$=26", so that the size of $n_{interf}$ is defined.

L 274: "size $n_{interf}$ x $n_{CO2}$": this refers to the size of $A_{xy}$, not $A_{xy}(y-y_a)$, as stated.

L 280: why wouldn't the full-layer values in this vector be 1/19 = 0.0526316? If so, the rounding to 0.52 is incorrect.

L 294-295: Hopefully more than two significant digits for these quantities were used in the actual calculations. For the first value in $h_u$, I calculate 0.034483, so the 0.035 that is given isn't rounded correctly.

L 304: Why wouldn't $f_{LMT}$ = 9/38 = 0.236842 ? That is 1% different from what you have.

L 305: Why wouldn't $f_U$ = 29/38 = 0.7631579 ?

L 309: You've left off the epsilon at the end of equation 5a.

L 312: Since you have defined your pressure weighting functions, h, as row vectors instead of column vectors (that is how I interpret 2a-c, since there are no commas between the values), you should not have a transpose symbol on $h_{XCO2}$ in the equation on this line. Equations 5a and 6a are correct as written.

L 316: Since the h's are row vectors, you want to indicate one row vector over another one here: to me, the comma between them doesn't achieve that. Though it is convoluted, h = [ $h_{LMT}^T$ , $h_U^T$ ]$^T$ would be correct. Or you could put the equation on more than one line, with $h_{LMT}$ on top of $h_u$ in matrix h.

L 321-322: The equations involving h need to be fixed here, too.

L322-323: This is the first that we have heard of the noise vector having a dimension related to the dimension of the measured spectra – put that earlier after equation (1)?

L 338: If the h vectors are indeed row vectors as you defined them, then equation (7a) is correct, but the transpose symbols in (7b) are incorrect – you should have a transpose on the right-most h in each term and not on the left-most. That will get you to the 2x2 matrices you want to end up with.

L 342: Predicted errors are mentioned here and given in Table 1, but you have not yet explained how you predict these errors. Is this what is described later in Section 3.2?

L 347-350: Is this information that the prior uncertainties for LMT and U are 34 and 9 ppm given in the tables somewhere, or is this the only place you say it? If so, instead of saying "note that", you should say "We assumed a priori uncertainties of 34 and 9 ppm on LMT and U in the calculations that follow" or something like that, so that the reader knows and isn't looking for that information somewhere else.

L 354: Again, the transpose symbols should be switched here, too, if the h's really are row vectors.

L 367: It would be better to change "corr" to some symbol ("c"?) so that people don't think it is some subscript that didn't get subscripted properly...

L 377-378: I think you want these equations to apply on an element-by-element basis. As written, the equations don't mean anything, since the h's are all vectors. (i.e. you can't divide a vector by a vector) Re-write these on an element by element basis.

L 380: It would be good to show where 2.5 km would lie in terms of pressure on Figure 3, on average.

L 398: Section 3.2 in general: I think this could use a bit more explaining to help the reader understand what you are doing. If I understand correctly, you are assuming that the $CO_2$ profile measured by aircraft at SGP is the true $CO_2$ profile, then you plug this in as $x_{true}$ in equations 5 and 6 to get the LMT and U that GOSAT would have seen

there, by assuming the GOSAT averaging kernels and priors in the equations. And you assume the measurement error and interference terms are zero? Also, this approach gives what you would expect at the SGP site, not generally, correct? If would be good to explain this for the reader. Also, how high do the SGP profiles go, and what do you do to get the $CO_2$ profile above the top of the flight?

L 429: Could you please describe this flat prior some more: is it flat in space as well as time, or does it vary by latitude?

L 475: Section 4.2: You should explicitly state your key assumption here, namely that the aircraft measurements are unbiased and have a small measurement error compared to the errors in the GOSAT profiles.

L 489: It looks like you forgot to add a reference between the parentheses here – maybe an O'Dell bias correction reference?

L 506: The GOSAT and aircraft data are compared if they are within a week on either side? So clearly we are not getting synoptic scale variability – that is noise.

L 529-530: Since Figure 3 only gives pressures rather than altitudes on the vertical axis, the reader cannot, in fact, see the sensitivities that you mention, without pretty good knowledge of how altitude lines up with pressure. Perhaps you could add some rough altitudes on Figure 3 to help her/him out?

L 577: "represents the standard deviation of all the sites" – not clear what this means: the standard deviation of the site means, or the standard deviation of the measurements across all sites rolled together (without accounting for the number of measurements at each site)

Tables 2 & 3: "Ocean Surface" data are not aircraft – maybe change the title to reflect that? It would be a good idea in the caption to say what data is in each column, referring to the text if necessary for details.

Section 5.1: This description of how the biases and standard deviations across sites

are calculated is quite confusing. I didn't see any description of how the 15 observations used to get the averages in Table 3 are selected, for example. Could you maybe give an example of how the different quantities are all calculated? Or else describe it better? Also, are these all GOSAT-minus-aircraft quantities being calculated? (I want to get the sign right on the biases.)

L 612: I read a value of -0.2 ppm on Table 2, as opposed to the -0.3 ppm given here in the text.

L 620-621: it is the variability in the bias, not the overall bias itself, that decreases, right?

L 624: I read 0.7 ppm on Table 2 for GOSAT LMT ocean, not 0.5 ppm as you say here.

L 624: I read 0.1 ppm on Table 2 for GOSAT U ocean, not 0.2 ppm as you say here.

L 625-627: "The LMT location dependent bias is no worse than the $XCO_2$ location dependent bias" I don't understand this statement: on Table 2, it looks like this co-location bias (on the top line) is several times larger for LMT than for U or $XCO_2$, over both land and ocean.

L 630-633: "The variability of the LMT ct_ct bias is 0.7 ppm, and when the 5 sites with ct_ct co-location error larger than 0.5 ppm are taken out, the GOSAT LMT bias variability drops to 0.7 ppm." Did you mean to put in a lower number for the second 0.7 ppm?

L 634-635: "Taking out the top 4 GOSAT bias outliers results in a GOSAT bias variability": which column type are you discussing here? Full $XCO_2$, or LMT, or U?

L 645-646: "The surface ocean has 1.0 ppm co-location error, also including the vertical co-location." I do not understand why the ESRL ocean comparison does not show biases at least this large, because it should reflect these same vertical bias errors that are measured using the CT co-location differences. Why is that?

L 646-648: "The AJAX comparisons, which are primarily from GOSAT underflights, has a co-location error half that of the ESRL land matches, which have coincidence criteria of 7 days, and 3-5 degrees." It might be useful to discuss the implications of this. It suggests that the +/- 7 day colocation criterion is matching up GOSAT columns with aircraft columns at all different phases of synoptic scale variability over mid-latitude land regions. Thus most of this standard deviation is due to true variability in column $CO_2$ due to weather systems moving by, rather than to any sort of measurement issues. It would be interesting to do a sensitivity study looking at only the closest co-locations in time/space to see how much the mis-match drops in those cases (even if there are only a few scenes to look at).

L 662: Please state how the 15 scenes in each n=15 average were selected. Were they all at the same site? All contiguous in time?

L 696-698: "is approximated by the standard deviation of the CarbonTracker model at the validation location and time and the model at the satellite observation location and time" You mean to say that you calculate the standard deviation of the *emphdifference* of the CarbonTracker values at the two different times/locations, correct? This could maybe be worded better to get that across.

L 738: Is the "location-dependent bias" the same thing as the correlated error, $a_o$?

L 740-741: "In this paper, we find this variability to be 1.0 ppm." Where on Tables 2-4 does this 1.0 ppm number come from? Is it for $XCO_2$ or LMT or U?

L 776: "all error terms multiplied by approximately *0.6": It is not clear which error terms (predicted or calculated) you are multiplying by the extra factor, or why you chose 0.6 to multiply them by. What is the basis for this?

L 778: "Over ocean, the error correlation is the same, but the multiplication factor is " Text to end the sentence is missing here.

L 781: "one location": from the text, I thought that multiple locations were captured in

Table 3. Why do you say that only one location was used? Do you mean to say that the validation location is fixed and the GOSAT data change around if, within the colocation criteria? But that there are more than one validation site?

L 797-803: It is not clear to me whether the co-location errors have been removed from the n=1 GOSAT errors in Table 3 or not. Maybe this could be mentioned here again (if it was already mentioned somewhere else) to remind the reader. It seems like the colocation errors ought to be removed from these numbers, since that error source is artificial.

L 824-826: "At the two sites where aircraft and TCCON are jointly observed, SGP in Oklahoma and LEF in Wisconsin, $XCO_2$ agrees with TCCON rather than the aircraft." This really cannot be seen in the figure.

Figure 7, caption: The order of the sites in the figures is described incorrectly in the first sentence. Reword to: "Seasonal cycle at 5 sites arranged from west to east (a-e) and north to south (f-j)" Reword the last sentence to: "The amplitude of the LMT prior is consistently too large for (i-j)..."

L 833-834: "There is also a shift to later in the seasonal cycle minimum going either west to east". From the figure, it seems this should read "east to west".

L 843: "...there could be a mismatch in airmass." To be clear, you are mainly worried about a mismatch in the vertical, right? That might be worth mentioning. You have mismatches in the horizontal, too, due to the +/- one week matching criterion, but these would be more of an issue over land than here over the ocean, well away from the continents.

L 849-852: "In Table 4, the correlated error for surface sites is higher than for ocean aircraft comparisons (1.0 ppm vs. 0.3 ppm, respectively), and the mean bias is also higher (0.7 ppm vs. 0.1 ppm, respectively)." I am unable to find these values in Table 4. The first pair of numbers (1.0 vs. 0.3) seems to related to co-location errors, not

correlation errors. I can't find the second pair anywhere.

Figure 8: What does "RET2" refer to?

L 891-893: "We look at the $\Delta CO/\Delta CO_2$ emission ratio in May and August to check the enhancements seen in LMT relative to MOPITT in these two months has a similar ratio is seen both months." This sentence is not clear. Please reword and correct the grammar.

L 888-916: This whole discussion of emissions ratios is very hard to follow. First, you ought to state somewhere in this section that you are comparing $CO_2$ from GOSAT to CO from MOPITT and to fire counts because you want to check whether fires in this part of the world might be responsible to the $CO_2$ patterns you are seeing (yes, perhaps obvious, but you ought to state this up front to introduce why you are looking at this other data – there are other processes that can cause variability in near-surface $CO_2$ over the tropical land that are not fire-related). Second, the emissions ratios you mention on lines 900-902 do not seem to correspond to anything in Table 5: why don't they? Why bother including a Table 5 if you don't seem to be talking about the same numbers? If the emissions ratios are calculated somehow from what you have in Table 5, you need to say how you calculate them.

L 926: "Measurements $CO_2$ and 4 km and 1 km are performed..." This is not clear – please reword.

L 939-946: This should be "Table 6a-c"

L 972: "in July for CAR, SGP, and SCA...". There is disagreement at the site in Nebraska as well that seems to be due to co-location error.

L 1338-1339: "As the bias correction for simulated OCO-2 data is very similar factor" – this wording is not clear – please correct.

L 1340: "the constraint" – it is not clear what constraint is being referred to.

Table A1: It is not clear what the column headings "ocean bias correction" and "land bias correction" refer to: are these the coefficients on each term that are solved for? If so, some other wording would be better. If not, and if these are some sort of average bias correction across all the data points, then it would be good to know what the average value of each parameter in the fit (e.g. albedo_2 or co2_grad_delta) was that corresponded to the bias correction.

Table A3a and A3b: in the captions of these tables, you should indicate which quantity ($XCO_2$, LMT, or U) the bias that you are discussing pertains to.

Editorial comments:

L 28: replace "over" with "upon"?

L 36: say either "a CO/$CO_2$ emission factor" or "CO/$CO_2$ emission factors"

L 85: add a comma before "which"

L 89: replace "which" with "that"

L 107: Reword to "Measurements of $CO_2$ vertical profiles from aircraft, which extend from the surface to..."

L 109: Reword to "The second dataset that is used is $CO_2$ measurements from remote surface flask sites"

L 111: Reword to "...assuming that $CO_2$ mixing ratios in the lower 0-2.5 km are well mixed..."

Figure 2 caption, first line: Reword to: "...ESRL aircraft profiles (orange), which occur over both land..."

Figure 2 caption, third line: replace "campaign" with "data", to make this parallel with the other 4 entries.

L 120: rather than "shortwave", why not say "near-infrared" to be consistent with what

you say elsewhere in the manuscript?

L 122: You should spell out what "ACOS" stands for.

L 122: capitalize "Lite"?

L 127: replace "sufficiently" with "sufficient"

L 129: replace with NOAA/ESRL. Also, you need to define what "ESRL" stands for. This would be a good place to do so. Maybe reword to "Aircraft and ocean measurements taken by NOAA's Earth System Research Laboratory (ESRL) are obtained from an ObsPack product..."

L 171-172: for clarity, I would suggest rewording "with additional screening for the new products" to ", with additional screening performed in our analysis here," (i.e. it wasn't clear to me what "new products" referred to)

L 185: change "which" to "that"

L 218: a second right paren should be added at the end of the line

L 226: Reword to "... contain information that allow each measurement to be split into 2 or 3 vertical columns"?

L 249: Add "to" after "chosen"?

L 277: change "contributes" to "contribution"

L 282: correct to: "...bottom level is half that of the other levels"

L 403: Add a "?" after 2nd question. Maybe add an "and" before the 3rd question.

L 440: "variation * airfraction" When I first looked at this, I couldn't figure out what the asterisk was for. Maybe write out "times" or take out the space on either side of the "*"?

L 501: fix: "the a"

L 587-590: What is the implication that both co-location biases are -0.3 ppm then? That their total is -0.6 ppm and thus that the actual bias with respect to HIPPO and ESRL ocean data is actually now +0.3 ppm? Or that the the the -0.3 ppm HIPPO and ESRL bias is actually removed entirely by the co-location bias? (I.e. are the two additive?)

L 605: "Appendix B, table b": I see a Table B1 – is this what you are referring to?

L 647: Change "has" to "have"

L 684-685: There are right parentheses missing in these two equations.

L 734: fix "XCO$_2$"

L 795: add "the idea" before "that"?

---

## Author Comment (AC1) · 14 Feb 2017

Reviewer 1 Major issues: I have one important question: to what extent do the results depend on 'bias correction'? The correction quoted in Appendix A is surprisingly large and variable for land, and its variability for ocean, while smaller, is considerable. Given my concern on this point I would like to see some of the comparisons to validation data (e.g. Figs 5, 7, and/or 8) shown for results without the GOSAT bias correction.

Response: Thanks to all reviewers for their comments.

The bias correction is large and essential for the new products to be useful. We added results without bias correction in the top panel of Figure 5. Additionally, also in response to another reviewer we added comparisons where none of the observations use observations used for the derivation of the bias correction terms, also in Figure 5.

Appendix A also shows results where the validation data is split into two part, one part is used to calculate the bias and the other part tests the correction. As the new panels in figure 5 shows, and from the results in Appendix A, a) the bias correction is needed, and b) the bias correction is robust and works for observations not used to derive the correction. Also expanded statement on bias correction in the conclusions, "After bias correction, shown in detail in Appendix A, and following the same process as the bias correction for ACOS-GOSAT XCO2,..."

Further, if I read Sec 4.3 and Appendix A correctly, it seems that 'bias corrected' U is derived solely from bias corrected LMT and XCO2, i.e. does not depend on the satellite's radiance measurements directly. If true this should be made explicit.

Response: XCO2 depends on the satellite radiance, as does LMT, and so XCO2 minus LMT depends on the satellite's radiance. As the LMT and U partial columns together sum to the XCO2 column by definition, the question is whether to bias correct both LMT and U independently, in which case the corrected LMT and U will not be consistent with the operational XCO2 column, or bias correct LMT and assume the column was correctly bias corrected in the GOSAT Lite products, in which case the corrected U result can be obtained from XCO2 and LMT. Added text to section 4.3 to be more explicit in what is done and why, "After LMT is corrected, the corrected U partial column is set using Eq. 9a, so that XCO2 is consistent with LMT and U. The purpose of setting U this way is a) there is a lack of validation data for the U partial column, so bias correction would be less certain for U, and b) it is useful to have the new products consistent with a current operational column results."

On another matter, I found the arguments very difficult to follow in many places. The terminology is not always clearly defined, and the definitions are hard to find. Generally, error and related terms need to be better defined, and their definitions better flagged; I am very confused by the table entries: e.g. in Table 3, 'predicted error', 'GOSAT standard deviation', 'true variability'; are these directly comparable quantities? In Table 2, I am not sure what 'bias' means here; 'co-location bias', 'prior bias', and 'GOSAT

bias' are not defined and their precise meaning is unclear.

Response: Added Table 3 to define all the terms. The terminology in the paper is now standardized to match the table. Previously some place the terminology described what was calculated (e.g. stdev of GOSAT minus true) and sometimes the wording said what it represented (e.g. GOSAT error). Now all tables and text use the terminology shown in Table 3.

Also, careful proofreading would make the paper more readable. Assuming my concern about 'bias correction' can be allayed, I recommend publication after the other issues raised have been addressed.

Specific items: Line 51: Connor et al has been published in AMT and the reference should be updated

Response: Done, thanks.

Line 75: unclear language: 'fluxes versus for XCO2'

Response: updated wording to ". . .whereas LMT back-trajectories show a more local influence to surface fluxes..."

Line 130: 'obspack' is used before it is defined

Response: Took out the abbreviation "obspack" as it is not needed. Spelled out "Observation Package" in the paper text.

Line 250: language: 'chosen constrain'

Response: changed wording to "which may be chosen to constrain absolute values and/or the shape of the retrieved result."

Line 267: the use of 'y' as a part of the state vector is confusing to those familiar with the symbols in Rodgers, Rodgers & Connor, 2003 etc., where 'y' is the measurement vector; Rodgers & Connor use 'u' instead.

[Figure]

Response: In this paper "U" is the upper column, so having u also represent the jointly retrieved non-CO2 parameters would be confusing. Therefore, changed confusing variable name to "v", which is similar to u, and added text to identify this as "u" from Connor et al., 2008.

Line 416: 'CCGCRV' is used before it is defined

Response: Wording changed to describe CCGCRV. Since this was not used in this paper, the following amount of updated explanation seems relevant. "The seasonal cycles of each partial column are studied by converting all aircraft measurements to lie between 2012 and 2013 by applying a 2 ppm/year secular trend, and averaging by month. This method was used rather than fitting the aircraft observations using the NOAA fitting routine (CCGCRV, described in Thoning et al., 1989) to estimate the seasonal cycle shape because the aircraft observations are not sufficiently smooth to result in a consistent fit."

Line 458: there is no section 5.6

Response: Changed wording to, "Sections 4.3-4.5. . ."

Line 463: define 'WRF-STILT'

Response: Changed text to, "To compare LMT and U sensitivity to surface fluxes, we look at 10-day back-trajectory footprints created using Weather Research and Forecasting (WRF) model combined with the Stochastic Time-Inverted Lagrangian Transport (STILT) model (WRF-STILT; Nehrkorn et al., 2010)."

Line 489: '()' what is this?

Response: I took "()" out.

Line 501: 'the a': 'a' is a mathematical symbol but as used it looks like an English word

Response: In this case it is an English word. I rewrote to use "any" and changed the sentence to be more clear, "GOSAT minus true is plotted versus various GOSAT

parameters described in Appendix A, and if a slope is found for the GOSAT error versus any parameter, then a correction is applied for that parameter."

Line 507: 'be used'?

Response: Changed to "been used", "5 degrees latitude/longitude, 1 hour has previously been used for GOSAT criteria. . ."

Line 598 et al: 'CAR' and many subsequent abbreviations for observing stations are never defined; sometimes they are capitalized and sometimes lower case (or do the 2 cases mean something different?); please list the stations with names and abbreviations

Response: All stations are now changed to upper case. The station names and abbreviations are now listed in the newly added Table 1.

Line 659: 'variability for ESRL ocean' is not clear; variability of what?

Response: Changed wording to, "The next entry, "true variability" shows how much the different partial columns vary by month. The variability of LMT over land is 5.4 ppm, about double that of U or XCO2, and the variability of LMT at remote ocean sites is 1.1 ppm, about 50% larger than U or XCO2 variability.

Line 778: 'factor is' what is it?

Response: There was wording missing. Text now says, "When the error correlation between LMT and U is set to 0.6 and the diagonal error terms are multiplied by 0.6, the predicted and actual errors of LMT, U, and XCO2 errors are consistent with actual errors versus validation data. Over ocean, the diagonal error terms are multiplied by 0.3 with the error correlation between LMT and U set to 0.6."

Line 793-5: sounds like there is no doubt that bias correction changes the correlation; why discuss it?

Response: Added the sentence, "This is the first characterization of the effect of bias

correction on the actual errors." This finding was not obvious to the authors and it was particularly surprising to the authors that the bias correction changed the error correlation from negative to positive.

Line 843: why a mismatch in airmass for these but not others?

Response: Text is updated to emphasize the issue is a mismatch in vertical airmass measured between LMT and surface values. "These comparisons are not used for estimating errors or bias corrections because there is a mismatch in vertical airmass: to compare validation data and GOSAT LMT properly, validation values are needed at every pressure level at which the GOSAT LMT averaging kernel (seen in Fig. 3) is not zero. Since there is only validation data at the surface, the only option is to directly compare the surface site value to the GOSAT LMT result, rather than integrating validation results over the pressure range where GOSAT LMT is sensitive. The vertical co-location error is estimated by comparing CarbonTracker LMT (estimated with Eq 6b, where xtrue is set to the CarbonTracker value, xa is the GOSAT prior, and cross-state error and measurement error are set to zero) versus CarbonTracker surface values. "

Sect 5.4: I find the discussion of emission ratios opaque: relative DoFs? vertical sensitivity? why would variations in CO and CO2 'be either zero or solely from locally influenced fires'?

Response: The authors agree this section was confusing. This section was redone to explicitly account for degrees of freedom in the calculation of emission ratios, in the added Eq. 13. Took out discussion of comparisons between surface CO/LMT and XCO/XCO2.

Line 926: 'CO2 and' and what?

Response: Changed "and" to "at": "...the differences between CO2 at 4 km and 1 km in the assimilated model..."

Reviewer 2

Section 4.5: The authors discuss extending the profiles above the aircraft ceiling, and then later they discuss the errors arising from this unmeasured region. AirCores, which measure from near the surface up to 20-30km, have been made at the SGP, in Sodankyla, in Boulder, and in Lauder (as well as other places). Could those measurements be useful in this discussion? Could they at least be useful to test whether extending the top aircraft measurement through the tropopause and tacking on CarbonTracker above is indeed a sensible thing to do?

Response: The lead author asked for and received AirCore observations during the GOSAT period. Although there are limited direct comparisons versus AirCore (8 if using 3 degrees latitude, 5 degrees longitude, 1 week), comparisons between AirCore and the CarbonTracker model yielded useful results, showing an overall bias in the upper Trop/Stratosphere. A new section and table was added into Appendix A comparing the extension of the aircraft profiles with AirCore versus CarbonTracker:

Table A5 now compares the extension with AirCore versus CarbonTracker. AirCore measures from the surface to up to 13 hPa, meaning that all but the top GOSAT pressure level is measured. 8 AirCore observations are found to matches aircraft and GOSAT observations within 3 degrees longitude, 5 degrees latitude, and 7 days. 6 of the matches are at SGP and 2 are at CAR. For these matches, the aircraft observations are extended either with AirCore (using CarbonTracker at only the top pressure level) or CarbonTracker. The finding is similar to the finding from Table A4, that there is uncertainty in the overall bias of 0.4 ppm, but that the standard deviation is not affected by which extension is used. The reason for this bias is that the CarbonTracker stratosphere is high compared to AirCore for these 8 observations. This propagates into a high bias in the "true" U and a low bias in the "true" LMT, through the averaging kernel. Because there is uncertainty in the true value of the stratosphere that is used to extend the aircraft profiles, there is some uncertainty in the overall bias of GOSAT LMT and U on the order of 0.4 ppm. This analysis was added to the analysis in Appendix A and the conclusions of the paper. P13L516: It doesn't seem reasonable to

me to use a free-tropospheric temperature coincidence criteria (such as in Wunch et al. 2011), which uses 700 hPa (3 km) temperatures for coincidences for the LMT or U partial columns. If I look at Figure 3, I see that the XCO2 sensitivity at 700 hPa is very high, but this is not the case for either U or LMT. The free-tropospheric temperature coincidence criteria exploits the sensitivity of the XCO2 to the free troposphere. The LMT and U are explicitly unraveling this sensitivity. Perhaps this could lead the authors to more robust "dynamical coincidence criteria" for their two new products.

Response: The first author checked the NCEP temperature difference at each GOSAT pressure to see which performed the best. The temperature at the lowest NCEP level did not perform well, perhaps due to uncertainties in the NCEP product itself, and most other pressure levels performed similarly to T700 for LMT, U, or XCO2 versus aircraft data. Added text in Section 4.4, " Different variations on the dynamic coincidence criteria were tested, e.g. using temperature comparisons at the surface, averaging from the surface to 2.5 km, or weighting temperature differences by the pressure weighting function. The different temperature criteria yielded similar results overall, other than using temperature differences at the surface did not work as well as the other levels. We therefore used the standard dynamic criteria from Wunch et al., (2011b)."

P6L253: Please discuss the implications of the fact that neither LMT nor U have a full DOF.

Response: Added text, "As seen in Fig. 3, the quantity LMT + U (i.e. XCO2) has a sensitivity of 1 between the surface and 600 hPa, with sensitivity dropping off slowly with altitude above 600 hPa. The 0.8 degrees of freedom for LMT indicates the sensitivity of the retrieved LMT to the true LMT. The missing 0.2 degrees of freedom indicates sensitivity to the prior and/or sensitivity to the U part of the true profile. Since the sensitivity of LMT and U together is 1 near the surface, it is mainly sensitivity to the U part of the true profile. Similarly the 0.8 degrees of freedom for U indicates some sensitivity to the LMT and some sensitivity to the U prior."

P33L1329: Please comment on the seemingly contradictory information that the strongest bias is related to delta_grad_CO2, which seems to be a measure of how oscillatory the retrieval of the CO2 vertical profile is, yet you are able to extract sensible LMT and U partial columns from the data.

Response: The two strongest biases seen in the CO2 profile are 1) an overall bias in the retrieval shape, where LMT is biased low and U is biased high. This is easily corrected with a constant bias correction, and 2) An issue that when the retrieved LMT is high versus the prior, it is higher than it should be, and when retrieved LMT is low, it is lower than it should be. My explanation of this is that the ACOS-GOSAT constraint is extremely loose at the surface and tighter in the mid-troposphere. Delta_grad_CO2 is used to redistribute a fraction of the variability attributed to the surface to the mid-troposphere. My explanation and understanding of why this works is based on the fact that simulated retrievals (having no spectroscopic or other forward model errors) require a very similar correction factor for delta_grad_CO2 as the actual GOSAT retrievals (unpublished result). The text in Appendix A is now updated to, "By far the strongest bias is related to delta_grad_CO2. This parameter is the difference between the retrieved CO2 and a priori dry-air molefraction between the surface and vertical level 13 (approximately 630 hPa for soundings near sea level), and represents the slope of the retrieved CO2 profile in the troposphere. The resulting coefficient for this term is 0.396 for ocean and 0.310 for land soundings. This indicates that, for ocean, approximately 40% of the CO2 attributed to the surface should be moved from LMT to U, indicating that possibly (a) the troposphere is constrained too much relative to the surface, (b) an issue with the forward model, such as systematic errors in spectroscopy, or (c) some other retrieval artefact. The bias correction coefficient for delta_grad_CO2 for simulated OCO-2 land data is 0.29, very similar to the value of 0.31 for actual GOSAT data (Kulawik, unpublished result). The simulated runs have no spectroscopic error or other forward model errors, so the need for delta_grad_CO2 correction could be a consequence of way the CO2 profile is constrained in the retrieval through the constraint matrix, which allows a lot of variability near the surface and damps variability

in the mid-troposphere. This could prejudice the retrieval system to attribute radiance variations to CO2 variations at the surface rather than elsewhere in the profile, with the delta_grad_CO2 correction factor undoing this tendency. This relationship should be explored further using a simulated system with different constraint matrices."

Technical Comments: In general, there are too many acronyms, the most confusing to me were: ct_ct, the site location acronyms, etc. Please simplify if possible.

Response: In response to this, and to reviewer 1, two new tables were added. Table 1 defines all site acronyms: their location, full name, latitude, longitude, type (aircraft, surface), and how many matches are found for that site. Table 2 defines how each comparison quantity is calculated. This allows "co-location bias", for example, to be used for all subsequent tables and text, knowing that co-location bias is defined in Table 2. "ct_ct" is no longer used in the paper.

The results section contains too many numbers that are already listed in the tables (especially sections 5.1.1, 5.1.2, 5.1.4, 5.1.5). Please include only the most important numbers in the text, especially when the numbers are already in tables and figures.

Response: I took out all repeats of table entries, other than for comparisons to additional tests or results.

The figure descriptions are too detailed in the results section. For example, only the figure captions should contain the detailed color, line style, and marker information.

Response: Done. Removed about 12 statements from figure captions.

P1L24: Does LMT stand for "lower-most troposphere"?

Response: Yes. The abstract was updated to include "most", "LowerMost Tropospheric (LMT, from 0-2.5 km)"

P2L50: Define TCCON here, as it's the first mention of it.

Response: Updated.

P3L125: Mention that you use only high gain GOSAT data, if that is indeed the case.

Response: This is true. This information is in line 126, "We use both nadir land observations (looking straight down) and ocean glint observations (sunglint tracking mode), but not medium gain over land, as there is not sufficient co-located validation data to validate medium gain observations."

P4L145: "... best matches the satellite overpass" -> do you mean it best matches the overpass *time*?

Response: Yes. Updated to add "...as it best matches the satellite overpass time"

P4L151: at -> within

Response: Updated the wording to be more clear, to " The "remote oceanic" locations used are selected to have at least 97% ocean along a circle with 5 degrees radius around the location. "

P4L170: Is the "15" in this line spurious?

Response: Yes, thank you. Took this out.

P6L227: TCCON's focus isn't solely on validation.

Response: Changed wording to, " Although the TCCON observations contain information to split into 2 or 3 vertical columns, the focus of the TCCON project has been on column observations of CO2."

P6L249: which may be chosen *to* constrain. This sentence as far too many instances of the word "constrain".

Response: Took out two of the "constraint" instances. Sentence now, " The constraints may be chosen to constrain absolute values and/or the shape of the retrieved result."

P12L489: Missing something in parentheses ().

Response: Took out ()

P14L579: "... the biggest detriment of use of..." -> consider rewording

Response: Reworded to, " Biases that vary by season or location are detrimental to the use of satellite data, as the assimilation will attribute these biases to spurious fluxes."

P15L601: I assume this -2ppm change is at the surface, and not throughout the (partial or total) column?

Response: The 2 ppm change for the LMT partial column. Updated the wording to " An investigation of the -2 ppm co-location bias in the LMT partial column at CAR in July (during the drawdown) finds that the GOSAT observations are always taken 3-4 hours later than the aircraft. The CarbonTracker model estimates the effect of +3 hours as resulting in a -2 ppm change in the LMT partial column."

P16L684: Where should the parentheses go? There is only an open bracket in both equations (11) and (12).

Response: Thanks. The parentheses are not needed in these equations. Took out parentheses.

P19L778: The paragraph ends mid-sentence. What is the multiplication factor over ocean?

Response: Added the rest of the sentence, " Over ocean, the diagonal error terms are multiplied by 0.3 with the error correlation between LMT and U set to 0.6."

P22L926: Measurements *of* CO2 *at* 4 km and 1 km.

Response: Updated to add "of".

P24L999-1001: This second sentence of the paragraph seems redundant with the first.

Response: Took out the first half of the second sentence (which was redundant as the reviewer points out.)

Figure 4, 5, 7, 8: I don't understand the legends. For example, in Fig 4, the red dashed

curve isn't the difference between LMT and XCO2, is it? Please clarify and simplify the legends.

Response: Simplified legends to remove extraneous text, as previously suggested. Updated and checked all captions. In Figure 4a, the dashed lines are monthly averages, "(a) Time series of LMT (red) and U (blue) with monthly averages of LMT (red dashed) and U (blue dashed);"

Figure 7(f): Expand the lower limit of the y-axes so we can see the seasonal cycle minimum from GOSAT LMT.

Response: Updated the range for the top row.

Reviewer 3

This work separates the information in the CO2 data from the GOSAT satellite into partial columns of CO2 below approximately 2.5 km elevation ("LMT") and above 2.5 km ("U"); the sensitivity of each necessarily bleeds into the other part of the column, due to the slim vertical information content of the measurements. Bias corrections are computed for both LMT and U by comparing each partial column to aircraft data, sampled in the vertical to be consistent with the sensitivity of each GOSAT partial column. The accuracy of these bias-corrected LMT and U values are then assessed by comparing them to a) aircraft measurements (including some not used in the bias corrections), b) in situ CO2 measurements at surface sites located far away from the influence of continental air, and (more qualitatively) c) to CO and fire count measurements from the MOPITT satellite over the tropics.

In general, the computed lower tropospheric CO2 partial columns (LMT) compare quite well to the independent surface CO2 data at oceanic sites. The LMT patterns over Africa and South America compare quite well, qualitatively, to patterns of MOPITT CO at the surface as well as to MODIS fire counts; this would be expected if CO2 produced by fires were to be the dominant source of CO2 variability in the tropics. The U patterns

also are similar to the seasonal outflow off those continents seen in MOPITT CO at 5 km. Finally, the bias-corrected LMT and U products agree well with aircraft CO2 profile data, when they are sampled with the appropriate GOSAT averaging kernels. Since the aircraft data themselves were what the bias corrections were computed from in the first place, this is perhaps to be expected, to some extent. The real question is how well the bias-corrected LMT and U values would compare to columns computed from aircraft data not used in the bias correction calculations. This is difficult to assess here: the bias correction and validation steps used different sets of aircraft profile data that overlapped each other by about 50% (Line 1384) and which were chosen using two different coincidence criteria. It would have been better to keep the aircraft profiles used in the bias correction separate from those used in the validation.

Response: We agree tests should be done where the same data is not used for bias correction and validation. However, there is a very limited amount of data (which is why the +- 1 week criteria is used) and removing 50% of the data results in worse statistics, particularly with respect to assessing how data does with averaging. Comparisons were added that remove observations used for bias correction. These are shown in Figure 5f and Appendix A, tables A3a and A3b. The results with the independent data are comparable to the results using all the data.

Further complicating the validation step over land, the criterion used for time coincidence was +/- one week: in other words, aircraft measurements seeing air with a complete synoptic weather cycle or more of CO2 differences were compared to the GOSAT data. This was done to increase the number of comparisons available, but the effect of this was to make it difficult to separate the impact of measurement biases from true CO2 variability over land. This "co-location error" was assessed by sampling the analyzed CO2 fields from CarbonTracker at the same GOSAT - aircraft location differences, but it is not completely clear from the text how these co-location corrections were used (subtracted off on a shot-by-shot basis or just statistically at the end).

Response: The co-location error was subtracted off statistically. Subtracting the colocation difference for one measurement did not add a lot of value. The value was added when the CarbonTracker model showed persistent differences, such as resulting from time-of-day differences. Text added in Section 5.1.3, " The co-location error is subtracted from the correlated error, to try to remove the effect of co-location on the error estimate. This is a statistical subtraction, as no value was found in subtracting the co-location error for individual comparisons (perhaps because the model is not accurate enough to capture the co-location differences case by case)."

The overall bias correction applied to the LMT data seems to vary by about 1.7 ppm between using a 48-hour time coincidence criterion and a 1-hour criterion (Lines 1352-1357), suggesting that the uncertainty in the bias corrections is indeed large.

Response: This section is an attempt to harmonize land and ocean data. Matching land and ocean using different criteria resulted in different land-ocean differences, even for XCO2. The data used for these match-ups is the same dataset with the same bias corrections. The reviewer is correct that the differences for different matching criteria vary quite a bit (and vary even for XCO2), leading to some uncertainty in the overall bias correction.

The main conclusion I take away from this manuscript is that the GOSAT data contain a lot of useful information about lower tropospheric CO2 variability that is mostly lost when packaged in the form of XCO2 that most modelers have used to infer fluxes from up to now. A second conclusion I take is that it is not particularly easy to validate the lower-tropospheric part of the column with aircraft data, due to the sparseness of the coincidences. Overall, the authors have done a very nice job with this work. I have made some suggestions below for clarifying the text. In particular, I think the description of the values presented in the tables is currently quite confusing and could be clarified by better descriptions in the captions.

Response: The following updates were made to address reviewer 3's comments and comments from other reviewers: a) Updated the figure captions to be more precise and

to remove extraneous text, b) Updated the table entries and terminology in the text to be consistently and better named. c) Gathered all definitions of comparison terms into Table 3.

Detailed comments: L 25: maybe call them "partial column mixing ratios"? The partial column amount could be in other units...

Response: Updated to make this change throughout the text.

L 30: rather than saying "errors", could you be more precise and say "root mean square errors" (or whatever is appropriate for the statistic you are using)? We say "errors" colloquially, but it would be better to be more precise in the published work.

Response: The comparison terms are now defined in Table 3. "GOSAT bias", "GOSAT error", etc., are now defined precisely in Table 3, e.g. " GOSAT error: The standard deviation of GOSAT minus the validation data."

L 40: I would say "separated better"

Response: Updated as suggested.

L 46: Again, "uncertainty" would be better than "error" here and elsewhere in the manuscript

Response: The comparison terms, including "error", are now defined in Table 3. Added text, "...where the error is estimated as described in Table 3."

L 61: "model assimilation" - do you mean "data assimilation"?

Response: changed to " data assimilation into models"

L 81: Stephens et al investigated the separation of the extra-tropical northern land fluxes against the sum of the tropical land fluxes and the extra-tropical southern land fluxes, not just the southern hemisphere land fluxes, as is currently stated. Please reword this.

Response: I got guidance from Britt Stephens on the wording. He suggested changed wording to, "Stephens et al. (2007) show that measuring atmospheric values of carbon dioxide at 2 vertical levels better constrains model transport and partitioning between northern extratropical land fluxes and land fluxes further south..."

L 122: For clarity, I would suggest rewording this to: "ACOS-GOSAT v3.5 XCO2 values from the Lite product with quality flag of 0 are used, along with the full CO2 profile...". Also, immediately after, it is not clear whether you mean the averaging kernel matrix or vector by "profile averaging kernel" - maybe you could word that more clearly.

Response: Updated wording to, " ACOS-GOSAT v3.5 XCO2 from the lite product with quality flag of 0 are used along with the full CO2 profile, full CO2 averaging kernel matrix, and full CO2 error matrices from ancillary GOSAT files. "

L 130-136: It might be worth noting that the aircraft data have errors themselves, but that you are considering them to be small compared to the other errors in the problem.

Response: Added text, " The aircraft flask measurements themselves have errors, but these are small compared to the other errors in the comparison (e.g. co-location, extending the aircraft to the top of the atmosphere, etc.) "

L 134: by "Tropopause" do you mean "tropopause height"? If so, please reword (and no caps).

Response: Updated wording to, " The measurements are [...] extended to the tropopause pressure using the aircraft value at the highest altitude (the tropopause pressure is from the National Centers for Environmental Prediction (NCEP), http://www.esrl.noaa.gov/psd/data/gridded/data.ncep.reanalysis.html).

L 140-141: Ok, here you define ESRL - it would be better earlier. Ditto for "ObsPack". You should mention that these are CO2 measurements you are looking at.

Response: Removed ObsPack abbreviation as it was only used 1 time. Added the definition of ESRL, " Aircraft and ocean surface measurements are obtained from an observation package (co2_1_PROTOTYPE_v1.0.4b_2014-02-13) from the NOAA Earth
System Research Laboratory (ESRL))."

L 145-148: This screening approach for the remote ocean data initially had me con-
fused. I understood why "nighttime" data would not be compared to the early afternoon
satellite data, but why would you throw out "marine" (which attempts to capture data
from the ocean, which should be less prone to diurnal variability than the continental
air"? And why would you accept "continental" air without ensuring that it is taken at
approximately the same time of day as the satellite data (early afternoon)? And finally,
if you accept "allvalid" data, it would seem necessary to check the time stamp on the
data to ensure that it is taken at approximately the same early afternoon local time that
the satellite data are taken, to avoid potential biases due to diurnal variability of CO2
(especiallu for air coming off of any nearby continents). Looking at Figure 2, I see that
all the sites you have selected are well off the continents, so maybe that is the answer:
you are probably not seeing much diurnal variability coming off the continents. Why do
you throw away "marine" air, though?

Response: Agree. For these purposes "marine" is appropriate. However, for the sites
used in this paper, that option was not available. Only different types of "representative"
files are available for the stations used in this paper. I checked which files were used
and updated the wording in the paper, "For each station, there can be different options,
represented by file names; in this study "representative" files are used, with outliers
removed, if that option is available."

L 170: "15 CO2-OMS" is this correct? (What is the "15" for?)

Response: The "15" is extraneous and was removed.

L 204: "the bottom 2 pressure levels" - of what, the retrieval? If so, what pressure
thickness do they go up to?

Response: wording changed to, "We also use a measure of sensitivity to near-surface

CO computed from the trace of the averaging kernel for the lowest 200 hPa of the atmosphere..."

L 210-211: Could you give a reference for the fire map work of Descloitres?

Response: The citation is given in the text, " Giglio, L., Descloitres, J., Justice, C. O., and Kaufman, Y. J.: An enhanced contextual fire detection algorithm for MODIS, Remote Sens. Environ., 87, 273-282, 2003." Another reference is Davies et al., 2004.

L 252-255: It would be interesting if you could come up with a figure that showed how these degrees of freedom split nicely at 2.5 km above the surface. Not required, but would be interesting...

Response: Figure 2 now shows how the degrees of freedom varies with altitude. As you can see in this figure, the dropoff in sensitivity is not very sharp.

L 258-262: If this is how LMT and U are defined (on a pressure grid), it would be good to note here explicitly that the 2.5 km definition that you use elsewhere in the document is an average definition, and that the split height goes up or down depending on surface pressure.

Response: Yes, the new products are defined by GOSAT pressure levels, which are sigma levels, with the bottom 5 levels approximately the same distance above the surface regardless of pressure. For example, at a location where the GOSAT surface pressure is 684 hPa, the 5th level is 2.35 km above the surface. Taking a location where the surface pressure is 1010, the 5th level is 2.50 km above the surface. However, the dropoff is by no means sharp at 2.5 km. Updated paper wording to, " In the ACOS processing, $CO_2$ is first retrieved as a 20-level profile, where the GOSAT pressure levels are sigma levels with the 5th level approximately 2.5 km above the surface. The retrieved $CO_2$ profile averages 1.6 degrees of freedom (DOF) with about 0.8 DOF for levels 16-20 (where level 20 is the surface) and 0.8 DOF for levels 1-15 (where level 1 is at the top of the atmosphere). "

L 272: replace "26" with "ninterf =26", so that the size of ninterf is defined.

Response: Updated as suggested.

L 274: "size ninterf x nCO2": this refers to the size of Axy, not Axy(y-ya), as stated.

Response: Updated to define Axv as nCO2 x ninterf.

L 280: why wouldn't the full-layer values in this vector be 1/19 = 0.0526316? If so, the rounding to 0.52 is incorrect.

Response: Yes, thanks, updated to 0.53.

L 294-295: Hopefully more than two significant digits for these quantities were used in the actual calculations. For the first value in hu, I calculate 0.034483, so the 0.035 that is given isn't rounded correctly.

Response: I agree with the reviewer that the ideal value should be 1/38.*19/14.5 = 0.034483. The pressure weighting values in the GOSAT products are not completely ideal and vary slightly from observation to observation. The average pressure weighting values by level in the GOSAT product for XCO2 are [ 0.0264733 0.0529388 0.0529048 0.0528572 0.0528246 0.0527999 0.0527778 0.0527542 0.0527294 0.0527055 0.0526802 0.0526461 0.0526027 0.0525557 0.0525009 0.0524324 0.0523446 0.0522445 0.0521680 0.0260674] This results in the value 0.035 for the first U pressure weighting function. Updated wording in text, " The GOSAT levels are chosen such that the pressure weighting is very similar for all layers and all observations. However, the pressure weighting is not identical for all layers and all observations and the values used in our analysis are the actual values in the files, with average values shown here, rounded to 2 significant digits."

L 304: Why wouldn't fLMT = 9/38 = 0.236842 ? That is 1% different from what you have.

Response: As discussed above, the pressure weighting in the GOSAT products does

not completely match the ideal values. We used the values given in the GOSAT products rather than the expected ideal values. In this section the values are the averages over the GOSAT record averaged to 3 significant digits.

L 305: Why wouldn't fU = 29/38 = 0.7631579 ?

Response: As discussed above, the pressure weighting in the GOSAT products does not completely match the ideal values. We used the values given in the GOSAT products rather than the expected ideal values. In this section the values are the averages over the GOSAT record averaged to 3 significant digits.

L 309: You've left off the epsilon at the end of equation 5a.

Response: Corrected, thanks.

L 312: Since you have defined your pressure weighting functions, h, as row vectors instead of column vectors (that is how I interpret 2a-c, since there are no commas between the values), you should not have a transpose symbol on hXCO2 in the equation on this line. Equations 5a and 6a are correct as written.

Response: You are right. I think it is more intuitive to define h as column vectors. Now h are defined as column vectors.

L 316: Since the h's are row vectors, you want to indicate one row vector over another one here: to me, the comma between them doesn't achieve that. Though it is convoluted, h = [ hTLMT , hTU ]T would be correct. Or you could put the equation on more than one line, with hLMT on top of hu in matrix h.

Response: With h redefined as a column vector, I believe that [h_lmt, h_u]ˆT is correct.

L 321-322: The equations involving h need to be fixed here, too.

Response: Updated.

L322-323: This is the first that we have heard of the noise vector having a dimension

related to the dimension of the measured spectra - put that earlier after equation (1)?

Response: Moved this information to after Eq. 1, "...G_x is the gain matrix, size nCO2 x ns, where ns is the number of spectral points, and ? is the spectral error, size ns."

L 338: If the h vectors are indeed row vectors as you defined them, then equation (7a) is correct, but the transpose symbols in (7b) are incorrect - you should have a transpose on the right-most h in each term and not on the left-most. That will get you to the 2x2 matrices you want to end up with.

Response: Thanks. With the h vectors now defined as column vectors, moved the h-transpose to the left side of the matrices.

L 342: Predicted errors are mentioned here and given in Table 1, but you have not yet explained how you predict these errors. Is this what is described later in Section 3.2?

Response: The predicted error is given by Eq. 7, just above this. This must not be clear. Updated wording to, "Equation 7 estimates the predicted errors for LMT and U, where..."

L 347-350: Is this information that the prior uncertainties for LMT and U are 34 and 9 ppm given in the tables somewhere, or is this the only place you say it? If so, instead of saying "note that", you should say "We assumed a priori uncertainties of 34 and 9 ppm on LMT and U in the calculations that follow" or something like that, so that the reader knows and isn't looking for that information somewhere else.

Response: Yes, this is the only place that this information is given. Updated the text to, " The a priori errors, calculated from ?ˆ2 = hTSah are 34 and 9 ppm for LMT and U, respectively, which are much larger than the posterior errors, indicating that these quantities are largely unconstrained by the retrieval's prior assumption."

L 354: Again, the transpose symbols should be switched here, too, if the h's really are row vectors.

Response: With the h changed to column vectors, the h-transposes in this equation are now OK.

L 367: It would be better to change "corr" to some symbol ("c"?) so that people don't think it is some subscript that didn't get subscripted properly...

Response: Changed to "c" as suggested. This was also updated to sigma_squared, rather than sigma, on the left side of the equation.

L 377-378: I think you want these equations to apply on an element-by-element basis. As written, the equations don't mean anything, since the h's are all vectors. (i.e. you can't divide a vector by a vector) Re-write these on an element by element basis.

Response: Agree, and updated as suggested.

L 380: It would be good to show where 2.5 km would lie in terms of pressure on Figure 3, on average.

Response: Altitudes are added to Figure 3.

L 398: Section 3.2 in general: I think this could use a bit more explaining to help the reader understand what you are doing. If I understand correctly, you are assuming that the $CO_2$ profile measured by aircraft at SGP is the true $CO_2$ profile, then you plug this in as xtrue in equations 5 and 6 to get the LMT and U that GOSAT would have seen there, by assuming the GOSAT averaging kernels and priors in the equations. And you assume the measurement error and interference terms are zero? Also, this approach gives what you would expect at the SGP site, not generally, correct? If would be good to explain this for the reader. Also, how high do the SGP profiles go, and what do you do to get the $CO_2$ profile above the top of the flight?

Response: The above is correct. The text was appended to include the information above, to " This section simulates GOSAT retrievals using the linear estimate given the aircraft in situ profiles at the SGP site (37N, 95W), the GOSAT prior, and the GOSAT averaging kernels. This analysis assumes that the $CO_2$ profile measured by aircraft

at SGP (extended by the CarbonTracker model above 5.5 km) is the true $CO_2$ profile, which is then plugged into Eqs. 5 and 6 to calculate the LMT and U that GOSAT would see at the SGP site, using the GOSAT averaging kernels and priors. The measurement error and interference terms are assumed to be zero for this analysis."

L 429: Could you please describe this flat prior some more: is it flat in space as well as time, or does it vary by latitude?

Response: The "flat" prior does not vary by latitude. The text is updated to, " Using this analysis, the importance of the prior is assessed by using either a prior that is constant in location and time (with only a 2 ppm/year secular increase) or the GOSAT prior..." The secular increase is needed since different years are averaged together in the seasonal plot.

L 475: Section 4.2: You should explicitly state your key assumption here, namely that the aircraft measurements are unbiased and have a small measurement error compared to the errors in the GOSAT profiles.

Response: Updated to add this text. "The aircraft measurements are assumed to be unbiased and have small measurement error compared to the errors in the GOSAT profiles."

L 489: It looks like you forgot to add a reference between the parentheses here - maybe an O'Dell bias correction reference?

Response: Took out the parentheses.

L 506: The GOSAT and aircraft data are compared if they are within a week on either side? So clearly we are not getting synoptic scale variability - that is noise.

Response: Yes, it is true that +-1 week will not capture synoptic variability.

L 529-530: Since Figure 3 only gives pressures rather than altitudes on the vertical axis, the reader cannot, in fact, see the sensitivities that you mention, without pretty

good knowledge of how altitude lines up with pressure. Perhaps you could add some rough altitudes on Figure 3 to help her/him out?

Response: Added altitudes to Figure 3.

L 577: "represents the standard deviation of all the sites" - not clear what this means: the standard deviation of the site means, or the standard deviation of the measurements across all sites rolled together (without accounting for the number of measurements at each site)

Response: That is a good point. The mean at each site is calculated, and the +- represents the site-to-site variability of the mean. The variability of each site is captured in the mean standard deviation. The wording is updated to, " The mean represents the average of all site means, and the $\pm$ represents the standard deviation of the means averaged by site (or campaign)."

Tables 2 & 3: "Ocean Surface" data are not aircraft - maybe change the title to reflect that? It would be a good idea in the caption to say what data is in each column, referring to the text if necessary for details.

Response: Changed to "... versus validation data".

Section 5.1: This description of how the biases and standard deviations across sites are calculated is quite confusing. I didn't see any description of how the 15 observations used to get the averages in Table 3 are selected, for example. Could you maybe give an example of how the different quantities are all calculated? Or else describe it better? Also, are these all GOSAT-minus-aircraft quantities being calculated? (I want to get the sign right on the biases.)

Response: Added a new table, Table 3, to describe how each comparison is defined and calculated. This table specifies the sign (always GOSAT minus validation) and how the (n=15) are selected, "All GOSAT observations that are averaged match the same validation data point", as well as defining all other quantities.

[Figure]

L 612: I read a value of -0.2 ppm on Table 2, as opposed to the -0.3 ppm given here in the text.

Response: This value is taken out of the text in response to reviewer 2 (who said there were too many repeats of the values from the table in the text). The table value was correct.

L 620-621: it is the variability in the bias, not the overall bias itself, that decreases, right?

Response: Yes, the variability of the bias is what decreases.

L 624: I read 0.7 ppm on Table 2 for GOSAT LMT ocean, not 0.5 ppm as you say here.

Response: These numbers are also taken out from the paper text because of comments from reviewer 2.

L 624: I read 0.1 ppm on Table 2 for GOSAT U ocean, not 0.2 ppm as you say here.

Response: These numbers are also taken out from the paper text because of comments from reviewer 2.

L 625-627: "The LMT location dependent bias is no worse than the XCO2 location dependent bias" I don't understand this statement: on Table 2, it looks like this colocation bias (on the top line) is several times larger for LMT than for U or XCO2, over both land and ocean.

Response: This is referring to the GOSAT site-dependent bias, not the co-location bias.

L 630-633: "The variability of the LMT ct_ct bias is 0.7 ppm, and when the 5 sites with ct_ct co-location error larger than 0.5 ppm are taken out, the GOSAT LMT bias variability drops to 0.7 ppm." Did you mean to put in a lower number for the second 0.7 ppm?

Response: This is referring to XCO2 bias variability which does not improve when sites

with large co-location error are removed. Text updated to, "Taking out sites with large co-location bias for XCO2 does not improve the GOSAT XCO2 bias variability. Taking out the top 4 GOSAT XCO2 bias outliers results in a GOSAT XCO2 bias variability of 0.5 ppm for the remaining sites, however these 4 sites are not the same sites where LMT has bias issues, nor are these sites where CarbonTracker shows a large co-location bias.

L 634-635: "Taking out the top 4 GOSAT bias outliers results in a GOSAT bias variability": which column type are you discussing here? Full XCO2, or LMT, or U?

Response: Yes, this was not worded clearly. It is XCO2. Text updated (same text as previous response).

L 645-646: "The surface ocean has 1.0 ppm co-location error, also including the vertical co-location." I do not understand why the ESRL ocean comparison does not show biases at least this large, because it should reflect these same vertical bias errors that are measured using the CT co-location differences. Why is that?

Response: This text was taken out in response to reviewer 2, but the text and results are seen in Table 3, 4, and 5. I'm not sure what the reviewer is asking by, " I do not understand why the ESRL ocean comparison does not show biases at least this large." In Table 4, GOSAT versus ESRL surface ocean flasks is biased high by 1.1 ppm. It seems like that is what the reviewer is expecting?

L 646-648: "The AJAX comparisons, which are primarily from GOSAT underflights, has a co-location error half that of the ESRL land matches, which have coincidence criteria of 7 days, and 3-5 degrees." It might be useful to discuss the implications of this. It suggests that the +/- 7 day colocation criterion is matching up GOSAT columns with aircraft columns at all different phases of synoptic scale variability over mid-latitude land regions. Thus most of this standard deviation is due to true variability in column CO2 due to weather systems moving by, rather than to any sort of measurement issues. It would be interesting to do a sensitivity study looking at only the closest co-locations

in time/space to see how much the mis-match drops in those cases (even if there are only a few scenes to look at).

Response: The lead author tested very tight coincidences (2 degrees, 1 hour). Instead of 14,000 coincidences at a wide range of sites, there are 146 coincidences of which 89 are at SGP and 39 are at HIL. The results at these sites are compared to the looser coincidence results at the same sites. Added text to Section 5.1.2, " To reduce the co-location error, a very tight coincidence criteria of 2 degrees, 1 hour was applied, yielding 146 matches, of which 89 are at SGP and 39 at HIL. Results for these tight coincidences are compared to the looser coincidence criteria results for these sites. For the tighter coincidences, the LMT co-location error is (0.3,0.7) ppm at (SGP, HIL, respectively), and the GOSAT LMT (n=1) error is (2.6,2.5) ppm. This is compared to the looser coincidence results, where LMT co-location error is (1.8,2.2) ppm and GOSAT LMT error is (3.9,3.8) ppm. This analysis suggests that the co-location error may be underestimated. The GOSAT LMT (n=1) error in Table 5 for ESRL land (which has co-location error subtracted) is 3.4 ppm, whereas the error when the tighter coincidence criteria is applied is actually much less, 2.6 ppm. For U, the GOSAT (n=1) error is (1.0,1.4) whereas it is (1.3,1.2) for the looser criteria, which is similar, so tight versus loose coincidence criteria did not matter a lot for U comparisons."

L 662: Please state how the 15 scenes in each n=15 average were selected. Were they all at the same site? All contiguous in time?

Response: Table 3 now specifies this information: All 15 of (n=15) match the same validation data point.

L 696-698: "is approximated by the standard deviation of the CarbonTracker model at the validation location and time and the model at the satellite observation location and time" You mean to say that you calculate the standard deviation of the difference of the CarbonTracker values at the two different times/locations, correct? This could maybe be worded better to get that across.

Response: The wording is updated and has been put into Table 3. " Co-location error: The standard deviation of CarbonTracker matched to the satellite minus CarbonTracker matched to the aircraft or surface flask. This represents error introduced by the satellite not observing at the exact time and location of the validation data. The surface flasks have an additional term, the standard deviation of CarbonTracker sampled with the LMT averaging kernel and CarbonTracker sampled at the surface."

L 738: Is the "location-dependent bias" the same thing as the correlated error, ao?

Response: No. They mean different things. The correlated error is the standard deviation that does not reduce no matter how much is averaged among "identical" data. All the averaging is among data that all matches to the same validation point. Think of the correlated error like a daily (or a bit longer) regional bias that fluctuates. The location-dependent bias is the mean difference over all observations for each site. Added text to Table 3 to capture the above.

L 740-741: "In this paper, we find this variability to be 1.0 ppm." Where on Tables 2-4 does this 1.0 ppm number come from? Is it for XCO2 or LMT or U?

Response: This is supposed to be the XCO2 bias variability, which is 0.9 ppm for ESRL land in Table 3. This text was updated to, " In this paper, we find the bias variability for XCO2 0.9 ppm over land and 0.3 ppm over ocean (see Table 4). "

L 776: "all error terms multiplied by approximately *0.6": It is not clear which error terms (predicted or calculated) you are multiplying by the extra factor, or why you chose 0.6 to multiply them by. What is the basis for this?

Response: The basis for this is that the actual errors are found to be much smaller than the predicted errors. The wording is updated to make more clear what was done and why, " When the diagonal error terms are multiplied by 0.6 and the error correlation between LMT and U is set to 0.6, to match the error correlations observed versus aircraft data, the predicted LMT, U, and XCO2 errors are consistent with the actual

errors."

L 778: "Over ocean, the error correlation is the same, but the multiplication factor is " Text to end the sentence is missing here.

Response: The wording was updated to fill in the missing text. " Over ocean, multiplying the diagonal error terms by 0.3 and the error correlation between LMT and U set to 0.6 makes the predicted and actual errors agree."

L 781: "one location": from the text, I thought that multiple locations were captured in Table 3. Why do you say that only one location was used? Do you mean to say that the validation location is fixed and the GOSAT data change around if, within the colocation criteria? But that there are more than one validation site?

Response: Yes, there are many validation sites. I'm trying to state that the "bias" error is not included in the errors in Table 5, which are just the standard deviation of the differences calculated at every validation location. Changed wording to, " The errors in Table 5 represent the standard deviation of GOSAT minus validation data calculated separately for each validation location. So, the errors in Table 5 do not include the bias errors from Table 4."

L 797-803: It is not clear to me whether the co-location errors have been removed from the n=1 GOSAT errors in Table 3 or not. Maybe this could be mentioned here again (if it was already mentioned somewhere else) to remind the reader. It seems like the colocation errors ought to be removed from these numbers, since that error source is artificial.

Response: Yes the co-location errors are removed from the (now) Table 5 results. Added statement to Table 5, " The co-location errors have been subtracted out from the GOSAT errors." Another similar statement in the text describing Table 5.

L 824-826: "At the two sites where aircraft and TCCON are jointly observed, SGP in Oklahoma and LEF in Wisconsin, XCO2 agrees with TCCON rather than the aircraft."

This really cannot be seen in the figure.

Response: Agree. I don't think that can be seen in the figure. Moved this discussion to Section 5.1.4, where biases and comparisons to previous work are discussed. Added this new paragraph, " We also compare GOSAT XCO2 comparisons aircraft and GOSAT XCO2 comparisons to TCCON at the two sites where both validation data are co-located, Park Falls, Wisconsin (LEF), and Lamont, Oklahoma (SGP). Averaging over these two sites, the GOSAT bias versus aircraft in this work is -0.4 ppm. The GOSAT bias versus TCCON in Kulawik et al. (2016) for these two sites is -0.1 ppm. The difference between these comparisons is on the same order as the uncertainty introduced by profile extension, as discussed in Appendix A."

Figure 7, caption: The order of the sites in the figures is described incorrectly in the first sentence. Reword to: "Seasonal cycle at 5 sites arranged from west to east (a-e) and north to south (f-j)" Reword the last sentence to: "The amplitude of the LMT prior is consistently too large for (i-j)..."

Response: Agree, updated caption. The last sentence was taken from the figure caption in response to reviewer 2.

L 833-834: "There is also a shift to later in the seasonal cycle minimum going either west to east". From the figure, it seems this should read "east to west".

Response: Agree. This sentence was also removed from the caption and is now corrected in the text (Section 5.2)

L 843: "...there could be a mismatch in airmass." To be clear, you are mainly worried about a mismatch in the vertical, right? That might be worth mentioning. You have mismatches in the horizontal, too, due to the +/- one week matching criterion, but these would be more of an issue over land than here over the ocean, well away from the continents.

Response: Correct, the issue is vertical mismatch. Added text to specify this. Agree

that there are also positive things about remote ocean sites, such as less synoptic variability. However, the vertical airmass mismatch, I believe, would likely cause systematic errors that would not be expected from + and - temporal and spatial mismatching at other sites. It is also useful to have these sites not used for bias correction and used as a validation test.

L 849-852: "In Table 4, the correlated error for surface sites is higher than for ocean aircraft comparisons (1.0 ppm vs. 0.3 ppm, respectively), and the mean bias is also higher (0.7 ppm vs. 0.1 ppm, respectively)." I am unable to find these values in Table 4. The first pair of numbers (1.0 vs. 0.3) seems to related to co-location errors, not correlation errors. I can't find the second pair anywhere.

Response: Yes, this is co-location error (fixed text). The values are updated for the second comparison also, " In Table 6, the co-location error for surface sites is higher than for ocean aircraft comparisons (1.0 ppm vs. 0.3 ppm, respectively), and the GOSAT bias versus ocean surface sites in Table 4 is also higher (1.1 ppm vs. 0.1 ppm, respectively)."

Figure 8: What does "RET2" refer to?

Response: An internal label now taken off the plot.

L 891-893: "We look at the _CO/_CO2 emission ratio in May and August to check the enhancements seen in LMT relative to MOPITT in these two months has a similar ratio is seen both months." This sentence is not clear. Please reword and correct the grammar.

Response: Removed the last part of the sentence. The sentence now reads, " We look at the quantitative values for the enhancements and background values for surface CO and LMT CO2 in Table 7 and use this to estimate ?CO/?CO2 emission ratios for May and August."

L 888-916: This whole discussion of emissions ratios is very hard to follow. First, you

ought to state somewhere in this section that you are comparing CO2 from GOSAT to CO from MOPITT and to fire counts because you want to check whether fires in this part of the world might be responsible to the CO2 patterns you are seeing (yes, perhaps obvious, but you ought to state this up front to introduce why you are looking at this other data - there are other processes that can cause variability in near-surface CO2 over the tropical land that are not fire-related).

Response: This suggestion was added at the start of Section 5.4, " The SH region is of particular interest for validation as the GOSAT prior is nearly spatially and vertically constant, varying primarily by month. Figures 9 and 10 compare GOSAT LMT and U partial column mixing ratios, respectively, to MOPITT multispectral CO retrievals and MODIS fire counts, to see how much fires in this part of the world are responsible for the patterns seen in the GOSAT partial columns."

Second, the emissions ratios you mention on lines 900-902 do not seem to correspond to anything in Table 5: why don't they? Why bother including a Table 5 if you don't seem to be talking about the same numbers? If the emissions ratios are calculated somehow from what you have in Table 5, you need to say how you calculate them.

Response: Agree that this section needed improvement. Added Equation 13, which shows how the emissions ratio is estimated using the information in Table 7 (Table 5 is now Table 7). The degrees of freedom is now included explicitly in the emission ratio estimate. The emissions ratios that were re-calculated using Eq. 13 did not agree with the previous calculations. The paragraph now reads, " The emission ratio is estimated using Eq. 13 with the information shown in Table 7. The emission ratio estimate ranges from 6-7% in May and 10-15% in August, for the different MOPITT sensitivity groupings. The emission ratio seen by the MOPITT and GOSAT LMT products are compared to those estimated from aircraft observations over tropical forests by Akagi et al. (2011, Table 1), which is 8.8%. The MOPITT/GOSAT ratio is similar to Akagi et al. (2011), but 2-3% lower in May, and 1-6% higher in August."

L 926: "Measurements CO2 and 4 km and 1 km are performed..." This is not clear - please reword.

Response: Changed wording to, " Aircraft observations of CO2 at 4 km and 1 km are done only at a few sites worldwide, primarily in the U.S. "

L 939-946: This should be "Table 6a-c"

Response: Updated to Table 8a-c (2 new tables added).

L 972: "in July for CAR, SGP, and SCA...". There is disagreement at the site in Nebraska as well that seems to be due to co-location error.

Response: Agree, BNE also has a bias and the same bias is seen in the CarbonTracker-based co-location bias. There is the same time-of-day difference for validation versus GOSAT observations.

L 1338-1339: "As the bias correction for simulated OCO-2 data is very similar factor" - this wording is not clear - please correct.

Response: Reworded to, " The bias correction for delta_grad_CO2 for simulated OCO-2 land data is 0.29, very similar to 0.31 for actual GOSAT data (Kulawik, unpublished result). The simulated runs have no spectroscopic error or other radiative transfer errors, so the need for delta_grad_CO2 correction could be a consequence of way the CO2 profile is constrained in the retrieval through the constraint matrix, which allows a lot of variability near the surface and damps variability in the mid-troposphere. This could prejudice the retrieval system to attribute radiance variations to CO2 variations at the surface rather than elsewhere in the profile, with the delta_grad_CO2 correction factor undoing this tendency. This relationship should be explored further using a simulated system with different constraint matrices."

L 1340: "the constraint" - it is not clear what constraint is being referred to.

Response: Wording updated, see above response.

Table A1: It is not clear what the column headings "ocean bias correction" and "land bias correction" refer to: are these the coefficients on each term that are solved for? If so, some other wording would be better. If not, and if these are some sort of average bias correction across all the data points, then it would be good to know what the average value of each parameter in the fit (e.g. albedo_2 or co2_grad_delta) was that corresponded to the bias correction.

Response: Yes, the bias correction was done separately for GOSAT nadir mode (over land) and glint mode (over ocean). Wording added to the text and tables to make this clear, e.g. " The ACOS-GOSAT XCO2 product undergoes bias correction (Wunch et al., 2011) which significantly improves the errors (Kulawik, 2016). We apply this same technique to correct the LMT product. Land nadir mode ("land") and ocean glint mode ("ocean") are bias corrected separately for LMT. "

Table A3a and A3b: in the captions of these tables, you should indicate which quantity (XCO2, LMT, or U) the bias that you are discussing pertains to.

Response: It is for LMT. Added "LMT" to the title for both Tables A3a and A3b.

Editorial comments: L 28: replace "over" with "upon"?

Response: changed.

L 36: say either "a CO/CO2 emission factor" or "CO/CO2 emission factors"

Response: Changed to "a CO/CO2 emission factor"

L 85: add a comma before "which"

Response: Done

L 89: replace "which" with "that"

Response: Done

L 107: Reword to "Measurements of CO2 vertical profiles from aircraft, which extend

from the surface to..."

Response: Reworded as suggested.

L 109: Reword to "The second dataset that is used is CO2 measurements from remote surface flask sites"

Response: reworded as suggested.

L 111: Reword to "...assuming that CO2 mixing ratios in the lower 0-2.5 km are well mixed..."

Response: reworded as suggested.

Figure 2 caption, first line: Reword to: "...ESRL aircraft profiles (orange), which occur over both land..."

Response: reworded as suggested.

Figure 2 caption, third line: replace "campaign" with "data", to make this parallel with the other 4 entries.

Response: changed to "HIPPO aircraft profiles" to be consistent with the previous suggestion.

L 120: rather than "shortwave", why not say "near-infrared" to be consistent with what you say elsewhere in the manuscript?

Response: reworded as suggested

L 122: You should spell out what "ACOS" stands for.

Response: changed to " The Atmospheric CO2 Observations from Space (ACOS) v3.5 processing of GOSAT XCO2 observations are used..."

L 122: capitalize "Lite"?

Response: capitalized "Lite"

L 127: replace "sufficiently" with "sufficient"

Response: Changed to " as there is not a sufficient amount of co-located validation data to validate medium gain observations."

L 129: replace with NOAA/ESRL. Also, you need to define what "ESRL" stands for. This would be a good place to do so. Maybe reword to "Aircraft and ocean measurements taken by NOAA's Earth System Research Laboratory (ESRL) are obtained from an ObsPack product..."

Response: Changed as suggested.

L 171-172: for clarity, I would suggest rewording "with additional screening for the new products" to ", with additional screening performed in our analysis here," (i.e. it wasn't clear to me what "new products" referred to)

Response: Changed wording to, " Due to the GOSAT glint coverage span of about 40 degrees, and after applying quality screening,..."

L 185: change "which" to "that"

Response: Changed as suggested.

L 218: a second right paren should be added at the end of the line

Response: Changed as suggested.

L 226: Reword to "... contain information that allow each measurement to be split into 2 or 3 vertical columns"?

Response: Changed as suggested.

L 249: Add "to" after "chosen"?

Response: Changed as suggested.

L 277: change "contributes" to "contribution"

Response: Changed as suggested.

L 282: correct to: "...bottom level is half that of the other levels"

Response: Changed as suggested.

L 403: Add a "?" after 2nd question. Maybe add an "and" before the 3rd question.

Response: Added in ? after the second question. Reformatted into a list. I don't know how to add an "and" in properly but think it is OK in list form.

L 440: "variation * airfraction" When I first looked at this, I couldn't figure out what the asterisk was for. Maybe write out "times" or take out the space on either side of the "*"?

Response: Wrote out "times" as suggested.

L 501: fix: "the a"

Response: took out "the"

---

## Author Response (AR2)

Thanks for the suggestions.

Response:

Updated acknowledgements to add, "We thank the three anonymous reviewers whose comments and suggestions significantly improved this paper."

Modified tables that went past the margins to fit within the margins.